# Somatic mutation and selection at population scale

Andrew R. J. Lawson[1,11], Federico Abascal[1,11], Pantelis A. Nicola[1,11], Stefanie V. Lensing[1,2], Amy L. Roberts[3], Georgios Kalantzis[4], Adrian Baez-Ortega[1], Natalia Brzozowska[1], Julia S. El-Sayed Moustafa[3], Dovile Vaitkute[3], Belma Jakupovic[3], Ayrun Nessa[3], Samuel Wadge[3], Marc F. Österdahl[3], Anna L. Paterson[5], Doris M. Rassl[6], Raul E. Alcantara[1,7], Laura O'Neill[1], Sara Widaa[1], Siobhan Austin-Guest[2], Matthew D. C. Neville[1], Moritz J. Przybilla[1], Wei Cheng[2], Maria Morra[2], Lucy Sykes[2], Matthew Mayho[2], Nicole Müller-Sienerth[2], Nicholas Williams[1], Diana Alexander[1], Luke M. R. Harvey[1], Thomas Clarke[1], Alex Byrne[1], Jamie R. Blundell[8], Matthew D. Young[7], Krishnaa T. A. Mahbubani[9,10], Kourosh Saeb-Parsy[9,10], Hilary C. Martin[4], Michael R. Stratton[1], Peter J. Campbell[1,7], Raheleh Rahbari[1], Kerrin S. Small[3] & Iñigo Martincorena[1 ✉]

As we age, many tissues become colonized by microscopic clones carrying somatic driver mutations[1–7]. Some of these clones represent a first step towards cancer whereas others may contribute to ageing and other diseases. However, our understanding of this phenomenon remains limited due to the challenge of detecting mutations in small clones. Here we introduce a new version of nanorate sequencing (NanoSeq)[8], a duplex sequencing method with an error rate lower than five errors per billion base pairs, which is compatible with whole-exome and targeted capture. Deep sequencing of polyclonal samples with single-molecule sensitivity simultaneously profiles large numbers of clones, providing accurate mutation rates, signatures and driver frequencies in any tissue. Applying targeted NanoSeq to 1,042 non-invasive samples of oral epithelium and 371 blood samples from a twin cohort, we report an extremely rich selection landscape, with 46 genes under positive selection in oral epithelium, more than 62,000 driver mutations and evidence of negative selection in essential genes. High-resolution maps of selection across coding and non-coding sites are obtained for many genes: a form of in vivo saturation mutagenesis. Multivariate regression models enable mutational epidemiology studies on how exposures and cancer risk factors, such as age, tobacco or alcohol, alter the acquisition or selection of somatic mutations. Accurate single-molecule sequencing provides a powerful tool to study early carcinogenesis, cancer prevention and the role of somatic mutations in ageing and disease.

In the past decade, increasingly sensitive sequencing methods have begun to unravel the somatic mutation landscapes of human tissues. They have revealed that mutations accumulate linearly with age in a tissue-specific manner[2,9,10], largely due to endogenous mutational processes but also influenced by mutagen exposures, germline variation and disease states. These studies have also revealed that as we age our tissues are colonized by myriad clones carrying positively selected driver mutations[1–7]. These clones provide a window into early carcinogenesis and may contribute to other diseases. However, most clones are microscopic and methods to detect them, such as laser microdissection[11] or single-cell cultures[12], are low throughput, which has limited our understanding to a few tissues and small donor cohorts.

An alternative approach is error-corrected bulk sequencing[13], such as duplex sequencing, which combines information from both strands of each original DNA molecule to eliminate sequencing and amplification errors[14–17]. Theoretically, duplex error rates should approximate the polymerase error rate squared (fewer than $10^{-8}$ errors per base pair (bp)). However, they are typically higher (around $10^{-7}$) due to interstrand error copying during library preparation[8]. We have previously described NanoSeq, a protocol that avoids error transfer by using restriction enzyme fragmentation without end repair, and dideoxynucleotides during A-tailing, achieving error rates below $5 \times 10^{-9}$ errors per bp in single DNA molecules[8]. As this rate is two orders of magnitude lower than the mutation burden of normal adult cells (around $10^{-7}$)[2,9,12],

[1]Cancer, Ageing and Somatic Mutation Programme, Wellcome Sanger Institute, Hinxton, UK. [2]Sequencing Operations, Wellcome Sanger Institute, Hinxton, UK. [3]Department of Twin Research and Genetic Epidemiology, King's College London, London, UK. [4]Human Genetics Programme, Wellcome Sanger Institute, Hinxton, UK. [5]Department of Histopathology, Cambridge University Hospitals NHS Foundation Trust, Cambridge, UK. [6]Department of Pathology, Royal Papworth Hospital NHS Foundation Trust, Cambridge, UK. [7]Quotient Therapeutics Limited, Saffron Walden, UK. [8]Early Cancer Institute, University of Cambridge, Cambridge Biomedical Campus, Cambridge, UK. [9]Department of Surgery, University of Cambridge, Cambridge, UK. [10]NIHR Cambridge Biomedical Research Centre, Cambridge Biomedical Campus, Cambridge, UK. [11]These authors contributed equally: Andrew R. J. Lawson, Federico Abascal, Pantelis A. Nicola. ✉e-mail: im3@sanger.ac.uk

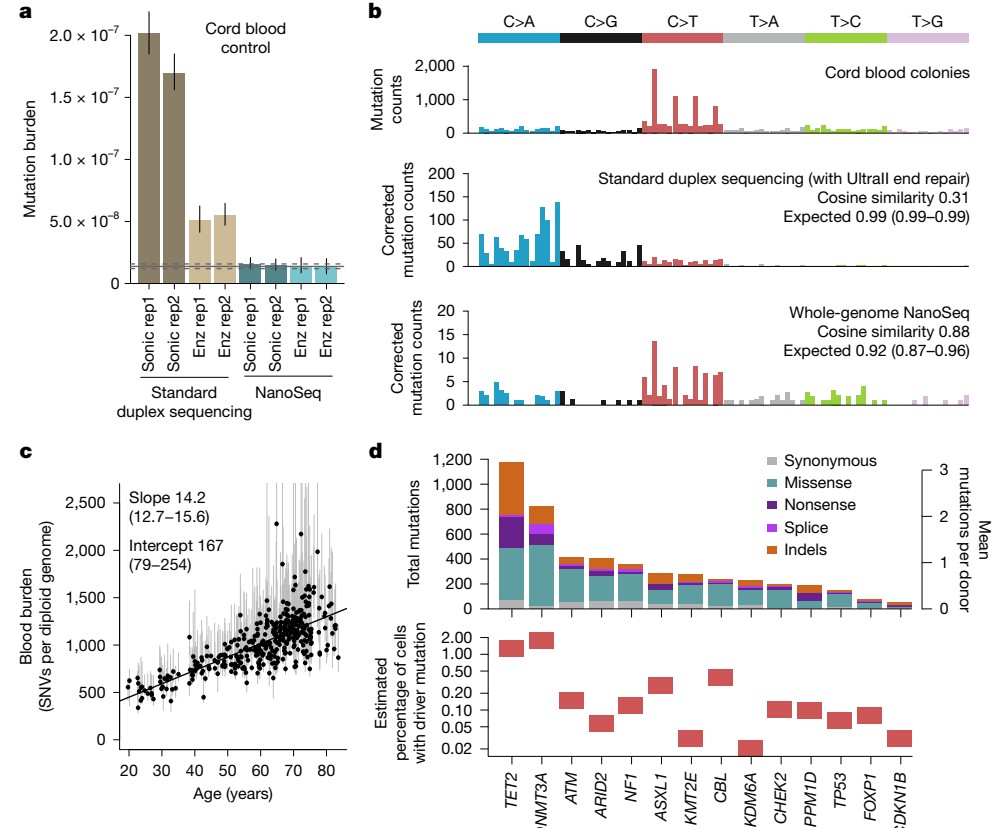

**Fig. 1 | Technical and biological validation of targeted NanoSeq. a**, Genome-wide SNV burden estimates, as mutations per base pair, for cord blood granulocytes, sequenced using four different fragmentation and library preparation protocols. Error bars show 95% Poisson CIs. Horizontal lines denote the observed burden (solid) and 95% Poisson CIs (dashed) for cord blood granulocytes sequenced by restriction enzyme NanoSeq[8]. Duplex sequencing and NanoSeq burdens are corrected for missed embryonic mutations, as described in ref. 8. Enz, enzymatic fragmentation; sonic, sonication; rep, replicate. **b**, Trinucleotide mutational spectra of single-cell derived cord blood colonies from a previous study[56] (top), and cord blood granulocytes sequenced using standard duplex sequencing (middle) and whole-genome NanoSeq (bottom). Duplex sequencing and NanoSeq spectra are corrected by the ratio of genomic to observed trinucleotide frequencies. Cosine similarity

95% CIs are calculated by drawing 1,000 random samples from each observed profile, as described in ref. 8. **c**, Linear regression of genome-wide SNV burdens (estimated using targeted NanoSeq) for whole-blood samples from 371 donors against donor age. Points and their associated error bars represent the point estimates and 95% Poisson bootstrapping CIs of passenger mutation burdens for each sample. Slope and intercept of the fitted model (point estimates and 95% CIs) are indicated. One sample was excluded due to the ratio between upper and lower confidence limits being greater than five. **d**, Mutation counts for each coding mutation consequence (top) and estimated mutant cell fractions (bottom) for 14 genes under significant positive selection in blood. Mutant cell fractions are shown for individuals aged 65–85 whose blood samples were not selected on the basis of their oral epithelium results.

mutations are accurately detected from single DNA molecules, enabling the quantification of mutation rates and signatures in any tissue. However, this protocol is unsuitable for driver discovery, as restriction enzymes only provide partial coverage of the human genome.

## Full-genome nanorate sequencing

To achieve full-genome representation while retaining ultra-low error rates, here we introduce two alternative fragmentation methods: (1) sonication followed by exonuclease blunting; and (2) enzymatic fragmentation in a buffer optimized to eliminate error transfer between strands. As in the original NanoSeq protocol[8], dideoxynucleotides prevent the extension of single-stranded nicks and quantitative PCR followed by a library bottleneck is used to optimize duplicate rates to maximize cost efficiency. After extensive optimization (Supplementary Note 1), we achieved full-genome coverage (Extended Data Fig. 1a,b) with similar efficiency and error rates as the original NanoSeq protocol.

To demonstrate their accuracy, we used cord blood DNA as a negative control, as neonatal blood cells carry just 60–80 somatic mutations (roughly $10^{-8}$ mutations per bp). Both new versions of NanoSeq (sonication -MB-NanoSeq- and enzymatic -US-NanoSeq-) yielded mutation

loads and spectra consistent with previous knowledge[8] (Fig. 1a,b). By contrast, standard duplex sequencing (with end repair and without dideoxynucleotides), using sonication or enzymatic fragmentation, showed error rates around $1.5 \times 10^{-7}$ errors per bp and $4 \times 10^{-8}$ errors per bp, respectively. We then tested these protocols on samples with high levels of DNA damage (pancreas biopsies fixed in formalin for 3 days or 17 days). Standard duplex sequencing error rates increased roughly tenfold due to error transfer at damaged sites, whereas both versions of NanoSeq yielded comparable mutation loads to a control formalin-free biopsy (Extended Data Fig. 1c). This raises the possibility of using NanoSeq on more heavily damaged sources of DNA. The clean fragmentation protocols introduced here may also be useful upstream of other error-corrected sequencing methods beyond duplex sequencing, such as CODEC, SMM-seq or HiDEF-seq[18–20], to lower error rates while providing full-genome coverage.

## Mutation detection in polyclonal tissues

Combining these new protocols with bait capture[21], targeted NanoSeq can accurately quantify somatic mutation rates, signatures and driver landscapes in any tissue. Unlike traditional bulk sequencing, which

only detects mutations over a certain variant allele fraction (VAF) (typically more than 1–5%), single-molecule sequencing detects mutations present at any cell fraction, even in single cells, with a detection probability proportional to the mutation frequency in the cell population. In highly polyclonal samples in which the number of clones is larger than the duplex depth achieved, most mutations are seen in just one molecule, providing an efficient way to profile driver mutations in hundreds of clones simultaneously with a single sequencing library.

Our first somatic mutation studies in skin and oesophagus revealed a rich clonal landscape but were limited to a few individuals due to technical limitations[1,2]. To investigate how mutation landscapes vary across the population, we chose oral epithelium, a tissue with varied mutagenic exposures and amenable to large-scale non-invasive collection using buccal swabs. Here we describe its mutation landscape across 1,042 individuals, applying targeted NanoSeq to buccal swabs using a panel of 239 genes (0.9 Mb) (Methods, Extended Data Fig. 2 and Supplementary Table 1). Samples were sequenced to an average depth of 665 duplex coverage (dx), achieving 693,208 dx coverage across all samples. We also applied targeted NanoSeq to 371 blood samples from these donors (cumulative 250,947 dx).

## Targeted NanoSeq of blood

Analysis of the blood data demonstrates that targeted NanoSeq recapitulates the known mutation rates, signatures and drivers of a well-studied tissue. Mutation rates and trinucleotide spectra were consistent with previous whole-genome sequencing of haematopoietic stem cell colonies (Fig. 1c and Extended Data Fig. 3a,b). Using dNdScv to detect genes under positive selection[22] (Methods), we identified 14 genes (Fig. 1d, Extended Data Fig. 3e and Supplementary Table 3), all known clonal haematopoiesis drivers[23,24]. Hotspot dN/dS (the ratio of non-synonymous (N) to synonymous (S) substitutions) analyses also identified evidence of selection on several extra drivers, including *JAK2*, *MYD88*, *SF3B1*, *SRSF2*, *GNB1* and *STAT3* (Supplementary Table 4 and Supplementary Notes 2 and 4).

Despite the modest size of the dataset (371 samples, mean 676 dx), we found 4,406 non-synonymous mutations in these 14 driver genes (11.9 mutations per donor), including 1,904 mutations in *DNMT3A* and *TET2* (Fig. 1d). Of the mutations detected, 95% were called by just one molecule, 99% had unbiased VAFs under 1% and 90% had under 0.1% (Extended Data Fig. 3c and Methods). For comparison, a recent study of clonal haematopoiesis in more than 200,000 individuals using standard sequencing (only sensitive to clones with more than 1% VAF) found 0.029 and 0.012 *DNMT3A* and *TET2* mutations per donor[25], roughly a 100–200-fold lower yield of driver mutations per sample. Overall, these results confirm the power of targeted NanoSeq to measure mutation rates, spectra and selection in highly polyclonal samples.

## Driver landscape in oral epithelium

Self-collected buccal swabs were received by post from 1,042 volunteers from TwinsUK[26]. The cohort had a median age of 68 years (range 21–91), 79% women, 37% smokers and 332 pairs of twins (214 identical or monozygotic, 118 non-identical or dizygotic) (Extended Data Fig. 2a–g). A protocol designed to reduce saliva and blood contamination was used, with methylation and mutation analyses confirming a mean epithelial fraction of more than 90% (Extended Data Fig. 2h, Methods and Supplementary Note 4). Across donors, we found 341,682 somatic mutations, including 160,708 coding single-nucleotide variants (SNVs) and 29,333 coding indels (Extended Data Fig. 4g). We found that mutations in oral epithelium accumulate linearly with age, with rates roughly 18.0 SNVs per cell per year (95% confidence interval (CI) 16.7–19.4) and roughly 2.0 indels per cell per year (95% CI 1.7–2.4) (Fig. 2a,b). Because these rates are extrapolated from genic regions, which often have lower mutation rates, we applied RE-NanoSeq on 16

samples, revealing a genome-wide rate for oral epithelium of roughly 23 SNVs per cell per year (Extended Data Fig. 4a and Supplementary Note 3).

The data also revealed an unprecedentedly rich landscape of selection. We found 49 genes under positive selection by dNdScv, with over 90,000 non-synonymous mutations in them across clones, of which around 62,000 are estimated to be drivers (Fig. 2c–f, Extended Data Figs. 4h and 5a, Supplementary Table 3 and Supplementary Note 2). Comparison to matched blood suggests that selection in three of the genes (*DNMT3A*, *TET2* and *FOXP1*) results from low-level blood contamination in the buccal swabs (Extended Data Fig. 3f, Methods and Supplementary Note 4). Several other genes, including *PPM1D* and *ASXL1*, are genuine drivers in both tissues. Detailed information on the drivers uncovered is available in Supplementary Note 4 and Extended Data Figs. 5 and 6.

The commonest oral drivers match those in skin and oesophagus[1–3]. However, 31 of the oral drivers have not been previously reported in skin or oesophagus, including several drivers of head and neck squamous cell carcinomas (HNSC) (Supplementary Note 4). The density of driver mutations is considerably lower in oral epithelium than oesophagus (three to four times lower for *NOTCH1* or *TP53*)[2], and a few strong drivers in normal oesophagus seem neutral or weakly selected in oral epithelium (*KMT2D*, *NFE2L2* and *PIK3CA*). The absence of some important oral cancer drivers from the 46 genes under selection is also interesting. *CDKN2A*, *NFE2L2*, *PTEN*, *HLA-A*, *SMAD4*, *B2M* and *RB1* seem neutral or weakly selected in normal oral epithelium despite being common drivers in HNSC (Extended Data Fig. 5e), suggesting that selection on these genes may be a later event in HNSC development. This includes *HLA-A* and *B2M*, which may facilitate immune escape later in carcinogenesis.

Although duplex sequencing is normally applied to small gene panels[14,21], the use of a quantitative PCR step followed by a library bottleneck simplifies the use of panels of any size, including whole-exome panels. To ensure that major drivers are not being missed by our 239-gene panel, we performed exome-wide NanoSeq on 12 samples to a total duplex coverage of 1,024 dx. This reidentified *NOTCH1*, *TP53*, *PPM1D*, *RAC1* and *ZFP36L2*, suggesting that our panel includes the commonest drivers in oral epithelium. We also found a significant excess of indels in the keratin gene *KRT15*, which is probably the result of a hypermutation process known to affect highly expressed lineage-defining genes[27].

Our data reveal that the oral epithelium is composed of large numbers of small clones, with 10–20% of all buccal cells carrying driver mutations in older individuals (Extended Data Fig. 5d). We found 95.5% of oral mutations in only one duplex molecule, and around 90% had unbiased VAFs less than 0.1%. These VAFs are consistent with the sub-millimetric size of most clones reported in other epithelia[1,2,5], emphasizing the importance of single-molecule sensitivity to study solid tissues in bulk.

Aggregating duplex VAFs (Methods), we estimate that, in donors aged 65–85 years, the average fraction of cells carrying a driver mutation is approximately 10% for *NOTCH1*, 3% for *TP53*, 1% for *NOTCH2*, *CHEK2* and *ATM*, and less than 1% for other driver genes (Fig. 2e and Extended Data Fig. 5d). The frequency of *NOTCH1* and *TP53* mutations in oral epithelium contrasts with their frequencies in HNSC (The Cancer Genome Atlas, TCGA) of 16% and 69%, respectively (Fig. 2f and Methods). The similar frequency of *NOTCH1* driver mutations in oral cancer and normal oral epithelium suggests that *NOTCH1* mutations lead to benign clonal expansions at similar risk of transformation than *NOTCH1*-wild-type cells. By contrast, *TP53* and most other driver genes found under selection in oral epithelium seem enriched in squamous carcinomas consistent with a genuine tumorigenic role of these mutations. We note, however, that comparisons for most genes are limited by the number of cancers sequenced so far, compared with the thousands of normal clones assayed in this study.

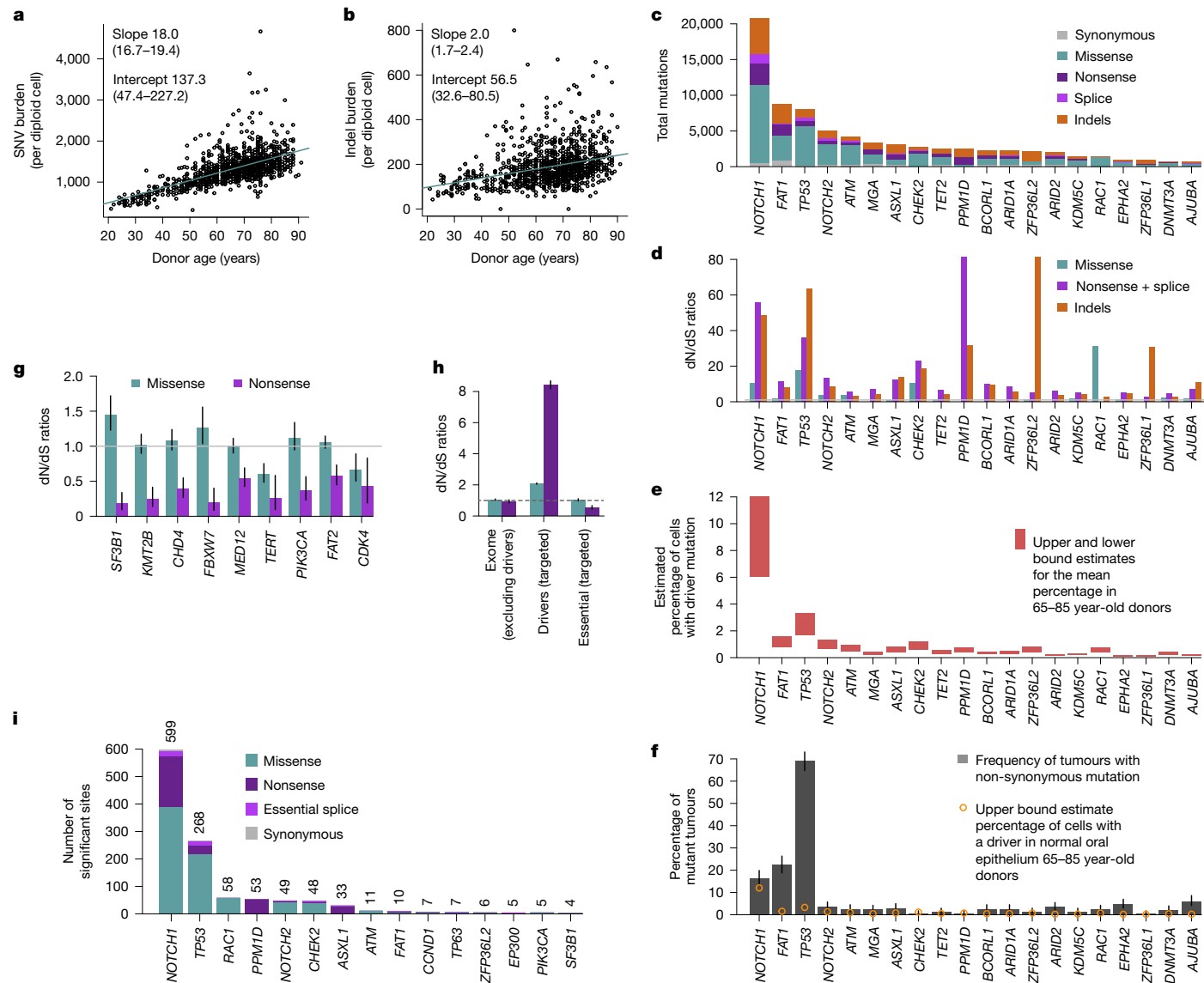

**Fig. 2 | Driver landscape of oral epithelium in 1,042 donors. a,b**, Linear regressions of the extrapolated genome-wide SNV (**a**) and indel burdens (**b**) in oral epithelium (estimated using targeted NanoSeq) against donor age. Points represent the point estimates of passenger mutation burdens for each sample. Slope and intercept of the fitted model (point estimates and 95% CIs) are indicated. **c**–**f**, For the top 20 significant driver genes based on driver mutation frequency, panels show mutation counts per mutation consequence category (**c**), dN/dS ratios per mutation consequence category (horizontal line indicates neutral dN/dS = 1, only categories with significant dN/dS ratios are shown for each gene) (**d**), estimated mutant cell percentages (upper and lower bounds for the mean across donors aged 65–85) (**e**) and percentage of tumours carrying a non-synonymous mutation (with error bars denoting 95% binomial CIs) (**f**). **g**, dN/dS ratios for missense and nonsense mutations in genes under significant negative selection. Error bars denote 95% CIs; horizontal line indicates neutral dN/dS = 1. **h**, Global dN/dS ratios for missense and nonsense mutations across non-driver genes (*n* = 18,767), targeted driver genes (*n* = 49) and 17 targeted essential genes. Error bars denote 95% CIs; horizontal line indicates neutral dN/dS = 1. **i**, Numbers of amino acid changes under significant positive selection based on site-level dN/dS (site-wide or under restricted hypothesis testing of known cancer hotspots), grouped by gene and mutation consequence category. Counts of significant amino acid changes per gene are shown above each bar.

## Negative selection on essential genes

The high number of mutations detected per gene also provides unprecedented power to detect negative selection, manifested as genes with a depletion of non-synonymous mutations (dN/dS < 1). Previous studies have shown that, exome-wide, most coding somatic mutations are tolerated and not negatively selected during somatic evolution, in contrast to long-term germline evolution[22,28]. However, strong negative selection in a small fraction of genes remains possible, particularly in essential haploinsufficient genes, but requires larger sequencing studies to be detectable[29].

Powered by the very high duplex depth and using new one-sided negative selection tests (Methods), we found nine genes under significant negative selection in our panel, mostly driven by selection against truncating SNVs (dN/dS < 1 for nonsense and essential splice site mutations) (Fig. 2g and Supplementary Table 3). This includes three essential genes from CRISPR screens (*SF3B1*, *CHD4* and *CDK4*) (Methods). *PIK3CA*, *SF3B1* and *TERT* showed negative selection against truncating mutations and positive selection on activating hotspot mutations (coding in *PIK3CA* and *SF3B1*, and promoter in *TERT*), suggesting that these genes are both essential genes in wild-type oral cells, and drivers upon acquiring activating mutations. Aggregating mutations from the 17 panel genes known to be essential in CRISPR screens revealed clear negative selection against truncating mutations in them (dN/dS = 0.69, 95% CI 0.61–0.78) (Fig. 2h). By contrast, dN/dS ratios for not-significantly selected genes in the panel as well as exome-wide dN/dS ratios, excluding selected genes,

were consistent with a largely neutral accumulation of coding somatic mutations (Fig. 2h and Extended Data Fig. 5f).

## In vivo saturation mutagenesis

The high number of mutations per gene provides an opportunity to start building high-resolution maps of selection across sites for the main driver genes[1,2,21,30] (Fig. 3a–c), a form of in vivo saturation mutagenesis. To formalize the analysis of recurrent mutation hotspots, we used site-level dN/dS models ('sitednds', Methods). Powered by the high number of mutations, we found 1,220 amino acid changes under significant positive selection ($q$-value less than 0.01), including 599 in *NOTCH1* and 268 in *TP53* (Fig. 2i and Methods). Restricting hypothesis testing to known cancer hotspots added several oncogenes to the list of positively selected genes in buccal swabs, including *PIK3CA*, *ERBB2*, *KRAS* and *HRAS* (Supplementary Table 4 and Methods).

The distribution of coding mutations in *TP53* mirrors that observed across thousands of cancers in the COSMIC database[31] (Fig. 3a and Methods). In *TP53*, we found nearly as many mutations as in 44,000 cancer exomes and genomes, and for several other driver genes, the number of mutations reported here far outweighs all previously observed mutations from cancer studies (Fig. 3b). For comparison, analysis of more than 7,500 cancer exomes from 29 cancer types from TCGA yielded around 15,000 driver mutations in known cancer genes[22], a quarter of the driver mutations observed in the current study.

Studying the distribution of mutations within genes revealed a diversity of selection patterns (Fig. 3a,c) (see Supplementary Note 4 and Extended Data Fig. 6 for detailed descriptions). *TP53* shows strong selection on missense mutations in the DNA binding domain and on truncating mutations across the gene. *NOTCH1* shows a characteristic clustering of missense mutations in EGF repeats 8–12, predicted to disrupt binding to NOTCH1 ligands Jagged and Delta[2]. Truncating mutations are subject to much weaker selection in the last exon of *NOTCH1* (dN/dS for nonsense mutations was 68.4 across the gene and 6.9 in the last exon), which probably reflects their inability to trigger nonsense-mediated decay. *RAC1* shows a classical oncogene pattern of strong selection on activating hotspots, with site dN/dS identifying 58 missense sites under significant selection ($q < 0.01$). Although these sites are scattered along the gene, they cluster around the GDP/GTP binding pocket in the three-dimensional structure of RAC1 (Fig. 3d). *PPM1D* encodes a known negative regulator of p53 and shows a characteristic pattern of recurrent nonsense SNVs and indels in the last exon, which results in the loss of a C-terminal degradation domain leading to a more stable isoform of the protein, hence increasing p53 suppression[32]. Finally, *TP63* shows an unusual selection pattern with a highly recurrent essential splice hotspot predicted to lead to an alternative isoform of p63 (ref. 33). Extra mutation maps are shown in Extended Data Fig. 6a–i.

Beyond coding mutations, we obtained high duplex coverage in exon-flanking sequences, and we targeted the promoters of many genes (Methods). To test for selection on specific subsets of coding and non-coding sites within a gene, we implemented a new function in dNdScv ('withingenednds', Supplementary Note 2). This identified several underappreciated driver sites, including strong positive selection on mutations causing stop codon loss in *TP53* and *PPM1D*, on intronic mutations near essential splice sites in *TP53*, *NOTCH1*, *CHEK2* and *NOTCH2*, and on some synonymous sites in *TP53* and *NOTCH1* predicted to affect splicing by SpliceAI[34] (Fig. 3e and Supplementary Note 4). In addition, we observed suggestive clustering of mutations at the *TP53* transcription start site, the *TP53* polyadenylation signal, and at splice sites in the first non-coding exon of *TP53* (Fig. 3f), as well as hotspots in non-canonical but previously reported 5′ untranslated region sites in *TERT*. These analyses also revealed a general inflation of mutations in the core promoters of many genes, suggestive of a higher background mutation rate in promoters rather than selection (Extended Data Fig. 6m), consistent with previous reports[35]. Despite our panel not being designed to search for non-coding *cis*-regulatory driver mutations, these examples show the potential of deep somatic mutation scanning to exhaustively discover coding and non-coding driver sites.

Variants of uncertain significance are germline or somatic variants identified by genetic testing whose clinical relevance is unknown. Evidence of selection in cancer is starting to be used for the classification of variants in some genes[36] but is limited by the sparsity of cancer genomic datasets. To investigate whether selection in normal tissues could contribute to these efforts, we compared the distribution of site-level dN/dS ratios for sites annotated in ClinVar as pathogenic, benign or of uncertain significance. Nearly all known pathogenic sites in *TP53*, *NOTCH1* and *PPM1D* had high dN/dS ratios, and nearly all known benign sites had low dN/dS ratios (Extended Data Fig. 6n). Looking at sites reaching or approaching significance ($q < 0.20$), we find many variants of uncertain significance (and zero benign variants) with comparable evidence of selection to known pathogenic variants (including 86 in *TP53*, 35 in *NOTCH1* and 5 in *PPM1D*) (Supplementary Table 4). Although deeper sequencing will be required to achieve true saturation (Supplementary Note 4), these results show that ultra-deep single-molecule sequencing of polyclonal tissues has the potential to provide in vivo saturation mutagenesis information for genes under somatic selection.

## Mutational epidemiology

The discovery of many clones carrying cancer-driver mutations in normal tissues has caused some confusion about their role in carcinogenesis. However, these clones are entirely compatible with a multistage model of carcinogenesis, and were in fact anticipated by some classical mathematical models (see Supplementary Note 5 for an extended description). In the 1950s, Armitage and Doll[37] proposed that the rapid increase in cancer incidence with age could be explained by a model in which cells acquire mutations linearly with age and 6–7 driver events are required for transformation. Lesser-known models with clonal expansions were proposed soon after and showed that the size and type of clonal expansion had large effects on cancer incidence. The current model of carcinogenesis is that cancers emerge by somatic evolution. Both mutation and selection (clonal expansion) increase the likelihood of a cell acquiring the complement of driver changes needed for transformation. Carcinogens may thus act by inducing mutations (mutagens) or by altering selection (promoters[38,39] or selectogens[40]) (see Supplementary Note 5 for an extended explanation). By studying the variation in mutation and selection across 1,042 individuals, we can begin to quantify these processes.

To investigate the mode of clonal growth in oral epithelium, we first studied how the frequency of driver mutations increases with age in our cohort. This showed that the estimated fraction of cells carrying driver mutations increases roughly linearly with age, through the accumulation of many small clones, with the VAF of the largest clone per individual growing slowly or plateauing with age (Fig. 4a). As new driver mutations occur continuously, this observation is inconsistent with models of continued clonal growth, including exponential growth, quadratic growth (expected if clones grow only at their edges)[41] and models predicting an acceleration of selection during ageing[42]. Instead, the pattern seems more consistent with a plateauing model of clonal expansion, in which clone sizes are constrained (by cell-intrinsic or cell-extrinsic mechanisms) (Supplementary Notes 5 and 6 and Extended Data Fig. 7). This contrasts with the pattern observed in blood in which both the driver density summed across clones and the size of the largest clone increase almost exponentially with age, consistent with previous clonal haematopoiesis studies[43]. Models suggest that the slower-than-expected increase in driver density with age and the small size of epithelial clones must be major barriers to carcinogenesis (Supplementary Note 5).

Targeted NanoSeq also provides information on mutation rates and signatures across individuals. Performing signature decomposition

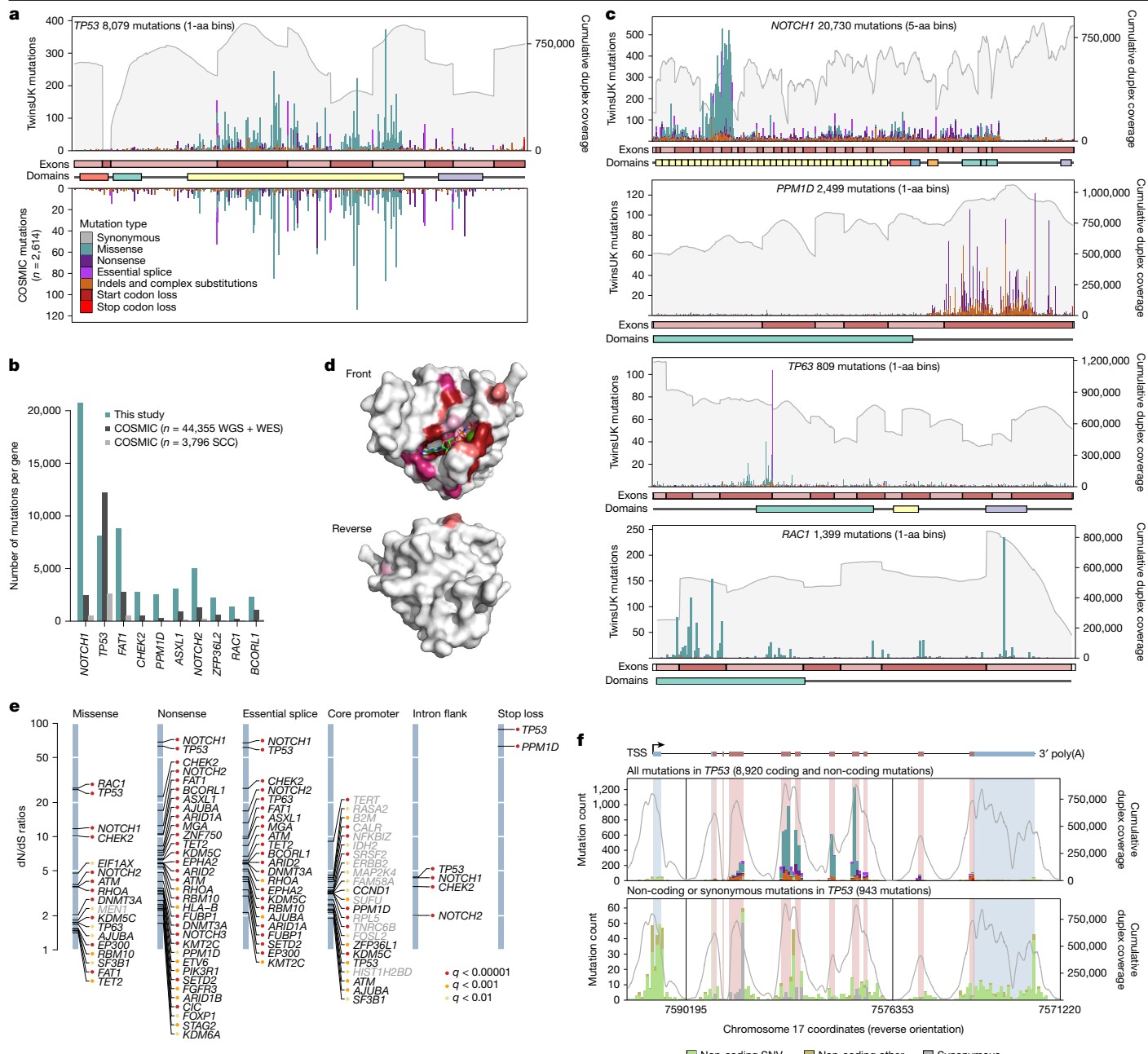

**Fig. 3 | In vivo saturation mutagenesis in oral epithelium. a**, Mutation bar plot for *TP53*. The *x* axis represents coordinates along the coding sequence. Exons and protein domains are indicated along the *x* axis. The *y* axis represents number of mutations, either in the 1,042 TwinsUK oral epithelium samples used in this study (top) or in squamous cell carcinoma from the COSMIC database (bottom). Mutations are coloured according to mutation consequence category. Grey shading indicates cumulative duplex coverage across TwinsUK buccal swab samples. **b**, Numbers of mutations per gene found in this study and in the COSMIC catalogue (obtained from across all whole-genome sequencing (WGS) and whole-exome sequencing (WES) studies or only squamous cell carcinoma (SCC) WGS and WES studies), for a selection of driver genes. **c**, Mutation bar plots for *NOTCH1*, *PPM1D*, *TP63* and *RAC1*. Elements are as indicated in **a**; COSMIC mutations not shown. **d**, Diagrams of the three-dimensional structure of RAC1, showing the clustering of sites under significant positive selection around the GDP/GTP binding pocket. Residues with site-level dN/dS *q* < 0.01 are coloured. Shading intensity denotes degree of significance. **e**, dN/dS ratios for driver sites under significant positive selection based on the withingenednds method. Driver sites are classified into six groups according to mutation consequence. Labels in grey indicate genes not identified as significant by gene-level dN/dS analyses. **f**, Mutation bar plot for *TP53*, including all mutations (top) and synonymous or non-coding mutations only (bottom). The *x* axis represents genomic coordinates along the gene body, with coding exons (red) and untranslated regions (UTRs) (blue) indicated by the gene diagram on top and the shading within each histogram. The grey line denotes cumulative duplex coverage across TwinsUK buccal swab samples. Coding mutation counts are coloured according to mutation consequence as indicated in **a**. TSS, transcription start site.

on all 1,042 donors, we found two dominant mutational signatures (Fig. 4b). Signature A resembles a combination of COSMIC single-base substitution (SBS) signatures SBS5 and SBS1 (94% and 6%, respectively, cosine similarity 0.90, Methods). SBS5 is a ubiquitous clock-like signature observed across tissues, believed to result from the occasional misrepair of the continuous DNA damage suffered by all cells[8,9,44], whereas SBS1 results from the deamination of 5-methylcytosine. Signature A is largely responsible for the life-long accumulation of mutations

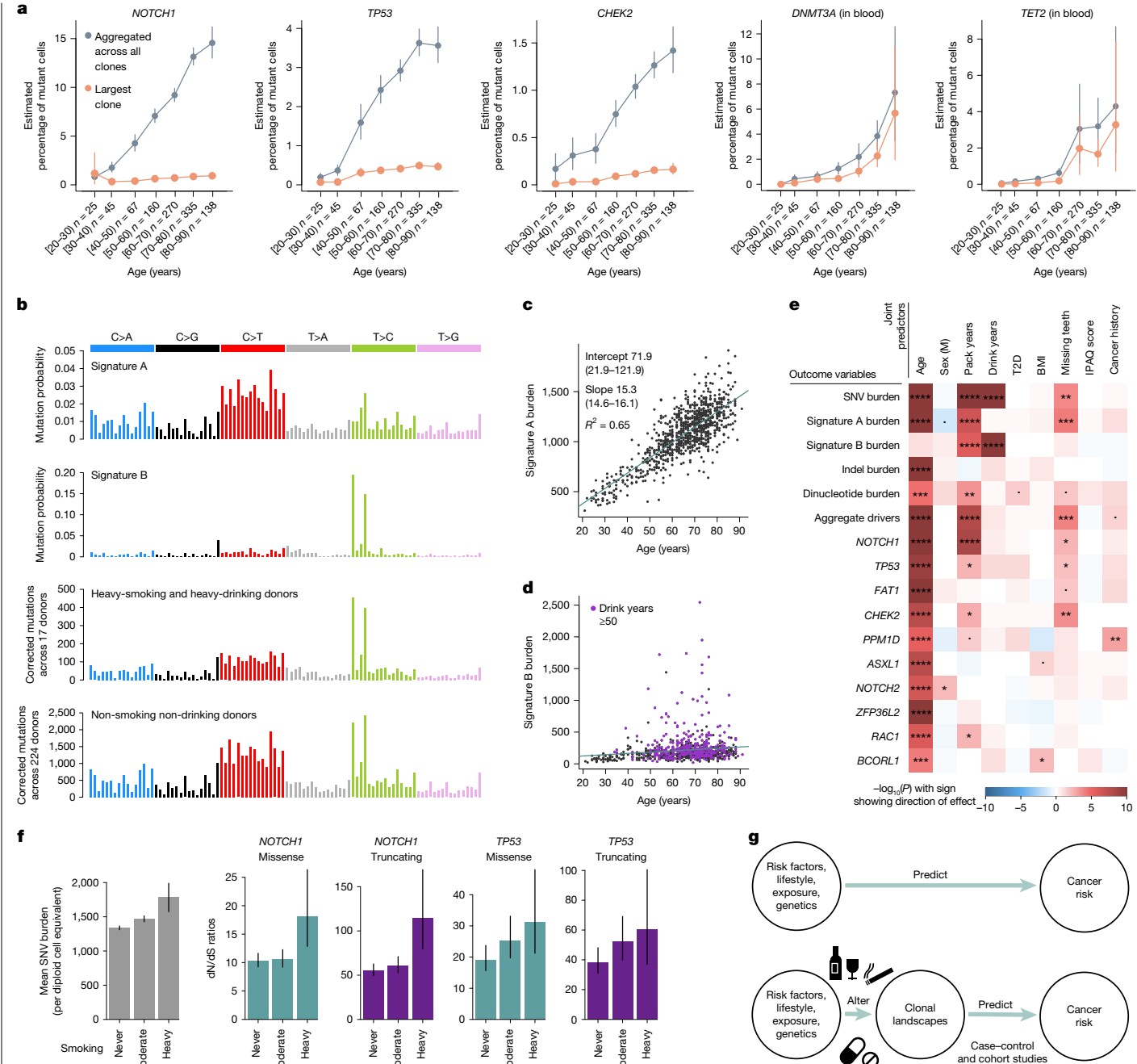

**Fig. 4 | Mutational epidemiology in oral epithelium. a**, Mutant cell percentages for the largest clone (orange) and for all mutant clones (grey) for *NOTCH1*, *TP53* and *CHEK2* in oral epithelium, and *DNMT3A* and *TET2* in blood, as a function of age. Error bars denote 95% CIs. **b**, Trinucleotide mutational spectra for (top to bottom) inferred signatures A and B, and mutations in oral epithelium from heavy-smoking heavy-drinking donors (*n* = 17) and non-smoking non-drinking donors (*n* = 224). Mutational spectra are corrected by the ratio of genomic to observed trinucleotide frequencies. **c,d**, Linear regressions of genome-wide signature A (**c**) and signature B (**d**) burdens in oral epithelium against donor age. **e**, Heatmap of associations between different measures of mutation burden, signature burden or driver density (*y* axis) and relevant donor metadata (*x* axis), inferred using linear mixed-effects regression models. The likelihood-ratio test

*P* value of each association is indicated by both colour shading (red and blue for positive and negative associations, respectively) and asterisk labels (****$q < 10^{-4}$; ***$q < 10^{-3}$; **$q < 0.01$; *$q < 0.05$; dot, $P < 0.05$; *q*-values are calculated using the Benjamini and Hochberg false discovery rate method). BMI; body mass index; IPAQ, International Physical Activity Questionnaire; T2D, type 2 diabetes. **f**, Change in SNV burden and in dN/dS ratios for missense and truncating mutations in *NOTCH1* and *TP53*, as a function of smoking status (never, 0 pack-years, *n* = 632; moderate, 0–20 pack-years, *n* = 283; heavy, more than 20 pack-years, *n* = 84). Error bars denote 95% CIs. **g**, Non-mechanistic (top) and mechanistic (bottom) risk models connecting predictor variables to cancer risk. Mechanistic risk models can offer insight into the impact of risk factors on mutational or clonal landscapes and may be used to predict cancer risk.

in oral epithelium, with a slope of roughly 15.3 mutations per cell per year (Fig. 4c, $R^2 = 0.65$, $P < 2.2 \times 10^{-16}$). Signature B resembles COSMIC SBS16 (cosine similarity 0.97), a common signature in oesophagus and liver, associated with alcohol consumption and aldehyde metabolism[3,45,46] (Extended Data Fig. 8). Signature B showed extreme variation

across donors, contributing low numbers to most individuals but more than 1,000 mutations per cell in some heavy drinkers (Fig. 4d and Supplementary Note 3).

The paucity of smoking-associated signatures (SBS4 and SBS92) and APOBEC mutagenesis (SBS2 and SBS13) in oral epithelium is remarkable

given their frequency in HNSC tumours, but seems compatible with a recent study of oral cancer evolution[47]. Further analyses supported their paucity, including alternative deconvolution methods, the absence of smoking-associated indel and double-base substitution (DBS) signatures (Extended Data Fig. 8d,e), and non-significant likelihood-ratio tests comparing models with and without these signatures (Supplementary Note 3). The absence of the classical smoking signature SBS4 may be explained by the low expression of *CYP1A1* in oral epithelium[47], the main metabolizer of benzo(a)pyrene and perhaps by a lower exposure to tobacco mutagens of the basal stem cells in the oral squamous epithelium compared with respiratory epithelium.

Oral cancer risk factors could act by increasing mutation rates or inducing clonal expansions. To test for such mutagenic and selectogenic effects while accounting for confounders, we used multivariate mixed-effect regressions using risk factors and other metadata as covariates, and different measures of mutation rates, signatures or driver densities as outcome variables (Fig. 4e, Supplementary Note 7 and Extended Data Fig. 9). As expected, SNVs, indels, dinucleotides, signature A (but not signature B) burden and the density of all major drivers increased strongly with age. Sex was not significantly associated with differences in any of these outcome variables when correcting for confounders, despite our power to detect differences greater than 5% (Extended Data Fig. 9). This suggests that the higher incidence of HNSC in men may be mostly explained by lifestyle factors, as predicted by some epidemiological studies[48].

Tobacco smoking is a major oral cancer risk factor, and we found pack-years to be strongly associated with total SNVs, signature A and signature B, dinucleotide substitutions (but not indels), driver density across genes, *NOTCH1* driver density and nominally significantly associated with three other drivers. Alcohol consumption is another major oral cancer risk factor, and we found estimated drink-years to be strongly associated with SNV and signature B burden, but not signature A burden, consistent with the known aetiology of signature B/SBS16 (Methods). Poor oral health is also an oral cancer risk factor[49], and we found that the number of missing teeth correlated with signature A burden and overall driver density. We were unable to study the effect of oral human papillomavirus (HPV) infection in our dataset, an increasingly important risk factor for oral cancer particularly in younger individuals[50], as HPV history was unavailable and sequencing-based detection of HPV yielded limited information (Methods).

The association of alcohol consumption with signature B is believed to result from DNA damage by alcohol-derived aldehydes[45]. However, the mechanistic basis for the association of smoking with signature B is less clear. Analyses in our dataset suggest that smoking increases signature B by exacerbating the mutagenic effects of alcohol consumption, consistent with epidemiological studies[51] (Supplementary Note 7). However, we cannot rule out the possibility that the association is partially caused by inaccurate self-reporting of alcohol consumption. Whereas our models also suggest that smoking and poor oral health are significantly associated with an increase in signature A/SBS5, the mechanistic bases for these associations remain unclear. Notably, these regressions suggest that 1 additional year of life causes as many mutations in the oral epithelium as roughly 2.8 pack-years or 19.1 drink-years (95% CI 2.03–3.63 and 13.9–24.3, respectively, see Supplementary Note 7 for caveats and interpretation).

More regression models can further disentangle the mutagenic or selectogenic effects of some risk factors. If a carcinogen acts solely as a mutagen without altering selection, driver density should increase proportionally to the increase in mutation burden, at least under some assumptions. Pure promoters or selectogens should alter clonal selection without changes in mutation rates, whereas dual carcinogens may alter both (Supplementary Note 7). Thus, to test for selectogenic effects, we used different regression models correcting driver density for changes in mutation rates (Supplementary Note 7). Putative selectogenic associations included an increase in *NOTCH1* clones with smoking,

*CHEK2* with poor oral health and trends for other genes (Fig. 4e and Extended Data Fig. 9d). These associations are suggestive of promoter or selectogenic effects, but they are only correlative and further studies are needed to confirm them. We also note that we have lower statistical power to detect changes in selection per gene than changes in mutation rates (Extended Data Fig. 9e), and that our detection of selectogenesis is limited to effects on clones observed in normal oral epithelium (for example, studying selectogenesis on more advanced precancerous lesions will require other cohorts). Nevertheless, these analyses show the potential of mutational epidemiology studies to illuminate the mechanisms of action of major cancer risk factors.

Altogether, these results indicate that tobacco may contribute to early oral carcinogenesis not through classical SBS4 or SBS92 mutagenesis, but through an acceleration of SBS5 and SBS16 and changes in clonal selection. Alcohol consumption seemed to cause fewer driver mutations than expected from its increase in mutation rates. Analysis of the distribution of signature B mutations revealed that this is due to a low driver-generation potential of signature B/SBS16, as ATA>ACA or ATT>ACT signature B mutations are heavily biased towards intronic sequences (Extended Data Fig. 10a,b).

Finally, this dataset offers an opportunity to start investigating germline influences on somatic mutation rates. First, we leveraged our twin design to test for heritability. We compared the difference in mutation rates and driver frequencies between identical (monozygotic), non-identical (dizygotic) and unrelated same-age pairs of donors, while accounting for confounders (Supplementary Note 8). This provided some evidence of heritability for signature A (monozygotic versus dizygotic $P = 0.004$), *NOTCH1* ($P = 0.023$) and *TP53* ($P = 0.018$) (Extended Data Fig. 10c). Similar signals were found with more formal ACE (A, additive genetic effects; C, common/shared environment effects; E, unshared environment effects) and genomic-relatedness tests (Supplementary Table 5 and Supplementary Note 8). Second, we took advantage of having genome-wide single-nucleotide polymorphism (SNP) genotyping and complete metadata from 590 donors to evaluate the effect on the mutation landscape of 52 SNPs associated with oral cancer or clonal haematopoiesis risk. This revealed a significant association between a SNP near *ALDH2* (rs4767364), a key enzyme in alcohol metabolism and the rate of signature B/SBS16 ($P = 9.4 \times 10^{-5}$, $q = 0.02$) (Supplementary Table 6 and Supplementary Note 8). This suggests that the known association between rs4767364 and HNSC risk[52] is driven by a higher mutagenicity of alcohol in these donors, consistent with the known effects of a different *ALDH2* SNP (rs671) common in East Asian individuals. Finally, although the statistical power was low, we performed genome-wide association studies (GWAS) of mutation rates and driver densities for completeness, not finding convincing genome-wide significant associations (Supplementary Note 8). Altogether, these analyses suggest that germline factors can influence somatic mutation rates and clonal selection, although larger cohorts are needed to comprehensively identify these associations. We note that discovering germline mutations influencing somatic mutation rates could illuminate the mechanistic bases of SBS5 and enable Mendelian randomization for causal inference on the role of somatic mutations across common diseases.

## Discussion

Building on duplex sequencing, we have developed a new version of NanoSeq that achieves accurate somatic mutation detection on single DNA molecules (with fewer than five errors per gigabase) while being compatible with whole-genome, whole-exome and deep targeted sequencing. This method greatly simplifies the study of somatic mutation rates, signatures and driver landscapes in any tissue, regardless of clonality.

Applying targeted NanoSeq to oral epithelium, we have unveiled an unprecedentedly rich landscape of selection in a normal solid tissue, with 46 genes under positive selection, more than 62,000 driver

mutations and several genes under negative selection. These data also exemplify how deep single-molecule sequencing of highly polyclonal tissues can yield high-resolution maps of selection within genes. This could complement in vitro saturation mutagenesis efforts to help variant annotation for genetic diagnosis. Whereas this approach is limited to genes under selection in a tissue, a wider range of disease-relevant genes can be assayed across tissues[28].

The ability to study somatic landscapes in large sample cohorts offers several opportunities to augment traditional cancer epidemiology. First, systematic studies of the mutation landscape across individuals, case–control studies[16,17] and intervention studies could help build mechanistic models connecting risk factors to mutation and clonal landscapes, and these landscapes to cancer risk (Fig. 4g). Such studies could provide insights into the mode of action of poorly understood risk factors (for example, obesity), as well as enable risk prediction or stratification. Second, studies of the mutation landscapes of normal tissues in populations with unusually high rates of certain cancers could shed light on unknown exposures, potentially helping develop prevention strategies. Third, mutation and clonal landscapes may be informative as surrogate risk markers in cancer prevention and molecular prevention trials. Although molecular prevention of cancer in the general population is rarely discussed, the discovery of simple markers of cardiovascular disease risk, such as low-density lipoprotein cholesterol and hypertension, enabled the development of statins and antihypertensive medications, which have transformed the management of cardiovascular disease[53].

Beyond cancer, somatic mutations have long been speculated to contribute to ageing and other diseases. Suggestive associations have now been found between somatic mutations in certain genes and many diseases[54,55]. However, systematic studies in polyclonal conditions have not been possible with available technologies. Accurate whole-exome single-molecule sequencing has the potential to enable sensitive and unbiased discovery of somatic driver mutations in any tissue and across diseases.

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

## Methods

### Cohort selection

The TwinsUK study contains around 16,000 participants. From a preselection of 4,800 donors, we invited 1,796 to participate based on several criteria, receiving buccal swabs from 1,236 donors (Extended Data Fig. 2a). The use of these samples was approved initially by the North West Research Ethics Committee (REC 19/NW/0187 and REC 24/NW/0106), and informed consent was provided by participants. To increase our statistical power to study associations with exposures, risk factors and germline factors, we included all available donors of age 80 or higher ($n = 230$), as many complete twin pairs as possible, smokers, individuals with obesity (BMI > 30), and individuals with available genome-wide genotyping information. We also favoured the selection of men and people of colour to reduce some of the demographic biases in the TwinsUK registry compared with the general population. To test for associations between the mutational landscape and medications or clinical histories, we favoured the inclusion of individuals with a history of cancer (including all donors with a history of oral cancer, $n = 12$) or a self-reported treatment history including tamoxifen, immunosuppressants, metformin, aspirin or ibuprofen. 194 samples were excluded from analysis based on several sequencing quality metrics, leaving a total of 1,042 samples in the study. Exclusion criteria included: removal of contaminated samples with either human ($n = 17$) or non-human ($n = 132$) DNA, exclusion of samples with mean duplex coverage lower than 50 dx ($n = 79$) and exclusion of swabs with genotyping information not matching the pre-existing genotyping information from TwinsUK ($n = 7$) (Extended Data Fig. 2a).

From the final 1,042 donors in the study, we also selected 380 individuals with archival whole-blood DNA available for sequencing in the TwinsUK BioBank. In total, 371 samples passed quality controls for study inclusion (Extended Data Fig. 2a). The selection of blood donors was based on several criteria: 12 donors (and their twins) treated with metformin, 30 donors (and their twins) with the highest mutation burden per year in the buccal swabs, 25 donors (and their twins) with the highest driver fractions, 25 donors (and their twins) with the lowest driver fractions, 25 donors with high driver fractions in the buccals for known clonal haematopoiesis drivers (*TET2*, *SF3B1*, *DNMT3A*) and 5 donors with high driver fractions in the buccal swabs for each of the following drivers: *PPM1D*, *ASXL1* and *NOTCH3*. The remaining twin pairs were sampled randomly.

### Metadata

Metadata were provided by TwinsUK, obtained through periodical questionnaires that were collected longitudinally for most donors. For each participant, TwinsUK provided age, sex, height, weight, BMI, twin zygosity and ethnicity. A few self-reported zygosities were corrected based on genotyping information. Self-reported medication histories were also obtained from questionnaires, however, these are expected to be incomplete. Further information on history of herpes labialis and a short list of prespecified treatments was provided by TwinsUK from anonymized medical records: metformin, tamoxifen, rapamycin, aspirin, non-steroidal anti-inflammatories and immunosuppressants. Cancer history was provided and coded as: 0 (no cancer), 1 (non-melanomatous skin cancer), 2 (other cancer) and 3 (oral cancer).

For major oral cancer risk factors and other relevant variables, we processed available questionnaires further to obtain summary metrics, including: tobacco smoking, alcohol consumption, physical activity, weight, height, BMI, oral hygiene, gastro-oesophageal reflux, diabetes, history of cancer and medication histories.

**Smoking and alcohol consumption.** Self-reported smoking and alcohol consumption was collated from 14 periodical questionnaires. We focused on the most recent questionnaires due to the relevance of the questions asked in them and the coverage of answers across individuals. For smoking, we kept the maximum value of reported pack-years per donor across questionnaires. As standard, 1 pack-year was defined as 365 packs of cigarettes (7,300 cigarettes). For alcohol intake, self-reported current weekly consumption was available for most donors, but self-reported information on lifetime consumption was only available for a few donors. An estimate of drink-years was calculated by multiplying the average current weekly alcohol consumption, across several questionnaires if available, by the duration of adult life (age minus 18). We note that this estimate is an extrapolation and should be used with caution, but regression models suggest that this estimate was more explanatory than self-reported lifetime consumption (see Supplementary Note 7 for analyses on alternative metrics).

**Oral health.** Self-reported information on gingivitis, periodontitis and gum bleeding was only available for a few donors. By contrast, the number of natural teeth remaining was available for most donors, recorded as an ordinal variable. For ease of interpretation in the regression models, we inverted this variable to reflect the number of missing teeth, as follows: 0, 20 or more natural teeth; 1, 10–19 natural teeth; 2, 1–9 natural teeth and 3, no natural teeth. Where several answers were available from questionnaires on different years, the lowest number of natural teeth left was used.

**BMI, weight and height.** Weight and height were provided by TwinsUK for most donors. Both metrics were averaged across questionnaires for each donor. BMI was calculated using the standard formula: $weight/(height^2)$.

### Buccal swab processing and sequencing

Puritan buccal swab kits with instructions for self-collection were posted to the homes of voluntary donors by TwinsUK (CamBio, CA-1723-H100). Kits contained a primary and secondary plastic container, an outer rigid container (Alpha Laboratories, RF95-LL1) and a prepaid return envelope. Participants mailed their buccal swabs directly to the Wellcome Sanger Institute. Swabs were refrigerated at 4 °C on arrival.

To extract DNA, buccal cells were dissociated into 1 ml of PBS solution in an Eppendorf tube through manual agitation for 1 min. The swab tip was then cut with scissors and left in the tube for 30 min before removal. The solution was then centrifuged at 1,000$g$ for 1 min. The supernatant was removed leaving a cell pellet with minimal residual PBS (less than 100 µl). The QIAamp DNA Micro Kit (QIAGEN, 56304) was used for cell lysis and DNA extraction. First, 180 µl of buffer ATL and 20 µl of proteinase K were added to the resuspended cell pellet, followed by overnight incubation on a thermomixer at 56 °C and 800 rpm. DNA extraction followed the manufacturer's protocol with several modifications: centrifugation steps were all performed at 20,000$g$, DNA was eluted in 50 µl if buffer EB (10 mM Tris-Cl, pH 8.5) (QIAGEN, 19086), incubation with the first elution step was for 5 min, and the eluent was passed through the spin column for a repeat elution into a DNA LoBind 1.5 ml tube (Eppendorf, 0030108051). The extracted DNA was quantified using a Qubit High Sensitivity and then stored at −20 °C before 40 µl of the thawed sample being diluted to a final volume of 120 µl with buffer EB (QIAGEN, 19086) and submitted for NanoSeq library preparation on an Abgene AB0800G plate (Thermo Fisher Scientific, AB0800G).

A detailed description of the targeted NanoSeq and standard duplex sequencing library preparation protocols is provided in Supplementary Note 1.

### Mutation calling

Sequencing data were mapped to the human genome (GRCh37, hs37d5 build) with BWA-mem[57] as described before[8]. Bases were called when there was duplex consensus with at least two reads per original strand, requiring a minimum consensus base quality score of 60, a VAF lower

than 0.1 in the matched normal, a minimum AS-XS of 10 (below), no more than an average of 3 mismatches per read (or 4 if a variant is called), a minimum coverage of 25× in the matched normal and trimming 8 bp from each read end. We note that by counting all mutant bases and all reference bases in each duplex molecule, NanoSeq implicitly considers the VAF of each mutation to calculate mutation burdens. This makes NanoSeq robust to differences in clonal composition across samples. Compared with ref. 8, instead of sequencing independent matched normals to filter out germline variation, we took advantage of the high coverage and polyclonality of the buccal swab samples to remove germline SNPs by filtering out variants with VAF ≥ 10%. We note that this is adequate as long as the samples are highly polyclonal. Relaxing this cut-off to VAF ≥ 30% did not seem to recover genuine mutations in the buccal swabs but led to an increase in mapping artefacts. Because all blood samples had matching buccal swab data, somatic mutations in blood were called using their buccal swabs as matched normals, excluding as probably germline any variants with VAF ≥ 10% in the buccal swabs.

A significant modification in the targeted NanoSeq calling pipeline compared with our published RE-NanoSeq pipeline is the relaxation of the AS-XS threshold from 50 to 10. AS-XS measures the difference in mapping quality between the primary and secondary alignments, excluding regions with ambiguous mapping from analysis. For mutation burden and signature analyses with whole-genome NanoSeq, we previously recommended a strict AS-XS cut-off to minimize the impact of mapping artefacts[8]. However, for driver discovery it is important to preserve regions with less unique mapping qualities. Using a list of 1,152 oncogenic hotspots from TCGA and MSKCC provided by the dNdScv package[22], we noticed that the original AS-XS cut-off would have filtered out a significant number of them. Reducing the AS-XS cut-off from 50 to 10 ensured the retention of duplex coverage on nearly all canonical cancer hotspots while still ensuring accurate mutation rates and signatures in control cord blood samples (Extended Data Fig. 2l–o).

Two extra filters are important to avoid recurrent mapping artefacts and to minimize the effect of inter-individual contamination. First, a 'SNP+noise' mask containing common germline SNP sites and recurrent mapping artefacts was generated for targeted NanoSeq as described before[8]. Second, we noticed that mapping errors not captured by this mask can manifest as recurrent artefacts where the mutant base is often seen at specific positions within a read. This can be caused, for example, by mismapping of reads from polymorphic segmental duplications. A Kolmogorov–Smirnov test on the position of the mutant bases within reads was applied to remove recurrent artefacts after mutation calling. Indels were also filtered out if their overlap with the 'SNP+noise' mask was 50% or greater, if they occurred at sites without a base called, if they had a VAF > 0.1 or if they were seen in more than 50 samples. This only removed a small number of artefactual indel sites, which also had a strong read positional bias.

## Duplex VAFs and unbiased VAFs

The VAF represents the proportion of reads at a specific site carrying a variant, relative to the total reads at that site. When working with standard duplex sequencing or targeted NanoSeq data, only a fraction of read bundles reach the '2 + 2' requirement for duplex calling (that is, read families with at least two reads from both strands). We can then calculate three separate VAFs: (1) the 'duplex VAF', defined as the fraction of callable (2 + 2) read bundles supporting a given mutation, (2) the 'BAM VAF', calculated using the deduplicated BAM file containing one representative read per read bundle (and including calling and non-calling read bundles) and (3) the 'unbiased BAM VAF', calculated using the deduplicated BAM file but excluding calling read bundles.

These VAFs can be used for different purposes. (1) Estimation of the fraction of cells in a sample carrying a specific mutation. If a mutation was discovered in a sample using duplex (2 + 2) reads, duplex VAFs or BAM VAFs tend to overestimate the fraction of cells carrying the

mutation in the sample due to the discovery bias resulting from the inclusion of reads used for mutation calling. For this purpose, 'unbiased BAM VAFs' provide an unbiased estimate of the VAF of a mutation in the sample as they are calculated from reads not used for duplex calling. (2) Estimation of the fraction of cells carrying somatic mutations in a given gene. The molecules that reach duplex calling (2 + 2) in a targeted NanoSeq experiment represent a random sample of all copies of a gene in a population of cells. The duplex VAF for a given site represents the fraction of mutant molecules at the site. If we assume that all (or nearly all) cells are diploid and that cells carry at most one driver mutation per gene (heterozygous), then we can estimate the fraction of cells with mutations in a given gene by summing the duplex VAF ($v_d$) of mutations across all sites in the gene ($F = 2\Sigma v_d$). If we assume that cells may carry up to two mutant copies of the gene per cell or if we are looking at a haploid region of the genome (for example, the X chromosome in male individuals), we can estimate the fraction of mutant cells in the sample using the sum of duplex VAFs across all sites in the gene ($F = \Sigma v_d$). Some genes, such as *NOTCH1* in squamous epithelia can show biallelic loss by one mutation in each allele (SNVs or indels) or by one mutation and a copy number change (either a deletion or a copy-neutral loss of heterozygosity). We have previously shown that for these conditions, as well as for populations with mixtures of heterozygous and homozygous mutant cells, the fraction of mutant cells in the population falls within the range [$\Sigma v_d$, $2\Sigma v_d$] (ref. 2). Unless described otherwise, other references to the fraction of mutant cells for a given gene assume a maximum of one driver mutation per cell and a largely diploid population.

As not all non-synonymous mutations in a driver gene are driver mutations[22], to estimate the fractions of cells with driver mutations (Figs. 1d and 2e), we multiplied the estimated fraction of cells with non-synonymous mutations by the estimated fraction of mutations that are drivers for each class. We estimated the fraction of mutations that are drivers using ($\omega - 1$)/$\omega$, for mutation classes with $\omega \geq 1$ (where $\omega$ is the dN/dS ratio per mutation type per gene). To account for potential differences in clone sizes for driver mutations, we used dN/dS ratios calculated without collapsing mutations reported by many molecules into single entries to dNdScv (Supplementary Code).

## Epithelial purity and targeted methylation

To quantify the epithelial fraction of a representative set of buccal swabs, we used two approaches: (1) targeted enzymatic methylation sequencing on 187 buccal swabs, and (2) comparing the VAFs of clonal haematopoiesis mutations in the buccal swabs of donors with blood and buccal swab data.

From 187 swabs, we generated low-input enzymatic methylation libraries and then undertook targeted capture with a panel of informative CpG sites, using the NEBNext Enzymatic Methyl-seq Kit (NEB, E7120L). We used a custom Twist Bioscience hybridization panel targeting 1,162 CpGs selected from the centEpiFibFatIC.m, centDHSbloodDMC.m and centEpiFibIC.m matrices in the EpiDISH R package[58], to deconvolute epithelial, fibroblast, fat and blood cell types. We also targeted 353 CpG from the original Horvath clock[59] and 50 CpGs in the promoters of 25 driver genes. The design is available in Supplementary Table 2.

For each sample, DNA was quantified and normalized to roughly 1 ng μl⁻¹. Normalized DNA samples were then sheared with the NEBNext UltraShear fragmentation mix (NEB, M7634L), end-repaired, A-tailed, adapter-ligated with a methylated TruSeq-compatible adapter stub (all using NEB Ultra II reagents) and, after a SPRI (solid-phase reversible immobilization) clean-up, the resulting libraries were oxidized using TET2 (converting methylcytosines to carboxylcytosines) and deaminated using APOBEC (converting bare cytosines to uracils but retaining the carboxylcytosines, thus preserving the locations of methylation marks). The deaminated libraries were amplified, and sequencing indexes (and the rest of the adapter sequence) were introduced

using NEB Q5U and the Sanger Institute's UDI primers. After a further SPRI clean-up, libraries were requantified and mixed in an equimolar pool with a cumulative DNA mass of 1–4 μg. Twist Bioscience probes targeting the sequences of interest were then added. After evaporating all the liquid, the probes hybridized to the DNA and the targets were pulled down and cleaned up (using Twist fast hybridization reagents and Thermo DynaBeads MyOne streptavidin-coupled beads). After a final PCR amplification (KAPA HiFi) and SPRI clean-up, a pool of all samples underwent quality control by Agilent Bioanalyser and sequenced in a single S4 lane of Illumina NovaSeq 6000.

Epithelial, fibroblast and blood cell fractions were estimated using EpiDISH and hEpiDISH[58]. The latter allows hierarchical deconvolution, first relying on centEpiFibIC to estimate epithelial, fibroblast and blood fractions, and applying centDHSbloodDMC to deconvolute the different types of blood cell. The median epithelial fraction across all 187 swabs was 95.1% (Extended Data Fig. 2h). Most of the non-epithelial cells were neutrophils, probably a result of saliva contamination of the buccal swabs.

As a complementary analysis of blood contamination in the buccal swab samples, we compared the VAF of blood mutations in buccal swabs. To do so, we used 43 pairs of buccal and archival blood samples in which the date of collection of the blood sample was within 3 years of the buccal swab, and which contained at least 1 large clone in blood (VAF ≥ 1%). The median of the ratio of buccal VAF to blood VAF for 58 blood mutations that met these criteria was 0.076, which provides an alternative estimate of the median blood contamination in these samples around 7–8%.

## Removal of DNA contamination

The ability of NanoSeq to detect somatic mutations in single molecules of DNA makes it particularly sensitive to DNA contamination, either from other humans (calling germline SNPs from the contaminant individual as somatic mutations in the affected sample) or from other species with sufficient conservation to map to the human genome (which is more likely in targeted NanoSeq due to the higher conservation of coding regions).

**Human DNA contamination.** We have previously shown[8] that when analysing whole-genome NanoSeq data, the percentage of contaminating DNA can be estimated using verifybamID[60]. However, we found verifybamID to be unreliable for targeted NanoSeq data. To qualitatively detect human DNA contamination on targeted NanoSeq data, a useful metric is the fraction of all substitutions filtered by the 'SNP+noise' mask. Although useful, this metric may not be reliable for samples with low duplex coverage and few mutations. As a complementary approach, we genotyped common SNPs in targeted regions to identify homozygous alternative (non-reference) SNPs. Presence of reference bases at these sites is indicative of contamination. Although this is not a direct estimate of the percentage of contamination given the difficulty of determining the genotype of the contaminant at those alternative homozygous SNPs, it can serve as a sensitive indicator of inter-individual DNA contamination.

We called SNPs with bcftools[61] using the following commands: bcftools mpileup --max-depth 20000 -Ou -f $genome $bam | bcftools call --ploidy GRCh37 -mv -Ob -o BCFTOOLS/$OUT_PREFIX.calls.bcf; bcftools view -i '%QUAL > = 100' BCFTOOLS/$OUT_PREFIX.calls.bcf > BCFTOOLS/$OUT_PREFIX.calls.filtered.vcf.

For the assessment of contamination, we restricted the analysis to SNPs overlapping both our SNP mask and our targeted panel. We used bam2R (from the deepSNV R package)[62] to obtain the number of reads supporting the alternative and reference alleles, and kept SNPs with a mean coverage across samples greater than 200×. For each SNP in each sample, the genotype was set to 'NA' if the coverage was less than 20×, to alternative homozygous (1/1) if the VAF was greater than 0.8, to heterozygous (0/1) if the VAF was between 0.3 and 0.7, and to reference

homozygous (0/0) if the VAF was less than 0.1. Finally, we only kept SNPs seen in 2 or more samples and in fewer than 1,000 samples. For each homozygous SNP, we calculated the reference fraction and we report the median across all homozygous SNPs in the sample. We considered 17 samples with a median reference base VAF > 0.01 at non-reference homozygous SNP sites to be contaminated and excluded them from all further analyses (Extended Data Fig. 2d).

**Cross-species contamination.** Donors were requested to rinse their mouths before buccal swab collection to minimize non-human DNA contamination from food or bacteria. However, some samples showed evidence of non-human DNA contamination, which resulted in mismapping of non-human DNA reads to the human genome, detectable as an excess of clustered synonymous mutations. To systematically identify these samples, we used Kraken v.2 (ref. 63), using 1 million unmapped reads per swab and a database of potential sources of contamination able to map to the human genome: *Mus musculus*, *Bos taurus*, *Ovis aries*, *Sus scrofa*, *Equus caballus*, *Oryctolagus cuniculus*, *Meleagris gallopavo* and *Gallus gallus*. Bacterial contamination should not be a problem given their sequence divergence from the human genome. In addition, for each sample we calculated the global dN/dS ratio across passenger genes, and compared the contamination fractions estimated with Kraken with the observed dN/dS ratios. dN/dS ratios decrease with non-human contamination because of evolutionary conservation of non-synonymous sites. On the basis of the impact of contamination on dN/dS ratios (Extended Data Fig. 2e), we excluded from further analyses 132 samples with more than 0.25% of non-human unmapped reads.

## HPV detection and characterization

The genome sequence of 19 HPV types considered high-risk[64] were retrieved from GenBank. We built a multiple sequence alignment of these genomes with MAFFT[65] using Jalview[66]. Based on conservation across these highly divergent HPV strains, we retained roughly 3,000 bp for each of the strains to design HPV-specific probes that we included in our Twist target gene panel. The GenBank accession numbers of the 19 selected HPV types were: KU298887.1, KU298893.1, KU298928.1, KX514417.1, KX514421.1, KX514431.1, KY225967.1, LR862061.1, LR862064.1, LR862079.1, MT218010.1, MT783412.1, MT783416.1, MT783417.1, MZ374448.1, MZ509108.1, NC_001357.1, NC_001526.4 and NC_001583.

Once our targeted sequencing data were mapped to the human genome we retrieved the unmapped reads and remapped them to the genomes of the 19 HPV strains using BWA-mem[57]. Mapping results were reviewed manually to distinguish between unreliable mappings (very repetitive, low-complexity, soft-clipped reads) and probably true HPV sequences. For ambiguous cases, we searched the mapped read with BLAST against the National Center for Biotechnology Information's non-redundant nucleotide database. This allowed us to identify some hits to HPV strains not originally covered in our panel.

We detected HPV in 12 samples, in some cases supported by thousands of reads while in others by as little as one single read. Several HPV strains were detected. The following (anonymized) list of donors show the results: X1 donor (HPV 16, 44 reads), X2 (HPV 53, 6 reads), X3 (HPV 33 and HPV 58, 20 and 5 reads), X4 (HPV 33, 115 reads), X5 (HPV 53, 6667 reads), X6 (HPV 59, 12 reads), X6 (HPV 56, 707 reads), X7 (HPV 51, 236 reads), X8 (HPV 56, 1 read), X9 (HPV 21 not in panel, 2 reads), X10 (HPV 24 not in panel, 1 read), X11 (HPV 30, not in panel, 1 read) and X12 (HPV 33, 6 reads).

Given that only 12 out of 1,042 samples had detectable HPV presence using the targeted capture and that this is not a validated assay for HPV detection, we were unable to study the impact of HPV on the mutation and selection landscape in the oral epithelium, which remains an important question for future studies. Instead, we excluded these 12 samples from the epidemiological regression analyses to reduce the risk of confounding effects.

## Germline genotyping

**Genotyping array data.** Pre-existing array genotyping data from TwinsUK were used for GWAS and other analyses. The samples had been genotyped with the following arrays: HumanHap300, Human-Hap610Q, 1M-Duo and 1.2MDuo 1M. Following genotype calling, some samples were excluded from analyses involving genotyping data based on different criteria: a sample call rate less than 98%, heterozygosity across all SNPs that were 2 or more standard deviations from the sample mean, evidence of non-European ancestry as assessed by principal components analysis comparison with HapMap3 populations, observed pairwise identity by descent probabilities suggestive of sample identity errors. We also used identity by descent probabilities to correct misclassified zygosity. We then excluded SNPs using the following criteria: Hardy–Weinberg $P < 10^{-6}$, assessed in a set of unrelated samples; minor allele frequency of 1%, assessed in a set of unrelated samples; SNP call rate less than 97% (SNPs with a minor allele frequency of 5% or more) or less than 99% (for 1% less than or equal to minor allele frequency of less than 5%). Following genotype and sample filtering, the data were imputed using the Haplotype Reference Consortium reference panel and SNPs with an imputation $R^2 < 0.5$ were excluded.

**Germline genotyping from sequencing data.** For analyses relying on common SNPs, we called SNPs using bcftools as described in the DNA contamination section. For analyses relying on both common and rare SNPs we run GATK's HaplotypeCaller (v.4.0.1.2)[67], using default options, setting ploidy to 2 except for the male chromosome X (haploid) and providing dbSNP v.141 (ncbi.nlm.nih.gov/snp, ref. 68) for annotation of the calls. The resulting VCF files were intersected with our panel regions using bedtools[69] and missing genotypes were annotated as REF with bcftools +missing2 (ref. 61), on the basis of the high coverage available.

## Selection analyses

A detailed description of the methods used to analyse positive and negative selection in this study is provided in Supplementary Note 2. This includes a description of the new one-sided tests in dNdScv, the use of duplex coverage correction in dNdScv, estimates of the number of driver mutations in the dataset and a description of dN/dS analyses at the level of single sites and groups of functionally related sites within genes.

## Mutation burdens and signatures

Mutation burden is defined as the number of mutations per base pairs in a given region, and it is calculated in NanoSeq data as the number of mutant bases divided by the total number of bases sequenced with duplex information. Estimating mutation rates from targeted data can have several challenges. First, the mutation burden of a given region of the genome will be affected by its sequence composition. We can remove this confounding effect by correcting mutation burdens by the trinucleotide frequencies of a targeted region (relative to the whole genome) and the mutability of each trinucleotide, as described before[8]. Whereas this corrects for the effect of different sequence composition, it does not correct for a systematic difference in the mutability of different regions, such as genic and intergenic sequences, as explained in the text. RE-NanoSeq (and the full whole-genome NanoSeq protocols introduced in this study) can be used for an unbiased genome-wide measurement of mutation burdens (for example, Extended Data Fig. 4a–f and Supplementary Note 3). Second, when estimating mutation rates from gene sequences, particularly from panels of positively selected genes, positive selection can lead to an inflation of the apparent mutation rate. To avoid this, the mutation burdens described in this paper were estimated only from passenger genes. Synonymous sites can also be used as a proxy for the neutral mutation rate, as described before[2]. Finally, mutation burdens estimated from targeted regions can be inflated or deflated by the undue influence of one or a few large clones. For example, if a sample is dominated by a large clone, the presence of a passenger mutation in the clone overlapping the target region would lead to an overestimation of the mutation burden, whereas the absence of any mutation in the clone in the target region would lead to a modest underestimate of the burden. This is apparent in the targeted NanoSeq data for blood (Fig. 1c), in which a few samples show inflated burden estimates due to high VAF passenger mutations. Some duplex sequencing studies avoid this by counting each mutant site only once, but this leads to a systematic underestimation of mutation burdens, leading to lower bound estimates of the mutation burden. Instead, for the targeted NanoSeq blood data, we calculated the CIs for the mutation burden using Poisson bootstrapping of the mutant sites, resulting in wider CIs when one or a few sites had an undue influence in the burden estimate. In general, burden estimates from targeted NanoSeq are expected to be most reliable when working with highly polyclonal samples or when the size of the panel is considerably larger than the inverse of the mutation rate per base pair.

We inferred mutational signatures of SBS using the sigfit (v.2.1.0) R package[70]. Genome strand information for each target gene was used to produce transcriptional strand-wise (TSW) trinucleotide mutation catalogues (192 mutation categories) for mutations within genes, using the build_catalogues function in sigfit. Inference was performed for a range of signature numbers ($N = 2,...,5$), using the TSW mutation counts from 92 oral epithelium samples having 500 or more mutations each. To account for variation in sequence composition, observed mutation opportunities (trinucleotide frequencies based on the NanoSeq coverage per site for each sample) were supplied to the extract_signatures function. Mutation opportunities were assumed to be equal between the transcribed and untranscribed strands. The best-supported number of signatures, on the basis of overall goodness-of-fit and consistency with known COSMIC signatures (v.3.0; cancer.sanger.ac.uk/signatures), was found to be $N = 2$. Of the two inferred signatures, signature A corresponded to a combination of COSMIC signatures SBS1 (6%) and SBS5 (94%) (cosine similarity 0.90), whereas signature B was highly similar to COSMIC SBS16 (cosine similarity 0.97). To estimate the contribution of both signatures to all oral epithelium samples, these two signatures were fitted to the TSW mutation counts for each sample using the fit_signatures function. Signature burdens (mutations per diploid genome attributed to each signature) were calculated by multiplying the signature exposure estimates by the whole-genome passenger mutation burden estimates for each sample. Before plotting using the plot_spectrum function, signatures were transformed to a genome-relative representation by scaling their probability values according to the corresponding whole-genome human trinucleotide frequencies, using the convert_signatures function.

A high rate of T>C mutations at ApT dinucleotides is common to the COSMIC SBS5 and SBS16 signatures. To explore whether these T>C mutations are caused by similar underlying processes, we studied the extended (pentanucleotide) sequence context of T>C mutations in several datasets. To do so, we obtained TSW pentanucleotide counts for T>C substitutions (256 mutation categories) applying a custom R function to mutations in the following sample sets: (1) matched blood samples ($n = 371$); (2) hepatocellular carcinoma (Liver HCC) samples from the Pan-Cancer Analysis of Whole Genomes study[71] (downloaded from dcc.icgc.org/pcawg) for which signature fitting estimated a COSMIC SBS16 exposure greater than 0.2 ($n = 4$); (3) oral epithelium samples with signature B exposure greater than 0.25 ($n = 121$) and (4) oral epithelium samples with signature B exposure less than 0.25 ($n = 921$). Before plotting using custom R functions, pentanucleotide catalogues were transformed to a genome-relative representation by scaling mutation counts according to the corresponding whole-genome human pentanucleotide frequencies. The results of this analysis are described in Extended Data Fig. 8 and Supplementary Note 3.

Mutation catalogues of DBSs (78 mutation categories) and indels (83 mutation categories) were produced for mutations in the following sample sets: (1) all oral epithelium samples ($n$ = 1,042); (2) oral epithelium samples from heavy-smoking non-drinking donors ($n$ = 27) and (3) oral epithelium samples from non-smoking non-drinking donors ($n$ = 224). DBS catalogues were produced using a custom R function, whereas indel catalogues were produced using the indel.spectrum function in the Indelwald tool (24 September 2021 version; github. com/MaximilianStammnitz/Indelwald). Although we attempted both de novo extraction and fitting of mutational signatures to the DBS and indel catalogues, mutation numbers were not large enough to allow inference of informative signatures or exposures. Mutation spectra for DBS and indels were plotted using the plot_spectrum function in sigfit. The results of this analysis are described in Extended Data Fig. 8 and Supplementary Note 3.

### Regression analyses

To test for associations between epidemiological variables and rates of mutational signatures or driver mutation frequencies, we used mixed-effect regression models (lmer function in the lme4 R package[72]) as described below.

**Outcome variables.** For the analyses shown in the main text, we ran a separate regression model for each outcome variable: SNV burden, signature A burden, signature B burden, indel burden, dinucleotide burden, the sum of all driver frequencies in a sample and the driver density per sample for ten major driver genes. To avoid excessive loss of statistical power due to multiple testing correction across all outcome variables and predictors, and to focus on the genes with the highest information content, we restricted the regression analyses to 10 driver genes with ≥1,000 or more coding mutations across mutation types (missense, truncating or no-SNVs) in the dataset, and with dN/dS ≥ 5 (that is, with an estimated driver fraction of 80% or more).

**Predictor variables.** For the analyses in the main text, we selected nine variables as predictors in multiple regression models, including major oral cancer risk factors as well as other potentially relevant variables: age, sex (female individuals or male individuals), pack-years, drink-years, type 2 diabetes (T2D, Y/N), body mass index, missing teeth, physical activity score (International Physical Activity Questionnaire (IPAQ)) and cancer history (Y/N). The twin structure was modelled with a random effect. The R code used for these and supplementary regressions is provided in the Supplementary Code, but for illustrative purposes the structure was as follows:

$$\text{lmer(SNVburden} \sim \text{age} + \text{sex} + \text{packyears} + \text{drinkyears} + \text{T2D}$$
$$+\text{BMI} + \text{missingteeth} + \text{IPAQ} + \text{cancer}$$
$$+(1|\text{familyID}), \text{REML} = F)$$

Only samples with a mean duplex coverage across genes ≥200 dx and available metadata for all the predictor variables and the outcome variable in each multiple regression model were included for analyses. Twelve samples with potential evidence of HPV reads (above) and six samples with a self-reported history of chemotherapy were excluded from the regression. These variables may be expected to have mutagenic and/or selectogenic effects on the oral epithelium, but the number of affected donors was too low for a robust analysis in the current study.

$P$ values were calculated for each covariate in each multivariate regression model using a likelihood-ratio test by comparing the likelihood of the full model with a model without each variable, using the drop1 function in R. Multiple testing adjustment using the Benjamini–Hochberg procedure was then applied to all $P$ values in the main text analyses (126 tests: 14 predictors × 9 outcome variables).

### Extra regression models, GWAS and heritability analyses

Extra regression analyses, including using extended medication data as predictors, interaction analyses between smoking and alcohol, and measures of selection (corrected for mutation rates) as outcome variables for the detection of selectogenic influences, are described in Supplementary Note 7. Methods and supplementary results for GWAS analyses and heritability tests are described in Supplementary Note 8.

### Reporting summary

Further information on research design is available in the Nature Portfolio Reporting Summary linked to this article.

### Data availability

Sequencing data have been deposited in the European Genome-Phenome Archive (EGA) under accession numbers EGAD00001015618 (TwinsUK_TargetedNanoSeq_Buccal), EGAD00001015619 (TwinsUK_TargetedNanoSeq_Blood), EGAD00001015620 (TwinsUK_ExomeNanoSeq_Buccal), EGAD00001015621 (TwinsUK_RENanoSeq_Buccal), EGAD00001015622 (TwinsUK_TargetedEMSeq_Buccal), EGAD00001015623 (TwinsUK_TargetedEMSeq_Blood) and EGAD00001015624 (Sanger_NanoSeq_RandD). Data access for EGAD00001015618, EGAD00001015619, EGAD00001015620, EGAD00001015621, EGAD00001015622 and EGAD00001015623 is managed by TwinsUK (EGAC00001000274) (Supplementary Table 7). Patient metadata are managed by TwinsUK. Anonymized mutational data are available in Supplementary Tables 8 and 9.

### Code availability

Supporting code can be found at GitHub through https://github.com/cancerit/NanoSeq/ and https://github.com/im3sanger/dndscv, and as an accompanying R HTML MarkDown file. All analyses have been done using the human genome assembly GRCh37.

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

**Acknowledgements** We are grateful to members of the TwinsUK cohort for volunteering samples and metadata for this study. We thank G. Davey Smith and P. M. Visscher for advice on epidemiology and heritability analyses; B. Sexton, A. Peltan and New England BioLabs for advice during protocol development; A. Teschendorff for advice on methylation; A. P. Butler

and V. Offord for assistance with establishing the bioinformatic pipeline; Y. Hooks for sample processing; K. Roberts, T. Baxter, K. Smith, N. Yilmaz, V. Uksaite, E. Ferla, H. Savin, L. Allen and C. Latimer for helping with sample shipments; and the CASM support and DNA pipeline teams at the Wellcome Sanger Institute for their essential role in data generation. This research was financed in whole, or in part, by the Wellcome Trust 220540/Z/20/A. For the purpose of Open Access, the author has applied a CC BY public copyright licence to any Author Accepted Manuscript version arising from this submission. I.M. is funded by Cancer Research UK (C57387/A21777), the Dr Josef Steiner Cancer Research Foundation and the Wellcome Trust. TwinsUK is funded by the Wellcome Trust, Medical Research Council, Versus Arthritis, European Union Horizon 2020, Chronic Disease Research Foundation, Zoe Ltd, the National Institute for Health and Care Research Clinical Research Network and Biomedical Research Centre based at Guy's and St Thomas' NHS Foundation Trust in partnership with King's College London.

**Author contributions** A.R.J.L., F.A., P.A.N. and I.M. conceptualized the project with support from M.R.S., P.J.C., R.R. and K.S.S. A.R.J.L., F.A., G.K., A.B.-O. and I.M. led the data analysis with support from P.A.N., A.L.R., M.F.Ö., M.D.C.N., M.J.P., N.W., D.A., L.M.R.H., J.R.B. and M.D.Y. A.R.J.L., P.A.N. and S.V.L. led the experimental work with support from N.B., S. Widaa, W.C., M. Morra, L.S., M. Mayho and N.M.-S. F.A., R.E.A., T.C., A.B. and I.M. developed algorithms and

software. A.R.J.L., F.A., S.V.L., S. Widaa and I.M. contributed to method development. A.L.R., J.S.E.-S.M., D.V., B.J., A.N., S. Wadge, K.T.A.M. and K.S.-P. collected samples. L.O. and S.A.-G. helped with sample and project administration. A.L.P. and D.M.R. provided histology support. K.S.-P., H.C.M., M.R.S., P.J.C., R.R., K.S.S. and I.M. provided supervision. A.R.J.L., F.A., P.A.N., A.B.-O. and I.M. wrote the paper. All authors contributed to reviewing and editing it.

**Competing interests** I.M., M.R.S. and P.J.C are cofounders, R.E.A., M.D.Y. and P.J.C. are employees and I.M., M.R.S., N.B. and F.A. have consulted for Quotient Therapeutics Ltd. The other authors declare no competing interests.

**Additional information**
**Correspondence and requests for materials** should be addressed to Iñigo Martincorena.

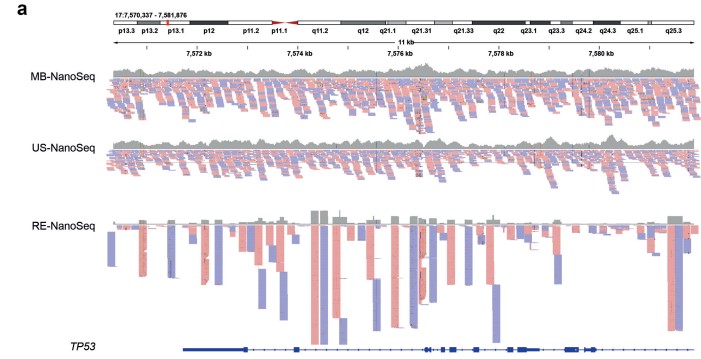

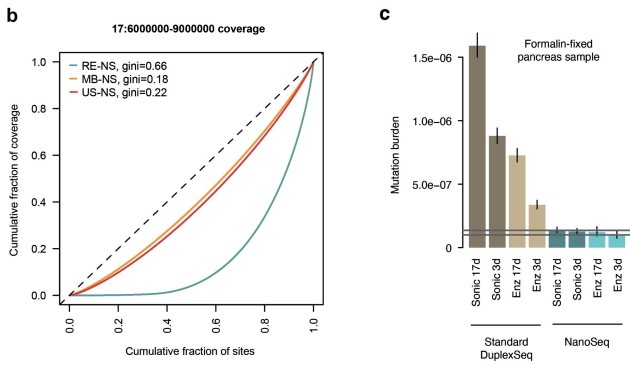

**Extended Data Fig. 1 | Sequencing coverage and description of the cohort.**
**a**, Integrative Genomics Viewer image[73] showing sequencing reads and coverage
for three NanoSeq protocols at the *TP53* locus: sonication followed by mung
bean nuclease treatment NanoSeq (MB-NanoSeq, top), Ultrashear enzymatic
fragmentation NanoSeq (US-NanoSeq, middle), and restriction enzyme
NanoSeq (RE-NanoSeq, bottom). Reads are coloured by mapping orientation.
MB- and US-NanoSeq show the coverage in merged bams containing all the cord
blood samples analysed in this study, whereas RE-NanoSeq shows the coverage
in 12 buccal samples. **b**, Evenness of coverage for the three protocols described
in **a**. Lorenz coverage curves for three million sites sampled from the
corresponding bams (locus chr17:6000000-9000000), resulting in Gini
coefficients of 0.66 for RE-NanoSeq, 0.18 for MB-NanoSeq and 0.22 for
US-NanoSeq. **c**, Genome-wide mutation burden estimates, as mutations per base
pair, for adult pancreas samples formalin-fixed for 3 days (3 d) or 17 days (17 d),
and sequenced using four different protocols (*sonic* refers to sonication and *enz*
to enzymatic fragmentation). Error bars show Poisson 95% CIs. Horizontal lines
denote the burden and the associated Poisson 95% CIs in a matching fresh-frozen
sample.

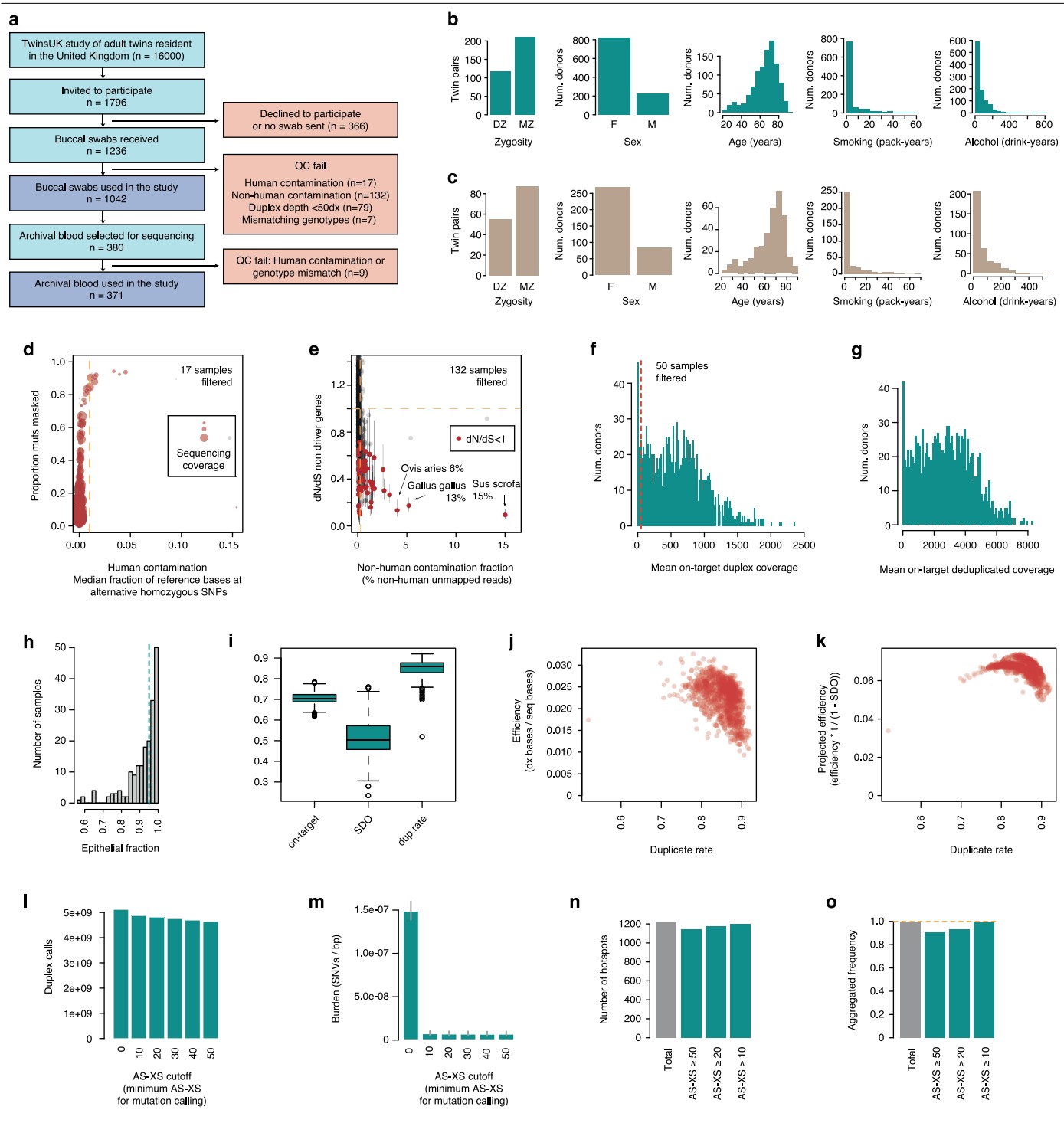

**Extended Data Fig. 2** | See next page for caption.

**Extended Data Fig. 2 | Targeted NanoSeq study design and quality metrics.**
**a**, Flow diagram describing the selection of the donor cohort used in this study.
**b**, **c**, Distribution of zygosity, sex, age, smoking (pack-years) and drinking
(drink-years) values for (**b**) buccal swab (n = 1,042) and (**c**) archival blood sample
donors (n = 371). **d**, Identification of samples contaminated with human DNA
from another individual, comparing the proportion of mutation calls falling in
the 'SNP+noise' mask *versus* the median fraction of reference bases at alternative
homozygous SNPs; point size is proportional to the duplex coverage. The vertical
dashed line indicates our exclusion criterion for human contaminated samples
(>0.01). **e**, Identification of samples contaminated with non-human DNA,
comparing the dN/dS values in passenger genes *versus* the percentage of
unmapped reads mapping to a set of potential contaminant species. Horizontal
dashed line indicates neutral dN/dS=1. Red points indicate samples with upper
bound 95% CI dN/dS ratio <1. Vertical dashed line shows our exclusion criterion
for non-human contaminated samples (>0.25). **f**, Histogram of duplex coverage
(dx) in the buccal swab cohort, at on-target and near-target regions. Vertical
dashed line shows our exclusion criterion for low coverage samples (<50dx).
**g**, Distribution of the mean deduplicated coverage (×) in the buccal swab cohort,
at on-target and near-target regions. Raw sequencing coverage is ~6.6 times
higher due to the average 85% duplicate rate required for duplex consensus
calling. **h**, Estimation of epithelial fraction in buccal swab samples by targeted
enzymatic methylation sequencing. Vertical dashed line shows the median
epithelial fraction of 0.95. **i**, Sequencing quality metrics including the on-target
capture fractions, estimated excess in strand drop-out (SDO), and the achieved
duplicate rates for the buccal swab cohort. Random binomial sampling is
expected to cause lack of coverage in one of the DNA strands in a proportion of
cases. We estimated the excess in SDO by subtracting the observed and
expected SDOs. Box plots show the interquartile range, median, 95% confidence
intervals and outliers as dots for the buccal cohort (n = 1,042). **j**, Relationship
between duplicate rates and sequencing efficiency, measured as the number
of bases with duplex support divided by the total number of bases sequenced.
**k**, Relationship between duplicate rates and sequencing efficiency after
factoring in the on-target fraction (t) and the excess in strand drop-out (SDO).
**l**, Number of duplex calls as a function of the primary alignment score minus
secondary alignment score (AS-XS) threshold. **m**, Substitution burdens
calculated within each AS-XS threshold corrected for trinucleotide context.
Error bars for substitution burdens indicate Poisson 95% CIs. **n**, **o**, Hotspots
covered by different AS-XS thresholds, shown as (**n**) total number of hotspots
and (**o**) their aggregated frequency in TCGA. Horizontal dashed line indicates
the detection of all studied hotspots.

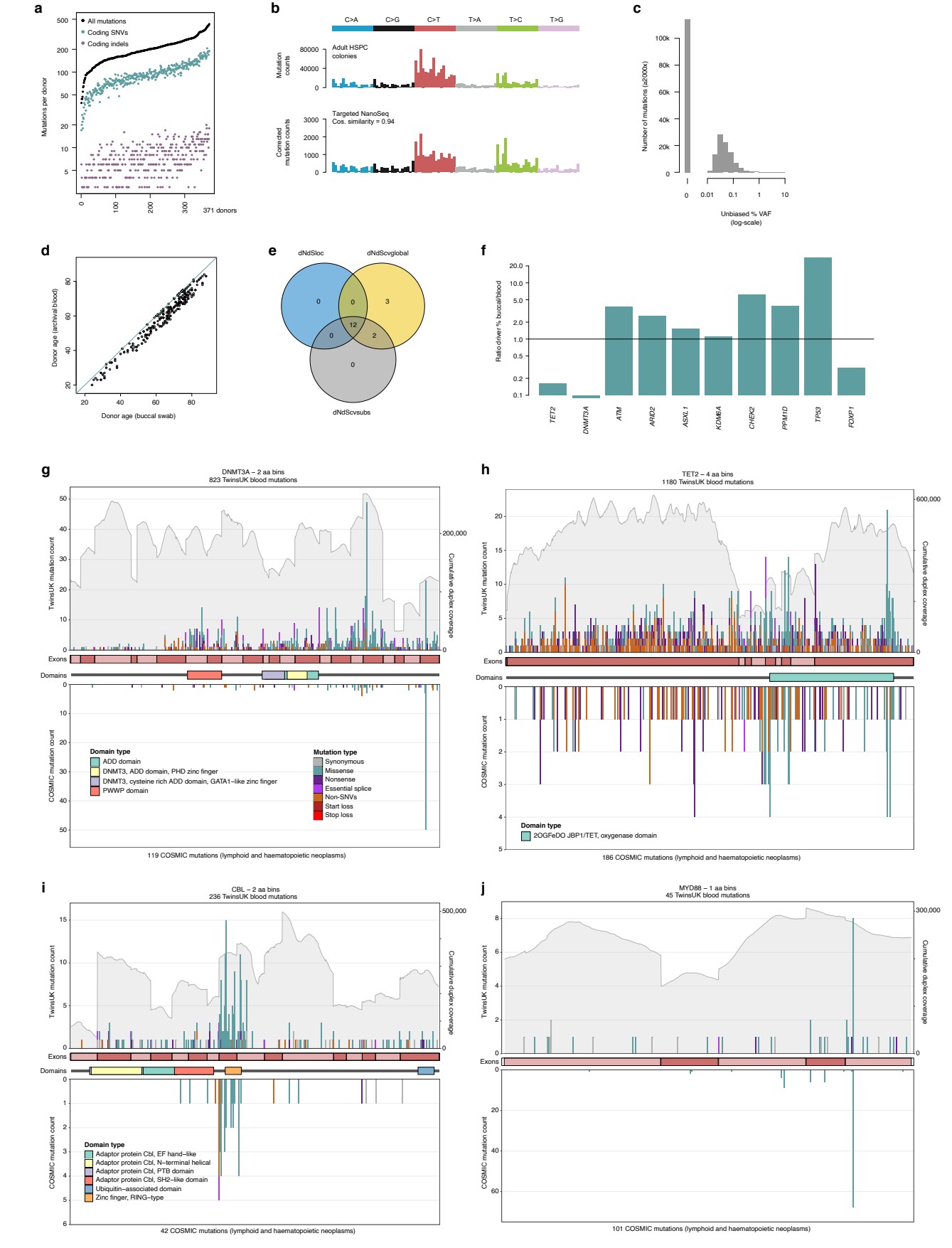

**Extended Data Fig. 3 |** See next page for caption.

**Extended Data Fig. 3 | Further description of the blood driver landscape.**
**a**, Numbers of total mutations, coding SNVs and coding indels identified in whole blood samples from 371 donors using targeted NanoSeq. **b**, Trinucleotide mutational spectra of adult haematopoietic stem and progenitor cell (HSPC) colonies from a previous study[56] and whole blood samples sequenced using targeted NanoSeq (corrected by the ratio of genomic to observed trinucleotide frequencies). **c**, Distribution of ($\log_{10}$-scaled) unbiased VAFs for mutations with sequencing depth ≥2000× identified in 371 whole blood samples using targeted NanoSeq. Unbiased VAFs are calculated from read bundles not used for duplex variant calling. **d**, Relationship between donor ages (years) for matched buccal swab samples and archival blood samples (n = 371). The diagonal line represents the identity function, $y = x$. **e**, Venn diagram summarising the overlaps between three approaches for identifying genes under significant positive selection by *dNdScv* in the archival blood targeted NanoSeq data. **f**, Ratio of estimated driver densities between buccal swab samples and archival blood samples, for 10 genes identified as being under gene-level significant positive selection in both blood and buccal swab samples. **g-j**, Mutation barplots for *DNMT3A*, *TET2*, *CBL* and *MYD88*. The *x*-axis represents coordinates along the coding sequence. Exons and protein domains are indicated along the *x*-axis. The *y*-axis represents number of mutations, either in the 371 TwinsUK archival blood samples used in this study (top) or in lymphoid and haematopoietic neoplasm samples (whole-exome and whole-genome) from the COSMIC database (bottom). Mutations are coloured according to mutation consequence category. Grey shading indicates cumulative duplex coverage across TwinsUK archival blood samples.

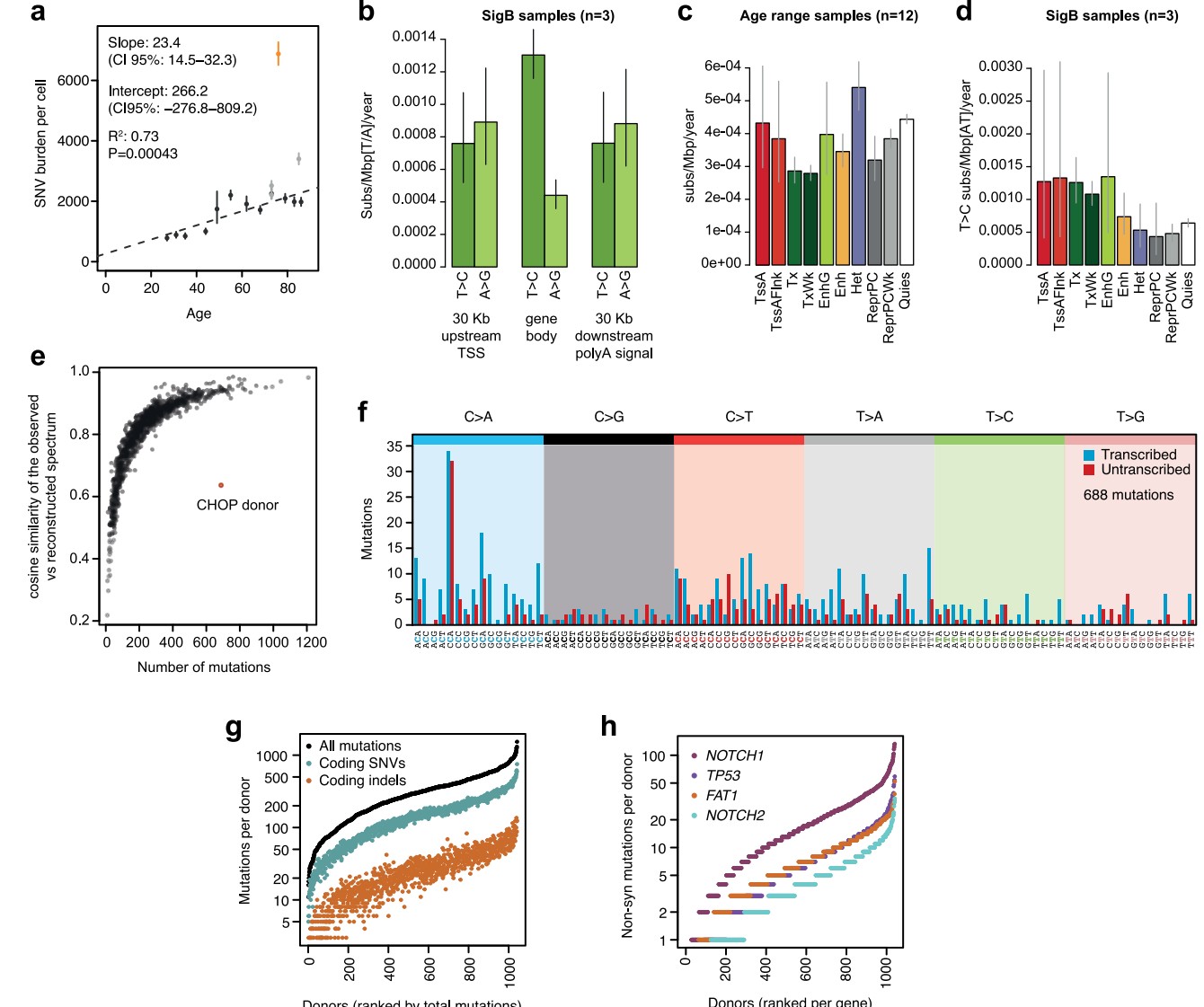

**Extended Data Fig. 4 | Restriction enzyme NanoSeq and targeted NanoSeq on buccal swabs. a**, Regression of SNV mutation burden with age for 12 samples selected from across the age range (black dots), three samples with high SigB signature contribution (grey), and one sample with a high mutation burden from a donor with a history of CHOP chemotherapy treatment (orange). The regression results listed within the plot were generated using only the 12 samples randomly selected for their age range. Error bars show Poisson 95% CIs for the estimated burdens (substitutions per cell). P-values calculated with t-test and 2 degrees of freedom. **b**, Transcription-coupled repair and damage in three donors with high contribution of SigB. Estimated substitution burdens plus their associated 95% CIs (error bars) across upstream, transcribed and downstream regions, showing T > C and A > G in the coding strand separately. **c**, Number of substitutions per Mbp per year in 12 age range donors for each of 10 major ENCODE chromatin states. Reference chromatin states were obtained from ENCODE E057 foreskin keratinocytes. Chromatin states BivFlnk, EnhBiv, TssBiv, TxFlnk, and ZNF/Rpts were removed given their smaller footprint and too large confidence intervals. Burdens were normalised to whole genome trinucleotide frequencies. Error bars show Poisson 95% CIs. **d**, Number of substitutions per Mbp per year in 3 donors with strong SigB exposure for each of 10 major ENCODE chromatin states. Only T > C rates are shown, calculated as the number of T > C substitutions observed and divided by the number of [TA] bps. Error bars show Poisson 95% CIs. **e**, Cosine similarities between the observed and reconstructed substitution profiles as a function of the number of mutations in each sample, highlighting the outlier donor with a history of CHOP chemotherapy treatment (brown). **f**, Transcriptional strand-wise trinucleotide SBS spectrum for the outlier CHOP donor. **g**, Numbers of total mutations, coding SNVs and coding indels identified in oral epithelium samples from 1,042 donors using targeted NanoSeq. **h**, Numbers of non-synonymous mutations identified by targeted NanoSeq per donor in oral epithelium for genes *NOTCH1*, *TP53*, *FAT1* and *NOTCH2*. Mutation counts are ordered independently for each gene.

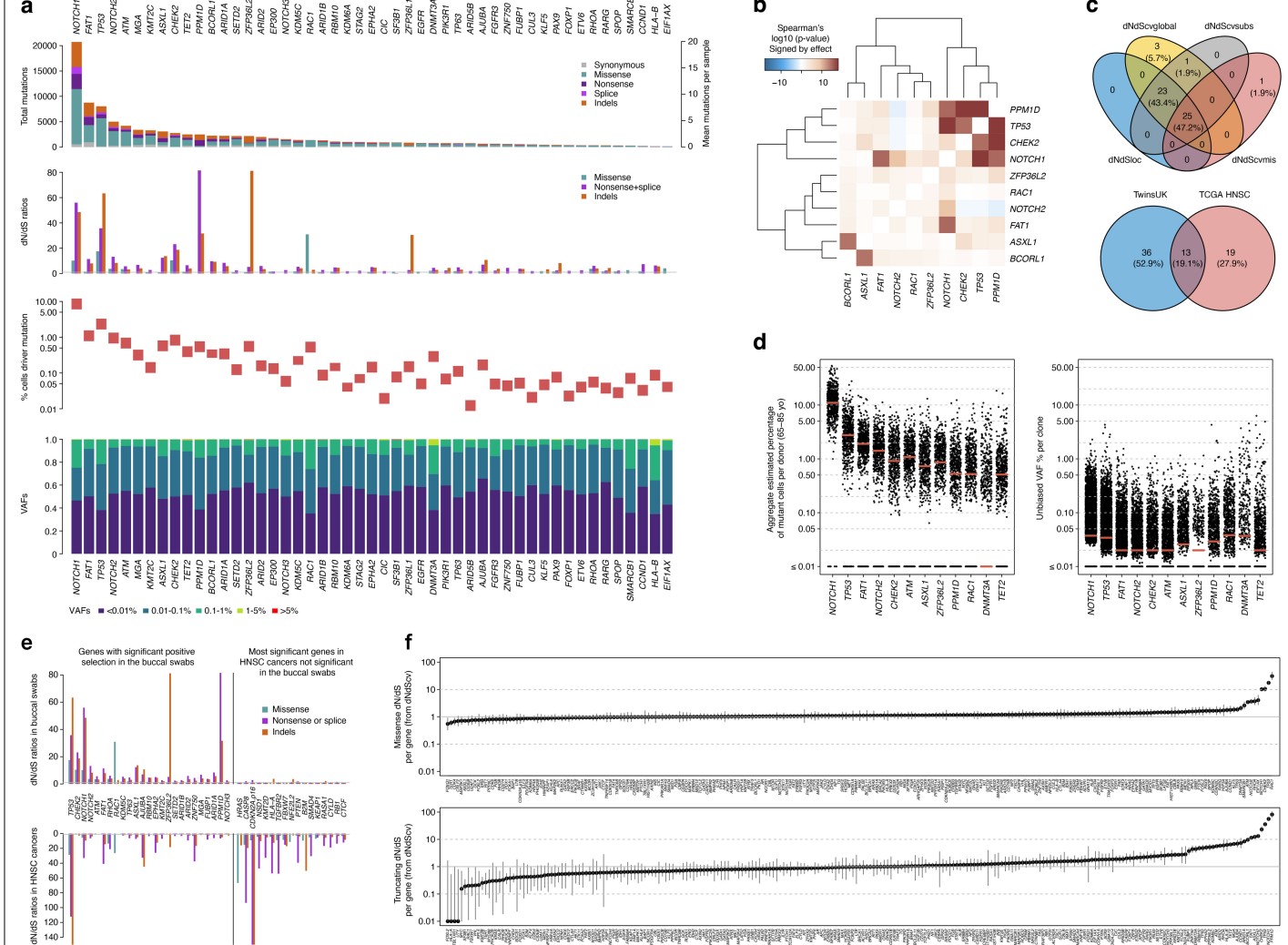

**Extended Data Fig. 5 | Full driver landscape in oral epithelium. a**, For the 49 significant driver genes in oral epithelium (of which *DNMT3A*, *TET2* and *FOXP*1 are likely attributable to low-level blood contamination, as shown in Extended Data Fig. 3f), panels show (top to bottom) mutation counts per mutation consequence category, dN/dS ratios per mutation consequence category (horizontal line indicates neutral dN/dS=1), estimated mutant cell percentages in donors aged 65-85, and the distribution of unbiased VAFs. **b**, Spearman's correlation and associated P-values between the generalised linear model residuals for driver burden across top driver genes (defined as genes with >1,000 driver mutations across mutational classes with >80% estimated driver fraction). **c**, Venn diagrams summarising (top) the overlaps between four approaches for identifying genes under significant positive selection genes in TwinsUK oral epithelium by dNdScv, and (bottom) driver genes in TwinsUK oral epithelium samples and head and neck squamous cell carcinomas (HNSC) in The Cancer Genome Atlas. **d**, **Left**. Estimated mutant cell fraction in oral epithelium per donor for 12 genes. Each dot represents one donor in the dataset (restricted to 65-85 year old donors, n = 583). The estimated mutant cell fraction represents an upper bound estimate using the sum of duplex VAFs multiplied by 2 for genes in diploid chromosomes (see **Methods** for an explanation of the assumptions and rationale). This analysis takes into account all non-synonymous mutations observed in each gene in each donor. The red line represents the median estimated cell fraction across donors. **Right**. Observed VAFs (shown as percentages) for all non-synonymous SNVs in each gene in each donor, restricted to sites with ≥1000× coverage (most values < 0.01% had unbiased VAFs = 0). Each dot represents one mutation. This highlights that the vast majority of mutations observed across genes have very low VAFs, with only a small number of mutations having VAFs>2% (largely in clonal haematopoiesis genes and a few in *TP53*). **e**, Comparison of dN/dS ratios per gene between healthy oral epithelium (top) and HNSC (bottom). **f**, Observed missense (top) and truncating (bottom) dN/dS ratios per gene for the 239 genes in the dataset, with 95% confidence intervals, showing that dN/dS ratios are close to 1 for the vast majority of genes, with only a small minority of genes showing clear negative selection.

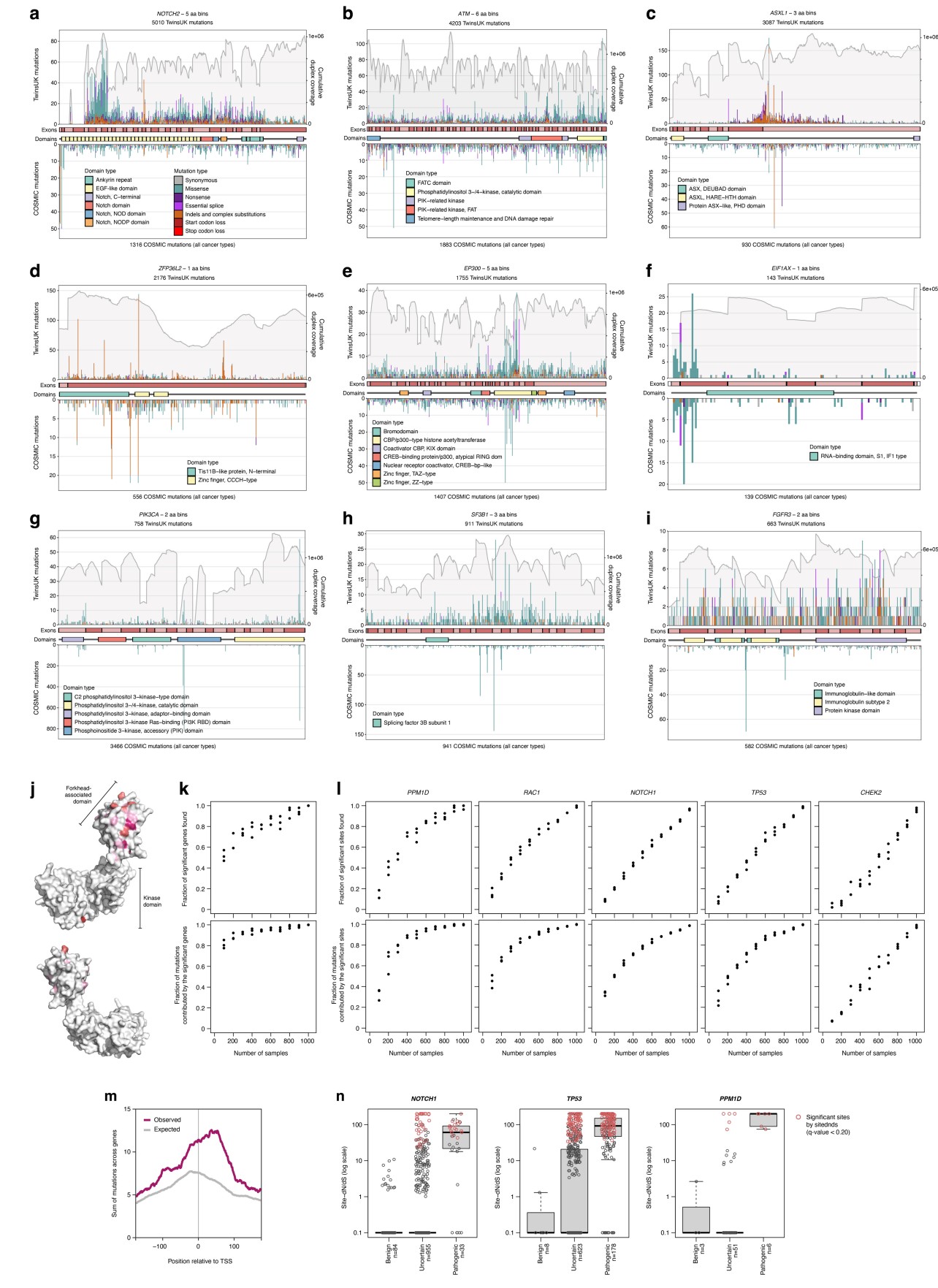

**Extended Data Fig. 6** | See next page for caption.

**Extended Data Fig. 6 | Distribution of mutations within selected buccal driver genes. a-i**, Mutation distribution within nine selected genes. The *x*-axis represents coordinates along the coding sequence. Exons and protein domains are indicated along the *x*-axis. The *y*-axis represents number of mutations, either in the 1,042 TwinsUK oral epithelium samples used in this study (top) or across whole-exome and whole-genome samples of any cancer type in the COSMIC database (bottom). Mutations are coloured according to mutation consequence category. Grey shading indicates cumulative duplex coverage across TwinsUK samples. **j**, Diagrams of the 3-dimensional structure of CHK2 (encoded by *CHEK2*), showing the clustering of sites under significant positive selection on one side of the forkhead-association domain. Residues with site-level dN/dS *q*-value < 0.01 are coloured. Shading intensity denotes degree of significance. **k**, Fraction of significant genes (top) and fraction of mutations contributed by genes identified as significant (bottom) identified by gene-level dN/dS across subsamples of the buccal swab cohort. **l**, Fraction of significant sites (top) and fraction of mutations contributed by sites identified as significant (bottom) by site-level dN/dS for five genes across subsamples of the buccal swab cohort. **m**, Observed and expected (*withingenednds*) density of mutations as a function of position relative to the transcription start site (TSS), aggregated across all targeted genes. **n**, Distribution of site-level dN/dS ratios for sites annotated in ClinVar as benign, pathogenic or of uncertain significance, in *NOTCH1*, *TP53* and *PPM1D*. Significant sites are shown in red. Box plots show the interquartile range, median, 95% confidence intervals and outliers as grey/red dots.

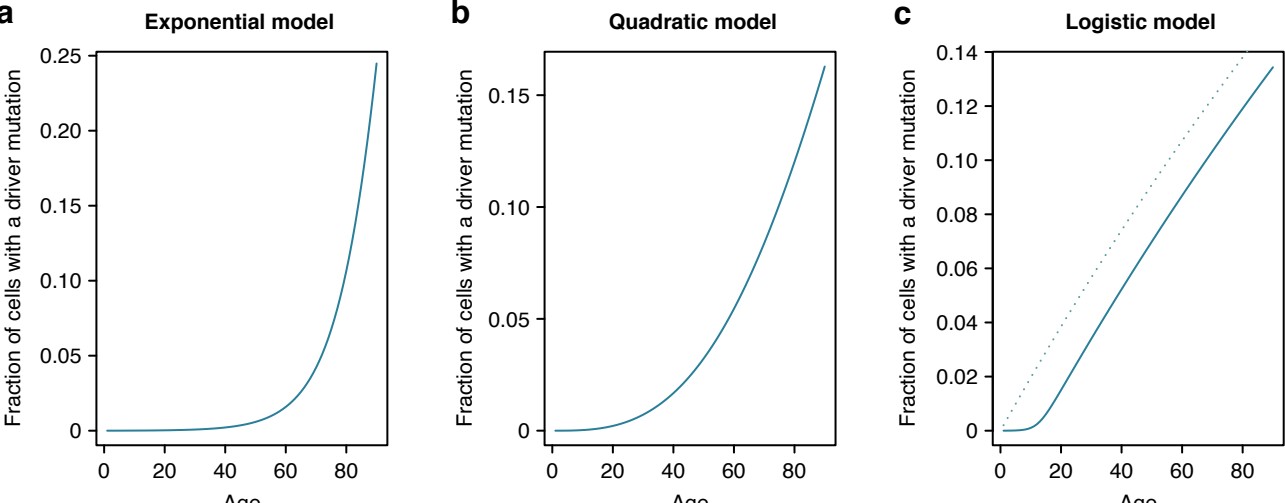

**a** Exponential model

**b** Quadratic model

**c** Logistic model

**Extended Data Fig. 7 | Models of clonal growth. a-c**, Graphs illustrating the increase in the fraction of cells carrying a driver mutation as a function of age, as predicted by (**a**) exponential, (**b**) quadratic and (**c**) logistic models of clonal growth. The dotted line in (**c**) represents the prediction from a linear growth model. In all three cases, we used a driver mutation rate per cell per year of $\mu = 4 \times 10^{-6}$ (1000 driver sites per genome, $4 \times 10^{-9}$ mutations per cell per year).

Other parameters for the three growth models were chosen to obtain a fraction of cells with a driver mutation around 10-15% by age 80, to facilitate comparison to the buccal swab data in Fig. 4a. The parameters used are the following: (**a**) exponential model using $r = 0.1$, (**b**) quadratic model using $r = 0.2$, (**c**) logistic model using $L = 500$ and $r = 0.5$.

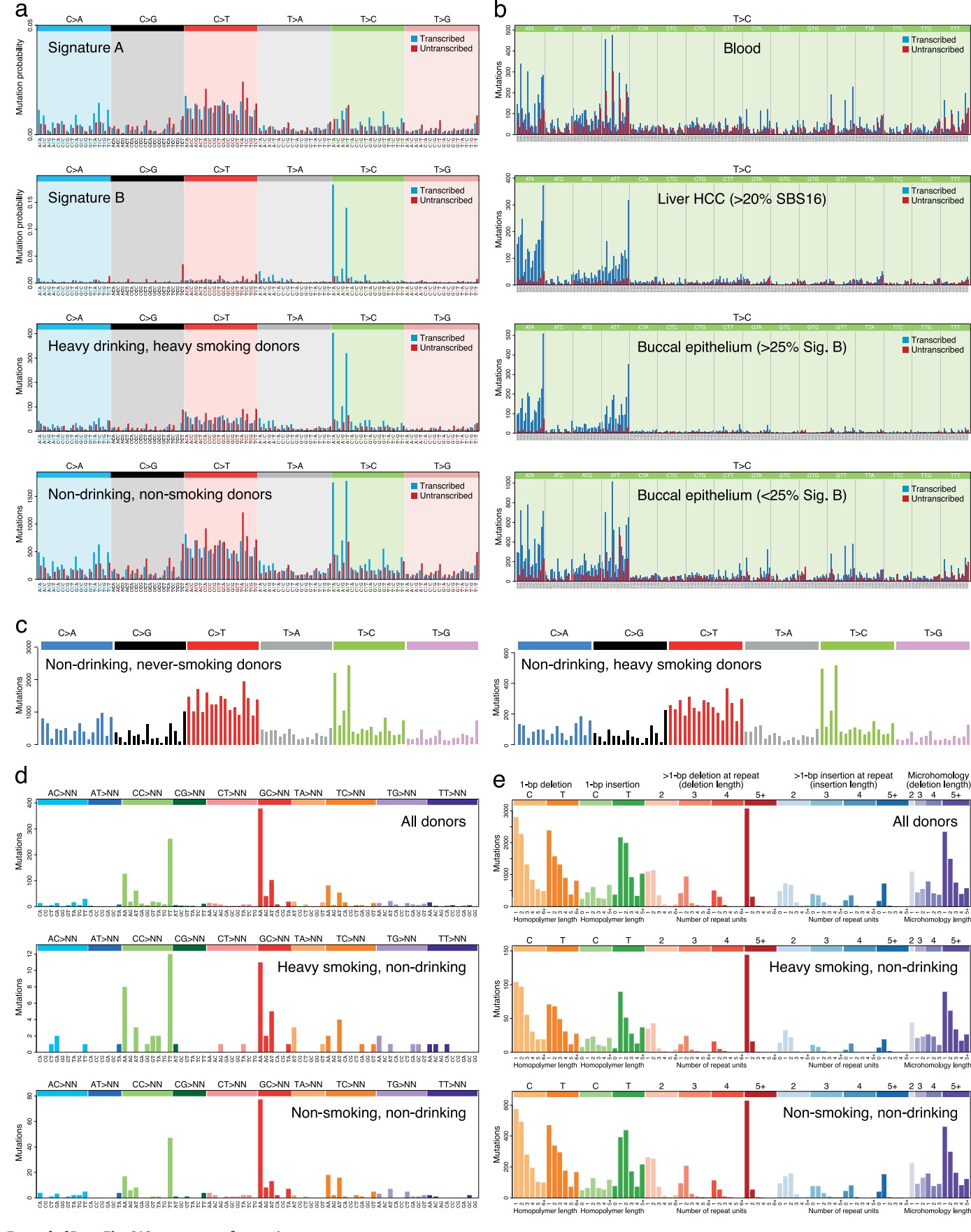

**Extended Data Fig. 8** | See next page for caption.

**Extended Data Fig. 8 | Mutational spectra of somatic single-base substitutions (SBSs), double-base substitutions (DBSs) and indels.** **a**, Transcriptional strand-wise versions of the trinucleotide SBS spectra shown in Fig. 4b. **b**, Transcriptional strand-wise pentanucleotide spectra of T > C SBSs in (top to bottom): blood samples (n = 371), hepatocellular carcinoma samples (Liver HCC; Supplementary Note 3) with >20% SBS16 exposure (n = 4), oral epithelium samples with >25% Signature B exposure (n = 121), and oral epithelium samples with <25% Signature B exposure (n = 921). **c**, Trinucleotide substitution spectra for non-drinking, never-smoking (left) and non-drinking, heavy smoking (right) donors. **d**, DBS spectra in (top to bottom): all oral epithelium samples (n = 1,042), oral epithelium from heavy smoking, non-drinking donors (n = 27), and oral epithelium from non-smoking, non-drinking donors (n = 224). **e**, Spectra of indels in the same sample sets shown in **d**.

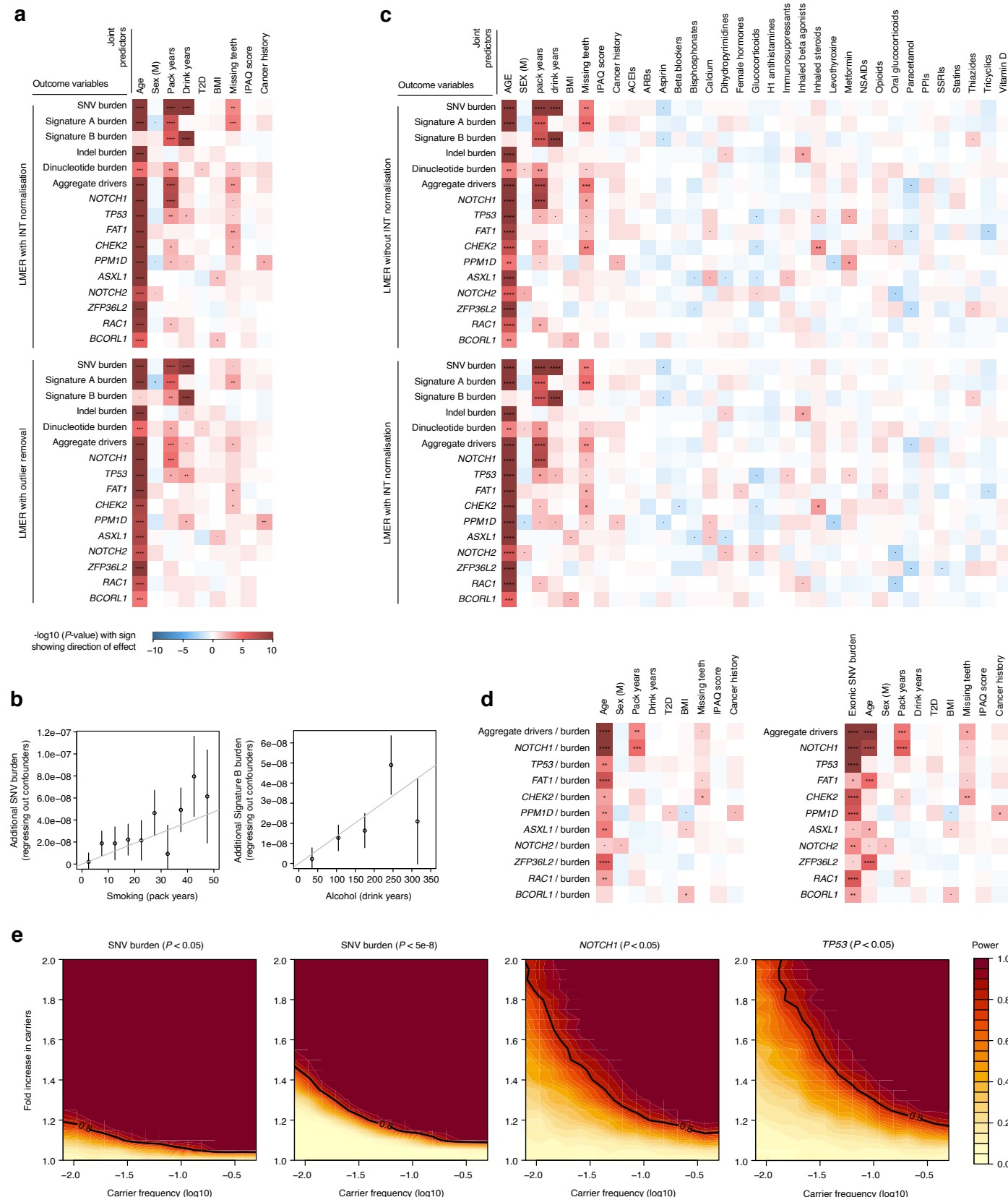

**Extended Data Fig. 9** | See next page for caption.

**Extended Data Fig. 9 | Additional regression models. a**, Heatmap of associations between different measures of mutation burden, signature burden or driver density ($y$-axis) and relevant donor metadata ($x$-axis), inferred using linear mixed-effects regression (LMER) models with (top) inverse-normal transformation (INT) or (bottom) outlier removal, where outliers are defined as those values larger than $3 \times IQR + Q_3$ (where IQR is the interquartile range and $Q_3$ is the third quartile for each outcome variable). The $P$-value of each association is calculated with a Likelihood ratio test and is indicated by both colour shading (red and blue for positive and negative associations, respectively) and asterisk labels (****: $q < 10^{-4}$; ***: $q < 10^{-3}$; **: $q < 0.01$; *: $q < 0.05$; •: $P < 0.05$; $q$-values were calculated with the Benjamini and Hochberg false discovery method). **b**, Additional SNV burden obtained after regressing out confounders using an LMER model for (left) smoking history across pack-year bins and (right) alcohol consumption history across drink-year bins. Error bars show 95% confidence intervals. Smoking pack-year bins: (−1,0] n = 632 (not shown), (0,5] n = 139, (5,10] n = 63, (10,15] n = 41, (15,20] n = 40, (20,25] n = 21, (25,30] n = 20, (30,35] n = 11, (35,40] n = 16, (40,45] n = 6, and (45,50] n = 4; Drink-year bins: (−1,0] n = 305 (not shown), (0,70] n = 378, (70,140] n = 200, (140,210] n = 92, (210,280] n = 31, and (280,350] n = 13. **c**, Heatmap (as described in **a**) with medication history included as predictors. BMI stands for body mass index; IPAQ score for International Physical Activity Questionnaire; ACEIs for angiotensin-converting enzyme inhibitors; ARBs for angiotensin receptor blockers; NSAIDs for non-steroidal anti-inflammatory drugs; PPIs for proton pump inhibitors; SSRIs for selective serotonin reuptake inhibitors. **d**, Heatmap (as described in **a**) using a multivariate LMER model with the driver density per gene per donor normalised by the exonic mutation burden in passenger genes per donor. Normalisation was achieved by either (left) using the ratio of driver density and mutation burden as a new outcome variable for each driver gene or (right) including the mutation burden as a covariate. **e**, Heatmaps showing the fraction of significant tests ($P < 0.05$ or 5e-8, as indicated above each heatmap) for simulations of different variables, for different effect sizes ($y$-axes), and different fractions of affected individuals ($x$-axes). The contour line in each heatmap shows the conditions that provide 80% power (i.e. 80% of significant tests).

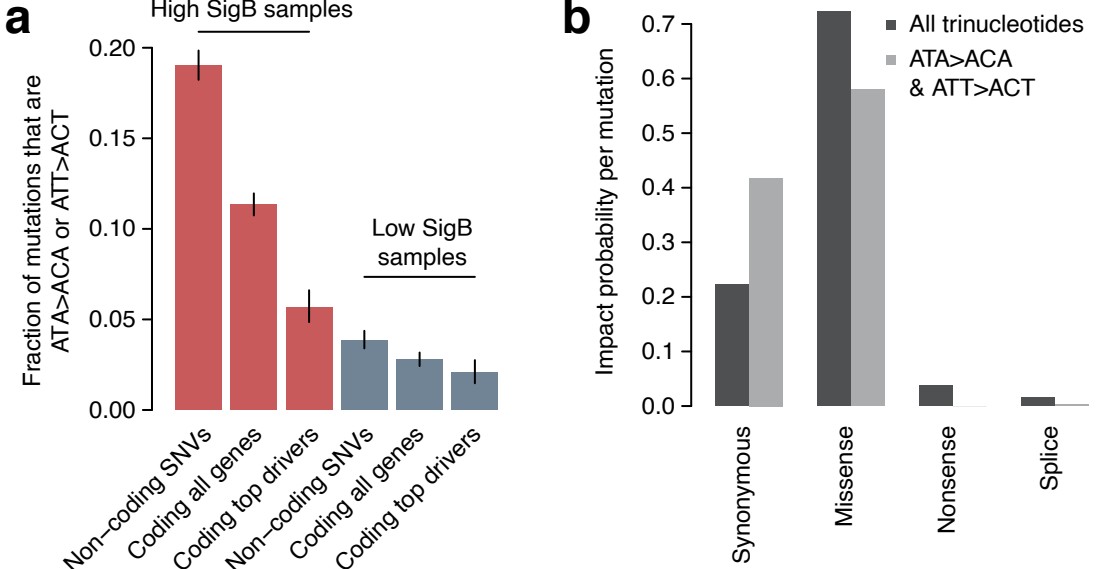

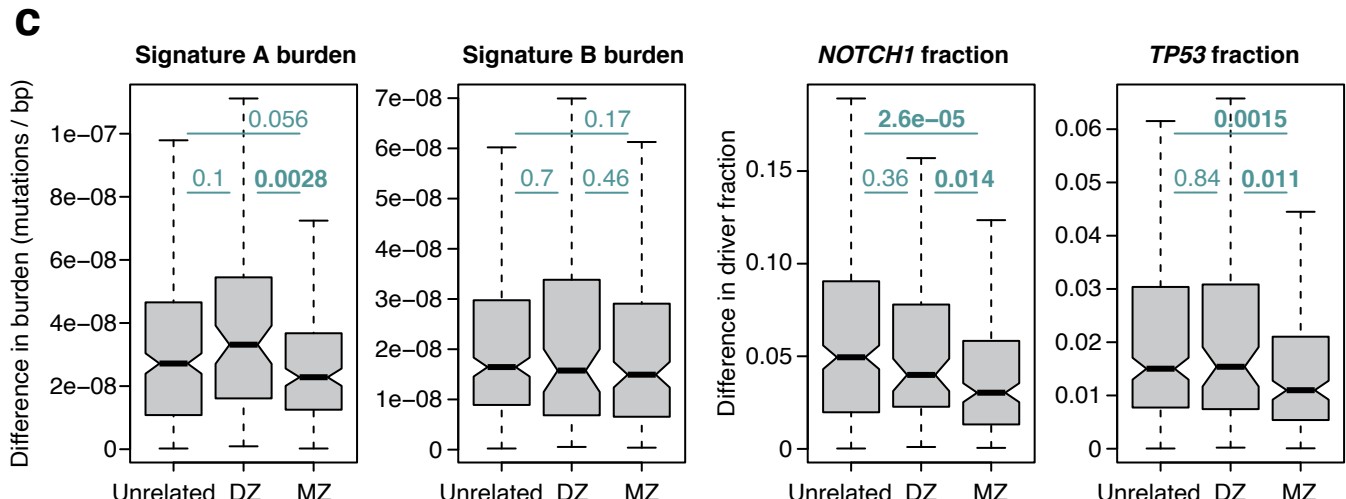

**Extended Data Fig. 10 | Functional impact of mutational signatures and heritability. a**, Fraction of ATA > ACA or ATT > ACT SNVs in samples presenting high (top 5%, red) or low (bottom 5%, blue) exposure to Signature B (SigB). Fractions are presented for non-coding SNVs, coding SNVs in all genes, and coding SNVs in top driver genes (defined as genes with >1,000 driver mutations across mutational classes with >80% estimated driver fraction). Error bars denote 95% CIs. **b**, Distribution of predicted mutation consequences for substitutions in any trinucleotide context (dark grey) and for ATA > ACA or

ATT > ACT substitutions only (light grey). **c**, Distributions of pairwise differences in generalised linear model residuals for Signature A and B mutation burdens and *NOTCH1* and *TP53* driver fractions, between pairs of unrelated (age-matched) individuals (n = 305), pairs of dizygotic (DZ) twins (n = 104), and pairs of monozygotic (MZ) twins (n = 211). The *P*-values for significantly different pairs of distributions (two-sided Mann–Whitney–Wilcoxon tests) are highlighted in bold. Box plots show the interquartile range, median, and 95% confidence interval for the median.

# Reporting Summary

## Statistics

For all statistical analyses, confirm that the following items are present in the figure legend, table legend, main text, or Methods section.

| n/a | Confirmed | |
|---|---|---|
| ☐ | ☒ | The exact sample size (*n*) for each experimental group/condition, given as a discrete number and unit of measurement |
| ☐ | ☒ | A statement on whether measurements were taken from distinct samples or whether the same sample was measured repeatedly |
| ☐ | ☒ | The statistical test(s) used AND whether they are one- or two-sided <br> *Only common tests should be described solely by name; describe more complex techniques in the Methods section.* |
| ☐ | ☒ | A description of all covariates tested |
| ☐ | ☒ | A description of any assumptions or corrections, such as tests of normality and adjustment for multiple comparisons |
| ☐ | ☒ | A full description of the statistical parameters including central tendency (e.g. means) or other basic estimates (e.g. regression coefficient) AND variation (e.g. standard deviation) or associated estimates of uncertainty (e.g. confidence intervals) |
| ☐ | ☒ | For null hypothesis testing, the test statistic (e.g. *F*, *t*, *r*) with confidence intervals, effect sizes, degrees of freedom and *P* value noted <br> *Give P values as exact values whenever suitable.* |
| ☒ | ☐ | For Bayesian analysis, information on the choice of priors and Markov chain Monte Carlo settings |
| ☐ | ☒ | For hierarchical and complex designs, identification of the appropriate level for tests and full reporting of outcomes |
| ☐ | ☒ | Estimates of effect sizes (e.g. Cohen's *d*, Pearson's *r*), indicating how they were calculated |

*Our web collection on statistics for biologists contains articles on many of the points above.*

## Software and code

Policy information about availability of computer code

| Data collection | No software was used for data collection |
|---|---|
| Data analysis | Nanorate sequencing data analysis relied on the NanoSeq pipeline, available at: https://github.com/cancerit/NanoSeq, version 3.3.0. Targeted methylation analyses were done with R package EpiDish v2.20. Most analyses have been done with bespoke pipelines, provided as an accompanying R markdown in HTML format. Sigfit v_2.1.0 and v_2.0 for signature analysis. Our pipeline makes use of samtools v1.9, bcftools v1.9, bwa v0.7.5a-r405, bedtools v2.29.0, Kraken 2.1.2. |

For manuscripts utilizing custom algorithms or software that are central to the research but not yet described in published literature, software must be made available to editors and reviewers. We strongly encourage code deposition in a community repository (e.g. GitHub). See the Nature Portfolio guidelines for submitting code & software for further information.

# Data

Policy information about availability of data

All manuscripts must include a data availability statement. This statement should provide the following information, where applicable:

- Accession codes, unique identifiers, or web links for publicly available datasets
- A description of any restrictions on data availability
- For clinical datasets or third party data, please ensure that the statement adheres to our policy

Sequencing data has been deposited in EGA under accession numbers EGAD00001015618 (TwinsUK_TargetedNanoSeq_Buccal), EGAD00001015619 (TwinsUK_TargetedNanoSeq_Blood), EGAD00001015620 (TwinsUK_ExomeNanoSeq_Buccal), EGAD00001015621 (TwinsUK_RENanoSeq_Buccal), EGAD00001015622 (TwinsUK_TargetedEMSeq_Buccal), EGAD00001015623 (TwinsUK_TargetedEMSeq_Blood), and EGAD00001015624 (Sanger_NanoSeq_RandD). Data access for EGAD00001015618, EGAD00001015619, EGAD00001015620, EGAD00001015621, EGAD00001015622 and EGAD00001015623 is managed by TwinsUK (EGAC00001000274) (Supplementary Table 7). Supporting code can be found in https://github.com/cancerit/NanoSeq/, https://github.com/im3sanger/dndscv, and as an accompanying R HTML MarkDown file. Patient metadata is managed by TwinsUK. Anonymised mutational data is available in Supplementary Tables 8 and 9. All analyses have been done with human genome GRCh37.

# Research involving human participants, their data, or biological material

Policy information about studies with human participants or human data. See also policy information about sex, gender (identity/presentation), and sexual orientation and race, ethnicity and racism.

| | |
|---|---|
| Reporting on sex and gender | We only focused on sex, not gender, for regression analyses where we used it as a covariate. For several analyses of twin pairs, such as the heritability analyses, we only used same sex twins. |
| Reporting on race, ethnicity, or other socially relevant groupings | We don't make any report on ethnicities or other socially relevant groupings |
| Population characteristics | Our cohort includes 1,047 donors, mean age of 65.2 years, mostly healthy with some donors reporting previous history of cancer, as well as oral health, body mass index, and medication. |
| Recruitment | From a preselection of 4,800 donors, we invited 1,796 to participate based on several criteria, receiving buccal swabs from 1,236. From the final 1,042 donors in the study, we also selected 380 individuals with available material the the TwinsUK BioBank for sequencing of archival whole-blood DNA.<br><br>To increase our statistical power to study associations with exposures, risk factors and germline factors, we included all available donors of age 80 or higher (n=230), as many complete twin pairs as possible, smokers, individuals with obesity (BMI > 30), and individuals with available genome-wide genotyping information. We also favoured the selection of males and individuals of non-white ethnicity to reduce some of the demographic biases in the TwinsUK registry compared to the general population. To test for associations between the mutational landscape and medications or clinical histories, we favoured the inclusion of individuals with a history of cancer (including all donors with a history of oral cancer, n=12) or a self-reported treatment history including tamoxifen, immunosuppressants, metformin, aspirin or ibuprofen. 194 samples were excluded from analysis based on several sequencing quality metrics, leaving a total of 1,042 samples in the study. Exclusion criteria included: removal of contaminated samples with either human (n=17) or non-human (n=132) DNA, exclusion of samples with mean duplex coverage lower than 50 (n=79), and exclusion of swabs with genotyping information not matching the pre-existing genotyping information from TwinsUK (n=7) |
| Ethics oversight | The use of these samples was approved initially by the North West Research Ethics Committee (REC 19/NW/0187) and later renewed by the same committee (REC 24/NW/0106), and informed consent was provided by participants. |

Note that full information on the approval of the study protocol must also be provided in the manuscript.

# Field-specific reporting

Please select the one below that is the best fit for your research. If you are not sure, read the appropriate sections before making your selection.

☒ Life sciences    ☐ Behavioural & social sciences    ☐ Ecological, evolutionary & environmental sciences

For a reference copy of the document with all sections, see nature.com/documents/nr-reporting-summary-flat.pdf

# Life sciences study design

All studies must disclose on these points even when the disclosure is negative.

| | |
|---|---|
| Sample size | Information on sample sizes is provided for all analyses. We estimated that, with optical duplicate rates, sequencing coverage translates into effective coverage on a 1/20 ratio. The requested sequencing coverage was chosen based on the amount of DNA for each sample. We aimed at a cohort size of 1000 donors, ending up with a cohort of 1042. This cohort size was chosen based on budgetary considerations. |

| | |
|---|---|
| Data exclusions | 194 samples were excluded from analysis based on several sequencing quality metrics, leaving a total of 1,042 samples in the study. Exclusion criteria included: removal of contaminated samples with either human (n=17) or non-human (n=132) DNA, exclusion of samples with mean duplex coverage lower than 50 (n=79), and exclusion of swabs with genotyping information not matching the pre-existing genotyping information from TwinsUK (n=7). |
| Replication | We haven't conducted explicit replication as this was not applicable for an observational study |
| Randomization | No randomization was performed  as this was not applicable for an observational study |
| Blinding | No blinding was undertaken because this is an observational study. |

# Reporting for specific materials, systems and methods

We require information from authors about some types of materials, experimental systems and methods used in many studies. Here, indicate whether each material, system or method listed is relevant to your study. If you are not sure if a list item applies to your research, read the appropriate section before selecting a response.

## Materials & experimental systems

| n/a | Involved in the study |
|---|---|
| ☒ | ☐ Antibodies |
| ☒ | ☐ Eukaryotic cell lines |
| ☒ | ☐ Palaeontology and archaeology |
| ☒ | ☐ Animals and other organisms |
| ☒ | ☐ Clinical data |
| ☒ | ☐ Dual use research of concern |
| ☒ | ☐ Plants |

## Methods

| n/a | Involved in the study |
|---|---|
| ☒ | ☐ ChIP-seq |
| ☒ | ☐ Flow cytometry |
| ☒ | ☐ MRI-based neuroimaging |

## Plants

| | |
|---|---|
| Seed stocks | *Report on the source of all seed stocks or other plant material used. If applicable, state the seed stock centre and catalogue number. If plant specimens were collected from the field, describe the collection location, date and sampling procedures.* |
| Novel plant genotypes | *Describe the methods by which all novel plant genotypes were produced. This includes those generated by transgenic approaches, gene editing, chemical/radiation-based mutagenesis and hybridization. For transgenic lines, describe the transformation method, the number of independent lines analyzed and the generation upon which experiments were performed. For gene-edited lines, describe the editor used, the endogenous sequence targeted for editing, the targeting guide RNA sequence (if applicable) and how the editor was applied.* |
| Authentication | *Describe any authentication procedures for each seed stock used or novel genotype generated. Describe any experiments used to assess the effect of a mutation and, where applicable, how potential secondary effects (e.g. second site T-DNA insertions, mosaicism, off-target gene editing) were examined.* |

