## [Peer Review File · Nature]

Somatic mutation and selection at population scale

Corresponding Author: Dr Inigo Martincorena

Version 0:

Reviewer comments:

Referee #1

(Remarks to the Author)

Lawson et al report a refinement of Nano-seq method that allows for the assessment of ultra-low frequency somatic mutations across the entirety of the human genome. This refinement reduces the error-rate of the original protocol and allows for the detection of mutation frequencies of $\sim 2 \times 10^{-8}$, which they validate with neonatal blood and also demonstrate the ability to reach this level of detection on FFPE tissue, which is notable. The authors apply Nano-Seq to blood and/or buccal swabs from >1000 individuals. As is typical from this group, this is an exceptionally well executed study. From a technical standpoint, I have nothing really to add and am satisfied with the study's execution. However, while this is a very well-done study, I struggle to identify what sort of major new breakthrough knowledge is being reported here that hasn't, in some form or another, been reported previously, even from the same group. For example:

1) While the refinement and expansion of Nano-seq is a welcome improvement to the technique and will be helpful for the research community, it is not what I would consider a significant advancement. Enzymatic fragmentation has been used with duplex sequencing to reduce error and is part of a commercial product, so adding this approach to the ddBTP approach in Nano-Seq is a logical next step. I will commend the authors for working out a non-commercial protocol, however.

2) The detection of strong somatic selection in blood and oral epithelium is widely known and this report does not significantly add to the genes already detected in CHIP and aging esophagus/oral mucosa, respectively. Thus, the findings are largely confirmatory of what has been previously published...just with more data.

3) The linear increase in mutations with age has been reported previously by several groups.

4) While it's certainly the case that the scale of mutation detected can provide a high-resolution map of where they occur in the protein, the Risques lab has been plotting out these sorts of mappings using duplex sequencing data and compared them to COSMIC in other publications. These papers do not appear to be cited in this context. At the very least it is important to cite these publications and also how the current findings extend these prior ones. Independently, it's unclear that the authors have achieved true saturation. Is that really the case? Has every amino acid or nucleotide been changed? If not, then the use of the term saturation is not warranted.

5) The authors note mutational signatures that coincide with age, smoking, and alcohol use. These signatures are already known to occur in oral mucosa. No additional signatures were noted.

6) I'm unclear on what is meant by 'selectogenic'. In my mind, there are several possibilities: 1) a chemical directly causes clonal expansions because it causes mutations at sequence contexts that are important for genes that are strongly selected for 2) a chemical directly causes clonal expansions by inducing cell divisions and the only ones that really respond are the cells with prior existing mutations in genes that cause cells to be more likely to divide faster or something similar or 3) a chemical causes damage to surrounding cells and cells that are more likely to divide take over in a wound healing-like response. The authors don't really define what they mean, so it's tough to really evaluate the interpretation of their analysis.

The study does have several unique qualities. For example:

- 1) The scale of the study does set a new standard for what's possible in studying somatic mutations and duplex sequencing.
- 2) The size of the sample cohort and the large number of detected mutations allows for a novel epidemiological analysis of somatic mutations. The authors do provide evidence that suggests that the "intensity" of SBS5 is heritable, but the effect is modest and, by their own admission, there was not enough power to observe a significant genome-wide association or see other associations.
- 3) They also note patterns of mutations and clonal expansions are seemingly constrained in solid tissues, but seemingly not in blood.
- 4) Another novel finding is the evidence for negative selection of mutations in some genes in somatic tissue, which, while not entirely surprising, is an important and previously unreported finding.

I do want to be clear here. This work will be of significant interest to the research community and definitely deserves to be published in a high-impact journal, but I'm less sure if the novelty of the findings rises quite to the level of Nature.

(Remarks on code availability)

The file format was not given (.unk) and changing to the standard .zip, .gz, and .tar did not open them. A link to a software repository was not provided, as far as I could tell.

Referee #2

(Remarks to the Author)

Overall:

Lawson, Martincorena and colleagues describe an improved version of NanoSeq, error-corrected duplex single-molecule sequencing. Such methods aim to substantially improve our understanding in the genesis and evolution of somatic mutations, ideally ultimately toward novel insights in cancer and non-cancer conditions. In increasing set of observations have shown that somatic mutations may have non-cancer implications, further highlighting the importance of such technologies. However, these is an early report of the technology with some early insights. The writing is clear with some early scientific observations. I have some initial concerns limiting my overall enthusiasm of the manuscript in its present form.

Major:

1. The authors cite two other recently described single-molecule duplex sequencing methods with low error rates for the detection of somatic mutations – HiDEF-seq (Long MH et al Nature 2024) and CODEC (Bae JH et al Nat Genet 2023). However, it is unclear what the advantage would be of the present method over the other methods. Benchmarking with comparisons would significantly enhance this contribution.
2. With very little exception, bulk WES/WGS of blood to genotype CH-driver mutations has identified DNMT3A as the top gene with driver mutations and TET2 as the second top gene. The only exception that I'm aware of is targeted NGS in the CANTOS trial (coronary artery disease and hsCRP>2), where TET2 was the top gene. However, Fig 1g shows TET2 as the top gene and DNMT3A as the second top gene. This significant deviation from the expectation leads to concerns about bias related to ascertainment, reduced generalizability of the findings, and potential errors/biases in sequencing.
3. In the analysis of blood mutations, the authors feature that their yield for DNMT3A and TET2 driver mutations is much higher than standard-coverage WES/WGS and is thus a feature of their single-molecule duplex sequencing approach. However, targeted error-corrected deep bulk sequencing will also substantially increase the yield (Young AL et al Nat Comm 2016) so this might not be a compelling primary argument to feature. An argument could be that error rates would be lower with a single-molecule approach would be reasonable to add but I have technical concerns given the unexpected distribution of driver mutation genes (see major point 2 above).
4. Given the genome-wide mutational profiles, it would be neat to apply methods derived from bulk WGS (Weinstock J et al Nature 2023) to understand the dynamics and estimate the ages of driver mutations.
5. Since the authors show that positively selected mutations appear rather ubiquitous in polyclonal tissues, the clinical relevance of these very very subclinical findings is unclear. Further analysis of the data to infer early predictions of future clonality could help towards this goal. Nevertheless, the authors describe that while NOTCH1 driver mutations are common and have high VAFs in buccal tissue, risk of transformation (relative to others) is low. This indicates that models purely describing clonality/fitness may not necessarily translate to neoplastic risk.
6. The authors suggest that 28 of 49 genes from buccal swabs are under positive selection. Replication in another dataset or validation in an experimental model would increase confidence in this potential novel finding. The claim that ~62,000 mutations are anticipated to be driver mutations seems potentially substantial, and would benefit from a validation of a subset of novel observed mutations to increase confidence that this is a suitable claim to make.
7. Based on Fig 2g, individuals with NOTCH1 driver mutations uniquely had much higher VAFs. Is there histopathology that such individuals already had dysplasia or neoplasia?
8. These authors (Martincorena et al Cell 2017) have done nice work in TCGA showing the absence of negative selection in cancer tissues. In the present work, of noncancerous tissues, they now show that there is some negative selection of somatic mutations. Is this a contextual difference between cancer and non-cancer? And given the very small number of genes identified, perhaps it's still safe to say that negative selection is significantly less common for somatic versus germline mutations?
9. Using prior germline genetic determinants of CH based on population-based WES/WGS studies (Kar SP et al Nat Genet 2022, Kessler MD et al Nature 2022), what are the mutational burdens in driver and non-driver genes for those with increasingly greater polygenic predisposition to CH? Might this give more insights into more clinically relevant clonality?
10. This metric on line 369 ("Notably, these regressions suggest that one additional year of life causes as many mutations in

the oral epithelium as ~2.8 pack-years or ~19.1 drink-years (see Supplementary Note 7 for caveats and interpretation”) is provocative. It would benefit from confidence intervals and a clarification of the age range where this is valid in the main text.

11. The inference that cancer is arising more from mutagenesis than selectogenesis does not appear expected. As the authors show and others have shown, driver mutations appear to be quite common without expansion, and others have suggested that many of these mutations have happened very early in life (Mitchell E et al Nature 2022, Fabre MA et al Nature 2022). So one might infer changes in selection across the life course may ‘unlock’ the potential of these quiescent driver mutations. Additional clarification would be helpful.

12. The authors make several claims about exposures causing various genomic features. With cross-sectional data, one must be cautious about these claims as there are likely numerous potential confounders correlated with tobacco smoking, alcohol, etc.

13. The authors note that they found no genome-wide signals for their GWAS of mutation rates or driver densities. Given the very small sample size this is not surprising. A power assessment accompanying this result would be expected.

Minor:

1. There is an open parenthesis on line 31 without a closed parenthesis in the sentence.
2. “In vivo” on line 45, “in vitro” on line 61, etc, should italicized per convention.
3. Fig 1e shows a linear association with age and mutation burden. Is this the same for driver mutations? For VAF>2% driver mutations in blood, there is a logarithmic relationship with age.
4. In the dataset with buccal swabs, 37% were smokers. This should be sufficiently powered to compare genic selection between smokers and non-smokers indicating selectogenicity effects.
5. Based on Fig 2d, it appears that most with buccal swabs had multiple nonsynonymous mutations in known driver genes. Subsetting to those inferred to be driver mutations would provide further clarity.
6. Natural premature menopause (i.e., premature ovarian failure), but not surgical premature menopause, has been associated with a greater prevalence of CH based on WES/WGS. If age of menopause is available, then it would be interesting to see if mutational burden is increased among women with natural premature menopause. The authors note that overall mutational burden was not different by sex but there may be a difference with this subgroup.

(Remarks on code availability)

Referee #3

(Remarks to the Author)

Lawson et al performed single-molecule sequencing to describe the landscape of somatic mutations in over 1,000 individuals of oral epithelium and 371 blood samples. They reported a new version of NanoSeq, which improve the whole-genome coverage to allow genome-wide, somatic mutation profiling in normal tissues. They reported additional genes and mutations under selection of clonal expansion through this large cohort. They also claimed to identify more somatic mutations in the blood given their ability to detect ultra-low VAF mutations from single molecule sequencing. The strengthen of the study is generating a large cohort of sequencing of normal oral tissue, and the analyses cover broad topics, from detecting new drivers to linking mutational signature with risk factors to infer somatic heritability. However, the novelty of the study is not clearly described, with some results are over interpreted and some conclusions require further investigation.

Major issues

1. The first part of the paper, developing a genome-wide single-molecule mutation detection protocol, is an extension of Abascal et. al. 2021 study, with increased whole-genome coverage compared to the old version of NanoSeq. However, Figure 1 was not focused on comparing whole-genome coverage between the old version and new version of NanoSeq, but instead, focused on comparing mutation detection accuracy of new NanoSeq to standard duplex sequencing, which was largely demonstrated in the 2021 paper using old NanoSeq. Moreover, in the second part of the study, the analyses were mostly based on targeted NanoSeq sequencing, so the advantage of whole-genome NanoSeq is very unclear. Similarly, they showed that NanoSeq can accurately detect low VAF mutations from FFPE samples, but the advantage appears coming from using ddBTPs, which was already in the old version. Together, the first part does not read as reporting a new technology, but showing additional benchmarking of Nanoseq, which was mostly reported in the 2021 study, with minimal improvement on the genome-wide coverage. This makes the novelty of new NanoSeq very confusing to the readers.
2. The strength of the paper is oral epithelium bulk sequencing at scale to detect ultra low VAF mutations and understand their selection, but the highlights of the main findings are mostly confirming the role of NOTCH1 in clonal expanding and TP53 in driving tumor development, which is known (PMID: 36658434; PMID: 36266286). The study should clarify the novel findings.
In addition, the novelty of the study might be increased if the authors can dive into some new pathogenic variants identified through selection, and using cancer data to provide insights into their functional impacts.
3. The authors try to investigate the role of environment-induced mutagenesis vs selection on increased cancer risk. However, it is unclear if the current cohort is sufficient to answer the question. Drivers with increased VAF can be difficult to detect from sequencing solid tissue due to sampling, which may explain the results in Fig4a, where large clones can be more easily detected from blood sample. To suggest stronger effect of mutation burden than selection, multi-sampling or comparing normal vs premalignant lesions are needed.
4. Although the concept that linking somatic mutations in normal tissue with epidemiological factors to improve cancer risk prediction is interesting, the study did not clearly show, for example, how knowing the clonal landscape related to smoking and alcohol use can improve the prediction beyond having smoking and alcohol drinking data. Moreover, from the clinical perspective, the advantage of clonal profiling using NanoSeq, compared to existing cancer screening approaches, is not

clearly demonstrated, perhaps given limited clinical information of the study cohort. However, as the investigators did not build any cancer risk prediction model, the clinical utility and significance should be further clarified in the text.

Minor concerns:

1. Given age differences between cohorts and its importance in mutations, age should be adjusted in several comparisons between their cohort and other cohorts.
2. Some references were missing, for example, for the paper in line 138-141.

(Remarks on code availability)

Referee #4

(Remarks to the Author)

In this manuscript, Lawson et al report an upgraded version of NanoSeq, a duplex sequencing method with an ultra-low error rate and full genome coverage, making it a powerful tool for studying somatic mutation and selection in normal tissues, particularly polyclonal tissues. The authors applied this method to buccal swab samples (oral epithelium) collected from a large cohort of twin participants, which reveals a rich landscape of somatic mutation and selection in human normal tissues at epidemiological scale for the first time.

The study sets up a new paradigm for the future research on somatic mutagenesis in normal tissues. It is impressive, novel and impactful in several key aspects: (1) Previous studies of somatic mutation in solid normal tissues were limited to small cohorts, limiting insights into mutation selection across diverse populations. The current study for the first time pushed the boundary to epidemiological scale and reveal groundbreaking findings on the influence of germline and environmental factors on somatic mutation and selection. This advance was enabled by NanoSeq, which allows for accurate detection of low-frequency somatic mutations in polyclonal tissues at a lower cost and without labor-intensive processes. Besides, buccal swab collection is non-invasive and logistically feasible, allowing for the recruitment of a large cohort without significant burden on participants. (2) It is a herculean effort to recruit the cohort, systematically collect data and metadata, and conduct comprehensive and rigorous analyses. Perhaps, this is not surprising, as it is consistent with the high-quality standards of publications from this group. (3) Most importantly, the study reports many novel findings and has conceptually advanced our knowledge of somatic mutation and selection in normal tissues. For example, negative selection was found to be weak or missing in previous studies. The current study leveraged the incredibly high mutation density across each gene and identified significant negative selections on 9 genes. (4) In addition, the manuscript is well-structured and well-written, equipped with incredibly detailed supplementary notes.

I believe the following points should be addressed and clarified before publication.

In the buccal swab cohort, 37% participants are smokers. Given this, why was SBS4 not detected in the mutational signature analysis? It is associated with tobacco smoking and commonly observed in oral cancer. Additionally, why did the authors choose to use Sigfit instead of other widely used methods, such as SigProfilerExtractor, for the mutational signature analysis?

In the regression analysis, sex was not significantly associated with differences in mutation rates, signatures or driver densities when correcting for confounders. Can the authors discuss or speculate on this finding, especially given the known sex bias in oral cancer?

Lines 162-163: the authors performed whole-genome NanoSeq on 16 samples to the genome-wide mutation rate. Why did they use the original restriction enzyme protocol instead of the new protocols (according to the description in Supplementary Note 3). In general, I feel the nomenclature could be clearer, especially when describing the comparison between previous and current versions of Nanoseq.

When estimating mutational burdens, has the authors considered the fraction of cells that carry each mutation (by summing VAFs)?

Line 85: can the full-genome duplex coverage be shown and compared with the previous NanoSeq version in a plot?

(Remarks on code availability)

Version 1:

Reviewer comments:

Referee #1

(Remarks to the Author)

I am satisfied with the revisions and commend the authors on a well executed study.

(Remarks on code availability)

Referee #2

(Remarks to the Author)

I do not have any further comments and believe that the authors have sufficiently addressed my concerns.

(Remarks on code availability)

Referee #3

(Remarks to the Author)

The authors fully addressed my comments. Novelty and technical advances are clarified in the text. Additional analysis in mutation rate vs selection strengthened the paper. Statistical tests are appropriate and clearly described in the text and figures.

(Remarks on code availability)

Referee #4

(Remarks to the Author)

The authors have addressed all my comments and I look forward to seeing this paper published.

(Remarks on code availability)

Referee expertise:

Referee #1: duplex sequencing

Referee #2: genetic epidemiology, somatic evolution

Referee #3: cancer genomics, epidemiology

Referee #4: cancer genomics, clonal evolution

Referee #1 (Remarks to the Author):

Lawson et al report a refinement of Nano-seq method that allows for the assessment of ultra-low frequency somatic mutations across the entirety of the human genome. This refinement reduces the error-rate of the original protocol and allows for the detection of mutation frequencies of $\sim 2 \times 10^{-8}$, which they validate with neonatal blood and also demonstrate the ability to reach this level of detection on FFPE tissue, which is notable. The authors apply Nano-Seq to blood and/or buccal swabs from >1000 individuals. As is typical from this group, this is an exceptionally well executed study. From a technical standpoint, I have nothing really to add and am satisfied with the study's execution. However, while this is a very well-done study, I struggle to identify what sort of major new breakthrough knowledge is being reported here that hasn't, in some form or another, been reported previously, even from the same group.

Authors' response: We thank the reviewer for the positive comment on the technical execution of the manuscript and for the constructive criticism on novelty. A challenge for us when writing this manuscript was to summarise a large amount of data, new analyses and new concepts within the space constraints. This included describing the new method, the study design, the driver discovery and positive selection results, the negative selection analyses, the high-resolution maps of selection within genes, the non-coding drivers identified, the VUS annotation, and the mutational epidemiology analyses (including the concept and analyses around selectogenesis, and the heritability analyses). As a result, in each section we focused on the most statistically significant results, leaving a lot of material for the supplementary text. In retrospect, this may have downplayed the novelty of the dataset. For example, in the driver discovery main text section we focused the attention on the top few drivers and on the similarities with skin and oesophagus, and failed to emphasise the novelty of the results (e.g. the vast majority of the ~ 50 drivers identified have never been reported under selection in normal tissues).

While we are still constrained by space, in the revision we have better clarified the novelty of the paper and expanded on several results. A short summary of key novel findings in this paper is:

1. First description of the mutation landscape of oral epithelium, including the first estimates of the mutation rate and mutation signatures of oral epithelium, and >30 driver genes not previously described under selection in any normal tissue.
2. Two new NanoSeq protocols with full genome coverage and error rates $< 5 \times 10^{-9}$ errors/bp. To our knowledge, this is the only sequencing protocol with full genome coverage and error rates two orders of magnitude below the mutation load of adult human cells, which in turn enables accurate measurement of somatic mutation rates, signatures and drivers on any human cell type. Although the modifications of the 2021 NanoSeq protocol may appear small, the new fragmentation protocols required extensive R&D over 2 years (see more details below). And, importantly, these protocols should be compatible with other error-corrected protocols beyond duplex sequencing (e.g. CODEC, HiDEF-seq, SMM-seq), which makes these advances of wider relevance to the error-corrected literature.
3. Largest collection of driver mutations in a single study. This includes >20,000 mutations in *NOTCH1* and >8,000 in *TP53*, with 24 driver genes having >1,000 mutations per gene. This study alone has yielded $\sim 3-4$ times more driver mutations than all $\sim 10,000$ cancer exomes in TCGA across more than 20 cancer types put together. The unprecedented number of driver

mutations in turn enables high-resolution analyses of selection within key genes. To achieve this we also introduce new dN/dS algorithms in this paper, leading to the discovery of 1,220 amino acid changes under positive selection, and of new non-coding and synonymous driver sites in key driver genes. This in turn allowed us to inform on the clinical relevance of many variants of uncertain significance (VUS) on key genes, including in *TP53*. In the revised manuscript, we have expanded on these results and clarified their novelty.

4. Convincing evidence of negative selection on several genes, which has been difficult to detect in smaller datasets due to insufficient statistical power.
5. Novel insights into the dynamics of clonal growth in a solid tissue, surprisingly supporting a model of highly constrained clonal growth (even at low clonal densities), where the average clone size does not grow with age. This includes novel analytic approaches for inferring clonal dynamics from aggregated duplex data. We consider this of significant importance and novelty.
6. Detailed multivariate regression models to establish associations between cancer risk factors and changes in mutation rates, mutation signatures and selection. Interesting and novel results include: (1) the extent of alcohol-induced mutagenesis in the oral epithelium, (2) an unexpected mode of action for tobacco-induced mutagenesis in the mouth (not through SBS4 or SBS92 signatures, but through an unexpected acceleration of SBS5 and SBS16), and (3) evidence of selectogenic effects (including stronger selection on *NOTCH1* or *TP53* in smokers). Some negative results are also of high relevance. For example, we see no difference in somatic mutation rates with sex or BMI, despite being very well powered to find these associations. In the revision, we have emphasised the novelty of this section, for which we thank the reviewer.

We expand on some of these points below, in our point-by-point response.

1) While the refinement and expansion of Nano-seq is a welcome improvement to the technique and will be helpful for the research community, it is not what I would consider a significant advancement. Enzymatic fragmentation has been used with duplex sequencing to reduce error and is part of a commercial product, so adding this approach to the ddBTP approach in Nano-Seq is a logical next step. I will commend the authors for working out a non-commercial protocol, however.

Authors' response: We thank the reviewer for this comment, which has motivated us to better describe the extent of R&D work that was required to achieve error rates $<5e-9$ errors/bp with whole genome coverage.

In the manuscript, we introduced two versions of NanoSeq with full genome representation, one using sonication and exonuclease blunting, and another using random enzymatic fragmentation. Both required extensive development over 2 years. The text below summarises some of this work, which we also now describe more clearly in the revised **Supplementary Note 1**.

R&D on the sonication and mung bean protocol. Our first implementations of this protocol suffered from very low library yields, ~10-100 fold lower than the original restriction enzyme protocol (2021). Optimising library yields while retaining the $<5e-9$ error rate required extensive testing and optimisation, including testing error rates in different sonication buffers for frozen and formalin-fixed DNA, different exonucleases for blunting, different enzyme concentrations, different adapter concentrations, and different DNA ligases and ligation conditions. After considerable optimisation we arrived at the protocol described in the **Supplementary Note 1**, which now has similar (slightly higher) library conversion efficiencies to the 2021 restriction enzyme protocol while providing full genome coverage and error rates $<5e-9$ errors/bp.

R&D on the enzymatic fragmentation protocol. To further increase library yields, we then developed an enzymatic fragmentation version of NanoSeq. As the reviewer notes, enzymatic fragmentation is

already used in some commercial kits for duplex sequencing, such as TwinStrand's. However, achieving NanoSeq-like error rates (on both undamaged and damaged DNA) using enzymatic fragmentation required extensive optimisation, and it was not simply achieved by the addition of ddBTPs to a commercial enzyme mix (as suggested by the reviewer). We first tested a commonly used enzymatic fragmentation kit called NEBNext Ultra II FS, which resulted in very high error rates around $1e-5$ errors/bp, suggestive of the presence of polymerase activity in the fragmentation mix. We then tested a new NEB enzymatic fragmentation mix called NEBNext UltraShear. Using UltraShear followed by standard end repair (e.g. Ultra II) resulted in too high error rates (as shown in **Fig. 1**). We thus tested a number of alternative solutions, including removing end repair altogether (relying on pre-existing blunt ends for A-tailing and ligation), including or excluding Mung Bean to increase the blunting of overhangs, testing error rates with and without SPRI clean-up of the UltraShear buffer, and testing UltraShear in alternative fragmentation buffers to minimise any undesired copying of errors between strands. This required multiple rounds of testing and sequencing. We concluded that UltraShear in UltraShear buffer without additional end repair, followed by SPRI cleanup of the UltraShear buffer (with or without mung bean nuclease) and A-tailing in the presence of ddBTPs greatly reduced error rates in frozen cord blood DNA, but still led to higher than optimal error rates. This was particularly visible in the formalin-damaged DNA, where an error rate on the order of $\sim 2e-8$ errors/bp and error-derived strand asymmetries were seen. We worked collaboratively with NEB to replace the UltraShear buffer with a suitable alternative. NEB's r1.1 buffer, which finally removed the low-level transfer of errors between strands, achieving NanoSeq-like error rates on both frozen and our formalin-damaged DNA. This high accuracy came at a cost of increased strand dropout (loss of one strand due to ssDNA breaks). After further R&D, this was solved by the incorporation of NAD⁺ in the r1.1 buffer, leading to the enzymatic fragmentation protocol that we describe in the manuscript.

R&D on library quantification, bottleneck size and bait capture. Additional R&D was required to optimise the library quantification and bottleneck size. These improvements were essential to enable our study of >1,400 samples (including buccal and blood samples). Although we introduced the qPCR and bottleneck steps in the 2021 protocol, new improvements in the current manuscript include: (1) improved equations for modelling sequencing efficiency (described in **Supplementary Note 1**), which help ensure optimal duplicate rates on any input DNA and gene panel (incl exomes), and (2) R&D to optimise the amount of amplified library per fmol of original library to be inputted into bait capture. This last point is important as enough copies of each original molecule need to be used for bait capture to reliably obtain sequencing data from both strands of DNA. By optimising this ratio, we maximised our ability to multiplex libraries into hybrid capture reactions, hence reducing the overall cost of large cohort projects.

Action: We have expanded the description of the R&D work that was required to develop two protocols with whole-genome representation and error rates $<5e-9$ errors/bp (see the revised **Supplementary Note 1**). We hope that these additional details better reflect the challenges overcome and the novelty of the protocols. These details may also help users interested in further optimising these or related protocols.

Importantly, in terms of novelty and technical advance, we note that the two new protocols should be compatible with other error-corrected sequencing methods that rely on sequencing both strands of DNA (e.g. CODEC, HiDEF-seq, SMM-seq), and so we hope that these advances have wider importance in the error-corrected sequencing literature even outside of duplex sequencing. For example, CODEC (Bae et al., *Nature Genetics*, 2023, PMID:37106072) and HiDEF-seq (Hong Liu et al., *Nature*, 2024, PMID:38867045) already used some of the improvements in the 2021 NanoSeq paper to achieve lower

error rates (e.g. restriction enzymes and/or ddBTPs). We expect that the new clean fragmentation protocols introduced here could be readily adopted by these and other methods to achieve lower error rates with full genome coverage. This is also made clearer in the revised manuscript.

2) The detection of strong somatic selection in blood and oral epithelium is widely known and this report does not significantly add to the genes already detected in CHIP and aging esophagus/oral mucosa, respectively. Thus, the findings are largely confirmatory of what has been previously published...just with more data.

Authors' response: We think that this is not entirely correct.

The driver landscape in blood is well known, and for this reason we use it in the manuscript as a positive control of the technology, rather than as a novel result. However, the driver landscape of oral epithelium is, to our knowledge, novel in several important ways:

1. To our knowledge, this is the first description of the mutation landscape of oral epithelium. The landscape of oesophageal epithelium has been described (by us in 2018 and by others), but oral epithelium and oesophageal epithelium are not the same tissue. Some shared biology was expected, but the extent of their similarities and differences was unknown. For example, oral epithelium is ectoderm-derived whereas oesophageal epithelium is endoderm-derived, and the squamous cancers that they generate are similar but not identical, in terms of clinical presentation, aetiology and genomics. Although in the original manuscript we focused on the similarities between the oral landscape and the oesophageal landscape (which may have downplayed the novelty of the results), there are also notable differences. For example: (1) the driver density is remarkably higher in oesophagus (e.g. *NOTCH1* clones occupy over 30-40% of the ageing epithelium compared to ~6-12% in the oral epithelium), and (2) the strength of selection in some genes varies starkly (e.g. *KMT2D*, *NFE2L2* and *PIK3CA* are strong drivers in normal oesophagus but neutral or only very weakly selected in oral epithelium).
2. Importantly, in the current manuscript we report 46 driver genes in buccal swabs. Of these, 31 have not been described in skin or oesophageal epithelium, and the majority of them have not been described under selection in any normal tissue, to the best of our knowledge. Hence, we respectfully disagree with the statement that "*this report does not significantly add to the genes already detected in CHIP and aging esophagus/oral mucosa*". In fact, this may be the largest addition of new genes under selection in normal tissues from a single study to date. These 31 genes are also clinically relevant, as they include multiple drivers of oral and head and neck cancers, which now emerge as being under selection in normal oral epithelium (e.g. *RHOA*, *RAC1*, *EPHA2*, *ZNF750*, *HLA-B*, *EP300* and *KDM6A* -all of which are significant in TCGA HNSC tumours by dNdScv-).
3. The absence of key drivers is an equally important result, which we failed to adequately highlight in the submission. An intriguing observation in previous studies on oesophagus was that some of the most common driver genes in oesophageal cancers were not detected as significantly mutated in normal oesophagus. However, whether this was simply due to insufficient statistical power was unclear given the small size of previous datasets (including ours). The very high depth of the current study provides a much clearer distinction between the selection landscape of normal oral epithelium and oral or head-and-neck cancers. For example, we find that mutations in *CDKN2A*, *NFE2L2*, *PTEN*, *HLA-A*, *B2M*, *SMAD4* and *RBI*, among others, are remarkably neutral (or very weakly selected) in oral epithelium despite being common drivers in HNSC. This suggests that selection on these genes is likely a later event in HNSC development, potentially in combination with other mutations. It is interesting that this list includes genes like *HLA-A* and *B2M*, which are believed to be immune escape genes, and so may be expected to be selected later in carcinogenesis, and not as first hits. Also interestingly, *CDKN2A* loss (which may enable growing clones to escape cellular senescence) and *SMAD4* loss are also known as relatively late events in colorectal, pancreatic and oesophageal adenocarcinoma evolution, rather than as first hits (Weaver et al. *Nat Gen*, 2014,

PMID:24952744; Papageorgis et al, *Cancer Res*, 2011, PMID:21245094; Cowan and Maitra, *Cancer J.*, 2014, PMID:24445769). Thus, the high depth of our study not only has identified many novel drivers in normal epithelium, but it provides suggestive information on whether key cancer drivers may act as first or later hits.

Action: We thank the reviewer for this constructive comment, which has motivated us to improve this section of the manuscript and to better describe the novelty of the driver discovery results. In the revised manuscript, we now describe some of the key differences between the oral mutation landscape and the oesophageal landscape better in the main text, and in greater detail in **Supplementary Note 4**. We have also added a new analysis and figure (**Extended Data Fig. 5e**, reproduced below) highlighting the differential selection of some genes in HNSC vs oral epithelium.

3) The linear increase in mutations with age has been reported previously by several groups.

Authors' response: We acknowledge that the linear accumulation of mutations with age has been reported for several tissues before by several groups (including ours). However, the mutation rate (i.e. the slope), the mutational spectrum, and the variation in mutation signatures across donors in oral epithelium were not previously known. Importantly, we also provide detailed and novel information on the extent to which mutation rates in oral epithelium vary across individuals as a result of different exposures, which is important to understand the mode of action of different risk factors and carcinogens.

4) While it's certainly the case that the scale of mutation detected can provide a high-resolution map of where they occur in the protein, the Risques lab has been plotting out these sorts of mappings using duplex sequencing data and compared them to COSMIC in other publications. These papers do not appear to be cited in this context. At the very least it is important to cite these publications and also how the current findings extend these prior ones.

Authors' response: We thank the reviewer for this comment.

We apologise for not citing these papers in this section. We had planned to include several more references to Rosana Risques's papers, and we only realised the oversight the day after submission. In fact, we emailed Rosana at the time to let her know of this oversight, which we planned to correct during revision. We confirm that we have corrected this now.

The reviewer is correct that maps of the distribution of mutations along some driver genes have already been shown in previous papers, including by us (e.g. Martincorena et al, *Science*, 2015, PMID: 25999502, and Martincorena et al, *Science*, 2018, PMID:30337457), by the Risques lab (e.g. Salk et al, *Cell Rep.*, 2019, PMID:31269435; Tee et al, *Gynecol Oncol*, 2024, PMID:39128337), and by others. However, it is important to emphasise that the size of the current dataset is far larger than any of these previous studies, both in its aggregate depth (~700,000 dx) and in the breadth of genes studied. This is critical, as it has enabled us to perform multiple novel analyses, yielding new insights. These include:

1. Detection of significant positive selection on single-sites, revealing over 1,200 individual amino acid changes under positive selection.
2. Demonstration of the ability to classify variants of uncertain significance (VUS) using site-level dN/dS.
3. Much more exhaustive discovery of coding and non-coding driver sites, leading to the discovery of previously unknown synonymous and noncoding cis-regulatory driver sites.

We kindly refer the reviewer to **Extended Data Fig. 6l**, which shows a saturation analysis revealing how only a small minority of the sites under selection discovered in the current study would have been found in smaller datasets.

To enable these analyses, we have introduced new dN/dS functions to study selection at single sites (including non-coding sites) and on subsets of functionally-related sites within a gene (freely defined by the user). These new algorithms (including the new *withingenednds* function and one-sided dNdScv tests for positive and negative selection), are made publicly available through the dNdScv R package, and are also a novel contribution of this study.

Action: We thank the reviewer for this comment. We hope that the explanations above clarify the importance (and novelty) of a dataset of this size. In the revision, we have tried to make this clearer in the main text. We have also added more references to Dr Risques's papers, including in the section pointed out by the reviewer. And we have greatly expanded **Supplementary Note 4**, providing many more details on the driver genes and driver sites discovered in the study. We now also point to this section more prominently in the main text.

For clarity, examples of novel findings using the depth of sequencing and the new methods include:

1. Discovery of >1,200 amino acid changes under significant positive selection
2. Demonstration of how significant recurrence of somatic mutations in normal tissues can aid annotation of variants of unknown significance
3. Identification of 5 synonymous driver mutations in *NOTCH1*, which all likely act via creation or disruption of splice sites
4. Discovery of non-coding driver mutations, including in *TP53* (polyadenylation signal, splice site of first non-coding exon, insertions and deletions near the transcription start site) and *TERT* (non-canonical promoter mutations)
5. Gene-specific insights on the structural impact of mutations (e.g. novel RAC1 hotspots clustering around the GTP/GDP binding pocket, ATM missense mutations affecting key residues that maintain the autoinhibitory conformation of the kinase domain and CHEK2 missense mutations clustering on one surface of the forkhead-associated domain (**new Extended Data Fig. 6j**))

Independently, it's unclear that the authors have achieved true saturation. Is that really the case? Has every amino acid or nucleotide been changed? If not, then the use of the term saturation is not warranted.

Authors' response: This is an interesting question. First, we would like to clarify that we did not intend to claim to have achieved saturation in this study, we only wanted to say that deep single-molecule sequencing of normal tissues can be used to achieve a form of in vivo saturation mutagenesis. This is because the broad mutational spectrum of SBS5 can mutate every possible site and because an almost limitless number of selected clones can be assayed cost-efficiently in highly polyclonal tissues using

accurate single-molecule sequencing, particularly using non-invasive or readily available samples. We already acknowledged this in the main text to avoid confusion: “Although deeper sequencing will be required to achieve true saturation (**Supplementary Note 4**), these results suggest that deep single-molecule sequencing of polyclonal tissues has the potential to provide *in vivo* saturation mutagenesis information for genes under clonal selection.”

Prior to writing this manuscript, we discussed the definition of saturation with three colleagues in the saturation mutagenesis field and they acknowledged that there are different definitions of saturation. To preempt this question, we discussed several relevant definitions in the section titled “Saturation analysis” at the end of the **Supplementary Note 4** (we paste this section below for clarity).

*As described in the main text, ultra-deep single-molecule sequencing of polyclonal samples has the potential to provide a form of *in vivo* saturation mutagenesis. In this section, we describe some supplementary analyses on the extent of saturation achieved in the current study.*

*First, a valuable metric can be the density of mutations per site at neutral sites. This value depends on the mutability of each site, which is particularly affected by the trinucleotide sequence context of each base. The mean neutral mutation rate per site for each trinucleotide substitution is calculated in the substitution model of dNdScv (`dndsout$mle_submodel`). In the buccal swab dataset, the highest average neutral mutation rate per site was ~0.43-0.60 mutations/site for C>T changes in all four possible CpG contexts, and the lowest rates per site were ~0.007-0.008 mutations/site for A>C mutations at certain contexts (**Supplementary Code**). The mean rate across all 192 possible trinucleotide changes was ~0.056 mutations/site. These rates refer to the neutral mutation rate for each possible trinucleotide change. When considering SNVs, each base can change to three other bases (e.g. A can change to C, G or T), and each codon can change to nine other codons, and so the average neutral mutation density per base pair or per codon will increase accordingly. This analysis reveals that ~2 or ~6 times higher aggregate depth than currently achieved will be required to obtain an average of one mutation per neutral codon or base pair, respectively.*

*The description above refers to the neutral mutation density per base change, per base pair, or per codon (or amino acid). However, we note that lower aggregate depths will be needed to find the most important sites under strong positive selection (e.g. with $\text{site-dN/dS} > 100$), while much higher depths will be required to find individual sites under negative selection. To explore the extent to which the landscape of driver mutations is approaching saturation in our dataset, we carried out a downsampling exercise, studying the number of genes and sites under significant positive selection for progressively larger random subsets of our dataset (**Extended Data Fig. 6k,l**). At gene level, this analysis suggests that a dataset half the size of the current dataset is sufficient to find ~80% of the 49 driver genes reported here, and that these genes account for ~90% of all non-synonymous mutations in driver genes. At single-site level (`sitednds`), the pattern of saturation varies considerably across genes. For example, the discovery of sites under positive selection in *RAC1* and *PPM1D* shows clear signs of saturation, where a dataset half the size of the current dataset is enough to find the hotspots responsible for >80% of the mutations at significant hotspots in the full dataset. A trend towards saturation is apparent but slower for *NOTCH1* and *TP53*, suggesting that larger datasets will identify additional relevant sites under positive selection. In contrast, other genes under weaker selection and where selection is spread across many sites, such as *CHEK2*, show no clear trend of saturation in subsamples of the current dataset.*

Overall, the current dataset represents an in depth description of the landscape of driver genes and driver sites in the oral epithelium, but larger studies are expected to continue to yield new sites under positive selection, particularly in genes under weaker positive selection. We also note that much larger datasets will be needed to comprehensively study the pattern of negative selection at the level of single genes, and particularly at the level of single sites.

As explained in this section, to complement the definitions of saturation, we performed a downsampling analysis of our dataset similar to Lawrence et al, *Nature*, 2014 PMID:24390350 (shown in **Extended Data Fig. 6k,l**). This analysis shows, as expected, that the discovery of sites under selection saturates at different rates for different genes, since detection sensitivity depends on the strength and direction of selection, as well as the site mutation rate and the duplex depth.

Beyond specific definitions, a major ambition of in vitro saturation mutagenesis experiments is to help annotate variants of uncertain significance for clinical diagnosis. Evidence of selection in cancer is starting to be used for the classification of pathogenic variants in some genes (e.g. Lai et al, *Genet Med.*, 2022, PMID:35997716) but is limited by the sparsity of cancer genomic datasets. In the current manuscript, we wanted to emphasise that deep sequencing of normal tissues has the potential to supersede cancer genomic efforts in generating large collections of driver mutations. We hope that our analyses in **Fig. 3h** exemplify how the detection of selection at single-site resolution using deep clonal scanning of normal tissues can help annotate many variants of uncertain significance (VUS) in clinically relevant genes, including ~86 VUS in *TP53* with evidence of significant positive selection in our dataset.

5) The authors note mutational signatures that coincide with age, smoking, and alcohol use. These signatures are already known to occur in oral mucosa. No additional signatures were noted.

Authors' response: As above, we note that the mutation rates and signatures in oral mucosa were not known before this study to the best of our knowledge. However, we acknowledge that in the original version of the manuscript we failed to clarify the novelty of our findings and we are grateful for the opportunity to correct this now.

Two important novel observations on mutational signatures in our study include:

1. Absence of APOBEC: although we may expect the signatures in oral epithelium to show similarities with those in oesophageal squamous epithelium or even transitional respiratory epithelium, the extent of these similarities was not known. In fact, although we see some similarities, we also see interesting differences. For example, in a small targeted NanoSeq study of ~20 oesophageal brushings we have detected frequent APOBEC mutagenesis in normal oesophagus, which contrasts with the absence of APOBEC in >1,000 buccal swabs. This is particularly relevant as APOBEC mutagenesis is common in head and neck cancers. Our observation suggests that APOBEC mutagenesis is rare in normal oral epithelium but is a common event later in HNSC development. This is, in fact, consistent with recent evidence from HNSC cancer genomes (Torrens et al, *medRxiv*, 2024, PMID:38699364).
2. Atypical effect of smoking: An important and novel result in our study is that tobacco smoking increases mutation rates in oral epithelium but not through the classical tobacco signatures (SBS4 and SBS92), but rather through an increase in SBS5 and an exacerbation of SBS16 in the context of alcohol consumption. This may be unexpected but it seems consistent with recent cancer genomics literature (Torrens *et al.*, *medRxiv*), which found SBS4 and SBS92 to be much less frequent in oral cancers than larynx cancers and not apparent when looking at oral cancer driver mutations. The apparent lack of SBS4 and SBS92 signatures in oral epithelium might be explained by two factors: (1) the stratified squamous epithelia may reduce the exposure of the basal stem cells, compared to the monolayered (pseudostratified) respiratory epithelium, and (2) the CYP1A1 enzyme, which is the main metaboliser of benzo(a)pyrene and required for SBS4 mutagenesis, is lowly expressed in oral epithelium compared to lung and larynx. Thus, our results suggest that smoking may cause oral cancer not through the typical SBS4 and SBS92 mutagenesis, but through an acceleration of SBS5 and SBS16, as well as additional selectogenic effects (**Fig. 4**).

Thus, the absence of key signatures, including APOBEC (SBS2/SBS13) and smoking signatures (SBS4/SBS92) is an important result, which we mentioned in passing but did not sufficiently explore and highlight in the original version of the manuscript. Convincingly demonstrating the absence of signatures, however, requires additional analyses, which we have added in the revision, as described below.

Action: We have revised the main text to better clarify the novel insights on mutational signatures. We have also added new analyses and figures. First, we have repeated the signature deconvolution analyses using SigProfiler instead of SigFit, obtaining analogous results. Second, since some of the most unexpected results refer to the absence (rather than the presence) of certain signatures, we have strengthened the analysis of the absence of APOBEC (SBS2+SBS13), smoking (SBS4 and SBS92) and oxidative damage signatures (SBS18, detected by Torrens et al, Medrxiv, 2024, in oral cancers) by using a supervised statistical test for their presence in every sample. To do so we use Likelihood Ratio Tests (which we introduced in Lawson et al, *Science*, 2020, PMID:33004514 and Cagan et al, *Nature*, 2022, PMID:35418684). These new and more sensitive analyses are described in **Supplementary Note 3** (and in our response to reviewer #4). These results are important as they provide novel and unexpected insights into the mode of action of tobacco carcinogenesis in the oral epithelium. Third, to further support the absence of classical smoking signatures (SBS4 and SBS92), we provide a new figure (new **Extended Data Fig. 8c**) with the aggregated mutation spectra for non-smokers vs heavy smokers, which emphasises the absence of SBS4 and SBS92 in smokers (pasted below). We also show the lack of the smoking-associated indel signature ID3 (**Extended Data Fig. 8e**). Additional details on the lack of smoking signatures are summarised in our response to reviewer #4.

New Extended Data Fig. 8c.

6) I'm unclear on what is meant by 'selectogenic'. In my mind, there are several possibilities: 1) a chemical directly causes clonal expansions because it causes mutations at sequence contexts that important for genes that are strongly selected for 2) a chemical directly causes clonal expansions by inducing cell divisions and the only ones that really respond are the cells with prior existing mutations in genes that cause cells to be more likely to divide faster or something similar or 3) a chemical causes damage to surrounding cells and cells that are more likely to divide take over in a wound healing-like response. The authors don't really define what they mean, so it's tough to really evaluate the interpretation of their analysis.

Authors' response: We thank the reviewer for raising this issue, providing us the opportunity to clarify what we mean by selectogenesis.

In the original submission, we explained what we mean by "selectogen" and why this term is helpful in a detailed supplementary note (please see **Supplementary Note 5**). This included several examples as well as a formal mathematical definitions of "mutagens" and "selectogens" in the context of the equations of multistage models of carcinogenesis. Some selected paragraphs from the current **Supplementary Note 5** are pasted below (green text). We also note that the term "selectogen" is not new and was coined by Vineis et al. (*Carcinogenesis*, 2010, PMID:20430846), which we cite in the main text when we first use the term "selectogen". To clarify, the first of the three activities listed by the reviewer corresponds to a mutagenic effect and the last two to selectogenic effects. We kindly direct the reviewer to **Supplementary Note 5** and to the paragraphs below for more details.

Mutagens and selectogens

Understanding cancer development as a process of somatic evolution can help unify different models of carcinogenesis, including the somatic mutation theory, the initiation-promotion theory and the tissue organisation field theory (see (Vineis et al. 2010)). These models are often presented as alternative or even opposing models. However, they can also be understood as emphasising a different aspect of the somatic evolutionary process, namely somatic mutations, non-mutagenic changes to clonal selection, and the role of tissue architecture and the microenvironment in shaping selection, respectively.

(...)

In the last few years, several discoveries have led to a renewed interest in the initiation-promotion model of carcinogenesis (Balmain 2020; Hill et al. 2023). This model originated in the 1940s with animal studies demonstrating that tumours could be induced in animals by the successive application of an initiator (a mutagen) and a promoter (e.g. a non-mutagenic irritant that favours the growth of mutant cells). The renewed interest in this model stems from the need to recognise the importance of non-mutagenic carcinogens in carcinogenesis. However, we argue that the current understanding of carcinogenesis as a somatic evolution process offers a natural and mechanistic way of incorporating the role of promoters in a multi-stage model of carcinogenesis, unifying the somatic mutation theory and the initiation-promotion theory.

Under the paradigm of somatic evolution, many non-mutagenic carcinogens or promoters are expected to act by favouring the expansion of mutant clones, that is, by altering clonal selection (Vineis et al. 2010). This can happen through a wide variety of mechanisms, such as increasing proliferation, inducing apoptosis of wild-type cells, causing injury and regeneration, altering interactions with the cellular microenvironment, or enabling immune escape. If these processes lead to an increase in the number of cells with cancer-driver mutations, they are expected to increase the risk of cancer, as shown in the multistage models above. In that context, we argue that many (perhaps most) promoters can be referred to as “selectogens”, which we think is a more precise and mechanistic term than “promoter”, just as we currently use “mutagen” instead of “initiator”. We note that the term “selectogen” has already been coined for this purpose (Vineis et al. 2010).

We think that the term “selectogen” can help reconcile the somatic mutation and the initiation-promotion models of carcinogenesis under the umbrella of somatic evolution. This borrows the important notion of non-mutagenic influences on clonal growth from the initiation-promotion model, while avoiding its classical association with a two-stage model, incompatible with the genomics of most cancers and premalignant lesions. Instead, cancer development can be understood as a multi-stage process of somatic evolution where both increases in mutation rates and changes in selection can increase cancer incidence by increasing the number of cells at risk of transformation.

The concept of mutagenesis and selectogenesis can also be formalised with the help of the mathematical multistage models described above. Mutagens are expected to increase cancer incidence $-I(t)-$ by increasing the number of mutations per cell. Selectogens can enter into the multistage equations in at least two distinct ways. A selectogen could increase incidence by favouring larger clonal expansions of cells with driver mutations. This class of selectogens, like mutagens, would increase incidence $-I(t)-$ by increasing the likelihood of a single cell acquiring the full complement of driver mutations needed for transformation. A second class of selectogens are exposures that alter the fitness landscape leading to a reduction of the exponent in the multistage model or a broadening of the set of driver mutations that could lead to transformation. For example, drug-induced immunosuppression could eliminate one barrier to transformation that may have otherwise required an immune-escape driver mutation.

The multistage equations also suggest that an exposure that increased the rate of cell division, cell death or regeneration equally on wild-type and driver-mutant cells without increasing mutation rates (mutagenesis) or the fraction of cells with driver mutations (selection), would not be expected to increase cancer incidence. The equations also suggest that other non-mutagenic ways of increasing cancer incidence could be an absolute increase in the number of cells susceptible to transformation (contained within the k parameter). Genetic predisposition to cancer due to a heterozygous germline hit in a two-hit tumour suppressor gene (e.g. RB1 mutations in familial retinoblastoma) can also be understood as acting by reducing the exponent in the multistage equations, whereas germline predisposition due to germline mutations in DNA repair genes could increase cancer incidence by changes in both mutation rates and the exponent.

Finally, whereas carcinogens are often classed as either mutagens or promoters, this classification is simplistic as some carcinogens act simultaneously as both. For example, ultraviolet light acts as both a mutagen and a promoter/selectogen on the epidermis, by causing mutations and by favouring the expansion of TP53-mutated cells which are more resistant than wild-type cells to UV-associated apoptosis or differentiation (Klein et al. 2010). Similarly, several chemotherapies as well as ionising radiation are directly mutagenic while also favouring the expansion of mutant clones more resistant to cell death (Hsu et al. 2018; Pich et al. 2022; Fernandez-Antoran et al. 2019). The ability to separately quantify mutation rates and selection through genomic studies of normal or precancerous tissues, as exemplified in the current manuscript (see Fig. 4 and Supplementary Note 7), provides a framework to quantify the mutagenic and selectogenic effects of different carcinogens, in humans and in experimental models (in vitro or in vivo).

(...)

Action: We thank the reviewer for this comment. To minimise confusion, in the revised manuscript, we point readers more clearly to **Supplementary Note 5** when we first use the term “selectogen” to reduce the risk of these details escaping unnoticed: “... selectogens (see **Supplementary Note 5** for an extended explanation)”. We have also made considerable changes to the text in **Supplementary Note 5** to improve its clarity.

The study does have several unique qualities. For example:

- 1) The scale of the study does set a new standard for what’s possible in studying somatic mutations and duplex sequencing.
- 2) The size of the sample cohort and the large number of detected mutations allows for a novel epidemiological analysis of somatic mutations. The authors do provide evidence that suggests that the “intensity” of SBS5 is heritable, but the effect is modest and, by their own admission, there was not enough power to observe a significant genome-wide association or see other associations.
- 3) They also note patterns of mutations and clonal expansions are seemingly constrained in solid tissues, but seemingly not in blood.
- 4) Another novel finding is the evidence for negative selection of mutations in some genes in somatic tissue, which, while not entirely surprising, is an important and previously unreported finding.

I do want to be clear here. This work will be of significant interest to the research community and definitely deserves to be published in a high-impact journal, but I’m less sure if the novelty of the findings rises quite to the level of Nature.

Authors' response: We thank the reviewer for highlighting these positive aspects of our study. Motivated by this reviewer's comments, we hope that we now better emphasise several novel aspects of our study that were admittedly insufficiently described in the original manuscript.

Referee #1 (Remarks on code availability):

The file format was not given (.unk) and changing to the standard .zip, .gz, and .tar did not open them. A link to a software repository was not provided, as far as I could tell.

Authors' response: We apologise for the confusion. The file that we provided was a .html Markdown, but the extension of the file seems to have been removed by the Nature online submission system. The .html Markdown file combines blocks of code with explanatory text and results (numbers, figures and tables), and was intended to make the code as readable as possible.

To maximise reproducibility, all analyses in the Markdown can be rerun in ~1 hour, which reproduces the majority of analyses and figures of the study. However, this requires access to intermediate files that cannot be shared publicly but that can be obtained from TwinsUK. The TwinsUK registry has strong data sharing rules to protect the anonymity of its participants, so files with patient-level identifiable information cannot be publicly shared. However, researchers wishing to access the full dataset, including the extended metadata and intermediate files, can request access to the TwinsUK registry using an easy online application system. The TwinsUK Resource Executive Committee meets monthly to discuss data access requests. Access to raw sequencing data is also available through the EGA database, under managed access by TwinsUK.

To maximise the utility of the public dataset, TwinsUK have also agreed to allow the publication of the ~340,000 somatic mutation calls openly as a **Supplementary Dataset** with the final publication, without inclusion of donor level information. The .html Markdown file is thus intended to help readers navigate the code and the outputs generated by it even without the need to request access to the restricted dataset.

Referee #2 (Remarks to the Author):

Overall:

Lawson, Martincorena and colleagues describe an improved version of NanoSeq, error-corrected duplex single-molecule sequencing. Such methods aim to substantially improve our understanding in the genesis and evolution of somatic mutations, ideally ultimately toward novel insights in cancer and non-cancer conditions. In increasing set of observations have shown that somatic mutations may have non-cancer implications, further highlighting the importance of such technologies. However, these is an early report of the technology with some early insights. The writing is clear with some early scientific observations. I have some initial concerns limiting my overall enthusiasm of the manuscript in its present form.

Major:

1. The authors cite two other recently described single-molecule duplex sequencing methods with low error rates for the detection of somatic mutations – HiDEF-seq (Long MH et al *Nature* 2024) and CODEC (Bae JH et al *Nat Genet* 2023). However, it is unclear what the advantage would be of the present method over the other methods. Benchmarking with comparisons would significantly enhance this contribution.

Authors' response: We thank the reviewer for this comment, as it allows us to preempt a common misunderstanding in the main text. As we explain below, while different error-corrected sequencing methods are often discussed as alternative protocols, the innovations in NanoSeq, CODEC and HiDEF-seq are in fact complementary as they refer to different steps in the protocols and they serve different purposes (Bae et al., *Nature Genetics*, 2023, PMID:37106072; Hong Liu et al., *Nature*, 2024, PMID: 38867045).

The original NanoSeq protocol was published in 2021 (Abascal et al, *Nature*, 2021, PMID:33911282) and introduced several changes to the standard Duplex Sequencing (DS) protocol to achieve an unprecedented error rate $<5e-9$ errors/bp (compared to the error rate of standard DS protocols $\sim 1e-7 - 2e-7$ errors/bp). This was key as it reduced the error rate of somatic mutation detection in single DNA molecules two orders of magnitude below the somatic mutation load of adult human cells, enabling for the first time the accurate quantification of somatic mutation rates in any human cell type. However, the original protocol provided coverage for $<30\%$ of the human genome, limiting its utility to some applications (e.g. while it enabled the study of mutation rates and signatures it was not appropriate for driver discovery or deep targeted studies). The new version described in the present manuscript solves these limitations and provides the first true genome-wide single-molecule sequencing method with $<5e-9$ error rates.

CODEC and HiDEF-seq were published in 2023 and 2024, respectively, and both adopted innovations from the 2021 NanoSeq protocol to reduce their error rates (i.e. they are not entirely independent protocols but borrowed innovations introduced in 2021 in NanoSeq). The key innovations in these two methods were: (1) changing the adapter design to reduce sequencing costs (CODEC), and (2) changing the sequencing platform (PacBio in HiDEF-seq instead of Illumina in NanoSeq) to obtain long reads and improve DNA damage detection. Thus, it is important to clarify that the innovations in CODEC and HiDEF-seq are not alternative solutions to the same problem, but rather complementary improvements of different aspects of error-corrected sequencing methods. This is particularly important to understand the value of the new NanoSeq protocols, since the two new genome fragmentation methods that we introduce in the new NanoSeq protocols could be used followed by CODEC adapters or HIFI PacBio sequencing to perform lower error-rate whole-genome sequencing. The choice between these methods will depend on the application. CODEC should reduce sequencing costs by $\sim 2-3$ -fold compared to NanoSeq, as it relies on 1+1 calling rather than 2+2 calling, but this comes at the cost of higher error rates due to the limitations of 1+1 calling. HiDEF-seq could be preferred for applications requiring long-read sequencing, but at a higher sequencing cost (PacBio vs Illumina), much higher DNA input requirements, and not being cost-effective for targeted gene sequencing.

Action: We appreciate that the point above was not made sufficiently clear in the original manuscript, and we thank the reviewer for highlighting this. We have clarified this point in the revised version of the main text, which we hope helps users appreciate the complementary nature of these technologies and the potential benefits of combining them. Importantly, the NanoSeq protocols introduced in the current manuscript are the only protocols to date that enable whole-exome or deep-targeted single-molecule sequencing with error rates $<5e-9$ errors/bp. This is not a small improvement compared to the existing technologies but a considerable leap, as they are the first protocols to enable accurate ($<5e-9$ errors/bp) single-molecule studies of mutation rates, mutation signatures and driver landscapes from any cell type, including non-invasive samples.

2. With very little exception, bulk WES/WGS of blood to genotype CH-driver mutations has identified DNMT3A as the top gene with driver mutations and TET2 as the second top gene. The only exception that I'm aware of is targeted NGS in the CANTOS trial (coronary artery disease and hsCRP >2), where TET2 was the top gene. However, Fig 1g shows TET2 as the top gene and DNMT3A as the second top gene. This significant deviation from the expectation leads to concerns about bias related to ascertainment, reduced generalizability of the findings, and potential errors/biases in sequencing.

Authors' response: We thank the reviewer for this comment. This apparent discrepancy has a simple explanation and is not a source of concern. In the original manuscript, we already addressed this in **Supplementary Note 4** (pasted below) and in **Fig. 1g** (bottom panel), but unfortunately we did not explain it in the main text due to space constraints.

Unlike most studies of clonal haematopoiesis conducted using standard sequencing, we observed more non-synonymous mutations in TET2 ($n = 1,104$) than DNMT3A ($n = 800$), which also corresponded to a higher estimated number of driver mutations in TET2 ($n = 853$ and 743 respectively). However, we obtained substantially higher cumulative duplex coverage across TET2 ($436,473$ dx) than DNMT3A ($177,102$ dx). Correcting for this difference, the DNMT3A:TET2 ratio of observed non-synonymous mutations or estimated driver mutations per duplex coverage (1.79 and 2.15 , respectively) is substantially closer to the DNMT3A:TET2 ratio of non-synonymous mutations observed in a large clonal haematopoiesis study (2.37) (Kar et al. 2022). The remaining discrepancy may reflect differences in the exponential growth of DNMT3A- and TET2-mutant clones (Fig. 4a), for example due to lower fitness coefficients for some TET2 mutations making them less likely to reach detectable clone sizes with standard bulk sequencing.

As the text above explains, the reason for the higher number of TET2 mutations detected was that the duplex (single-molecule) depth achieved in TET2 was ~ 2.5 times greater than DNMT3A (i.e. we sequenced 2.5 times more copies of TET2 than of DNMT3A). This affects the absolute number of mutations detected per gene (shown at the top of Fig. 1g), but it does not affect the estimated fraction of cells with a driver mutation in TET2 or DNMT3A as this number is adjusted for duplex coverage. This was already shown in the bottom part of Fig. 1g, but motivated by the reviewer's comment we have made this clearer in the revised main text. In summary, our results are not inconsistent with previous studies.

Having clarified that, it is interesting to note that some discrepancies between bulk sequencing and single-molecule sequencing could in fact have been expected. This is because bulk sequencing can only detect clones that reach a detectable size (above a VAF threshold such as $\sim 1\%$ or 5% depending on coverage), which in blood typically restricts detection sensitivity to driver mutations conferring annual growth advantages $>5\%$ (see for example Watson and Blundell, 2023, *Nat Genet*, PMID:37697102). In contrast, accurate single-molecule sequencing can detect variants at any VAF, with a sensitivity proportional to the VAF of a mutation, which can reveal driver mutations with $\ll 5\%$ growth rates. This means that single-molecule sequencing can identify a broader range of driver sites below the detection sensitivity of bulk sequencing, which could manifest on differences in the relative frequencies of

different driver genes. Having said that, as we clarify above, our corrected ratios of *DNMT3A* to *TET2* are not inconsistent with previous bulk sequencing reports.

Action: We now clarify this point in the revised main text, citing the estimated fraction of cells with driver mutations in *DNMT3A* and *TET2* (**Fig. 1g**), which are not inconsistent with previous studies. We have also taken this opportunity to highlight the interesting biological implications of potential differences between bulk sequencing and single-molecule sequencing.

3. In the analysis of blood mutations, the authors feature that their yield for *DNMT3A* and *TET2* driver mutations is much higher than standard-coverage WES/WGS and is thus a feature of their single-molecule duplex sequencing approach. However, targeted error-corrected deep bulk sequencing will also substantially increase the yield (Young AL et al Nat Comm 2016) so this might not be a compelling primary argument to feature. An argument could be that error rates would be lower with a single-molecule approach would be reasonable to add but I have technical concerns given the unexpected distribution of driver mutation genes (see major point 2 above).

Authors' response: We thank the reviewer for this comment. We agree. We want to clarify that when we described the ~100-fold higher yield of driver mutations in blood using NanoSeq compared to standard sequencing, we were only describing the nature of our dataset. We did not intend to imply that this was an unexpected or a novel result for those very familiar with the relevant literature.

For clarity, our description of the landscape of blood drivers was intended to be confirmatory, which is why we included it in **Fig. 1**, next to the description of the new NanoSeq protocols. Since clones in blood expand to large enough sizes to be detected by previous error-corrected methods and the driver landscape in blood is well known, the drivers that we found in blood are not novel, but serve to showcase the power of targeted NanoSeq (notice that we have found approximately as many driver genes in blood with 371 samples as previous studies with tens of thousands of samples). Having said that, whereas the drivers discovered in blood are not new and were used as a validation, a clear advantage of using NanoSeq in blood instead of previous error-corrected sequencing methods is NanoSeq's ability to accurately measure mutation rates and signatures as well as drivers in large cohorts of individuals, which had not been possible with previous bulk technologies. So, to clarify, the two main purposes of the blood data in the paper were to: (1) show the power of targeted NanoSeq in a well-known tissue as a validation, and (2) complement the buccal swabs analyses (e.g. to help quantify the low-level presence of blood clones in the buccal swabs).

We emphasise that the novelty of the paper resides on the buccal swab data and the analyses and discoveries stemming from it.

Action: We have modified the relevant paragraph in the main text to clarify that the high yield of driver mutations in blood is expected for deep error-corrected sequencing, even if it remains striking for those readers only familiar with the large-cohort bulk sequencing studies restricted to VAFs>1%.

4. Given the genome-wide mutational profiles, it would be neat to apply methods derived from bulk WGS (Weinstock J et al Nature 2023) to understand the dynamics and estimate the ages of driver mutations.

Authors' response: We thank the reviewer for this question. However, this point (#4) and the point below (#5) suggest a misunderstanding of the NanoSeq data and of the mutation landscape of solid tissues that it is important to clarify.

The clonal landscape in blood is often dominated by one major clone, occasionally at VAFs>5%. Weinstock et al (PMID:37046083) neatly took advantage of this to estimate the age of driver mutations using the number of passenger mutations in the expanded clone, which could be detected with standard

WGS data. With a single very large clone in standard bulk sequencing data, the number of passenger mutations in the clone can inform about its age (with some assumptions), and the VAF of the clone can inform about its size. Based on the estimated age and size of the dominant clone, and assuming a model of exponential growth, they could obtain an estimate of its annual growth rate.

Whereas this was a clever and elegant analysis, this is not possible and it would be incorrect in oral epithelium for several reasons. First, unlike blood, where a major clone dominates and WGS can be used to count the number of passengers in the dominant clone, the oral epithelium is composed of thousands of microscopic clones, without a large dominant clone. The polyclonality is so extreme that ~96% of the mutations we detect were found in a single molecule, and ~99.6% of the clones that we found occurred at VAFs < 1%. Thus, standard WGS cannot be used to detect the ancestral passenger mutations in a clone (which is necessary to estimate the age of a clone in Weinstock et al). Second, the whole-genome data that we describe in the paper is whole-genome NanoSeq data, which provides an effective coverage of ~0.5-1 dx (duplex depth) per genome. Third, and importantly, as we show in the manuscript, clones in the oral epithelium do not grow exponentially, which is a key assumption in Weinstock et al, and so the Weinstock et al model would not be applicable to oral epithelium. In fact, as we explain in the manuscript, our evidence suggests that clones in the oral epithelium do not grow continuously over time (another assumption in Weinstock et al), which means that clone sizes cannot be used to infer the age or growth rate of a clone in oral epithelium (and likely in other solid tissues).

Although the approach described by Weinstock et al cannot be applied to the current study, in the manuscript we showed that we can derive important and novel information about clonal growth dynamics from targeted NanoSeq data by comparing mutation rates and driver frequencies across individuals of different ages. We kindly refer the reviewer to **Supplementary Note 6** for full details (also see **Fig. 4a** and **Extended Data Fig. 7**). Briefly, since we now know that mutations are accrued linearly in the oral epithelium (**Fig. 2b**), the way the aggregate fraction of cells with a driver mutation increases with age (e.g. linearly, quadratically, cubically, exponentially...) depends on the type of clonal growth active in the tissue. Analysis of our data shows that clones in normal oral epithelium are highly constrained, more consistent with a logistic growth model than with models of continued clonal growth with age.

5. Since the authors show that positively selected mutations appear rather ubiquitous in polyclonal tissues, the clinical relevance of these very very subclinical findings is unclear. Further analysis of the data to infer early predictions of future clonality could help towards this goal. Nevertheless, the authors describe that while NOTCH1 driver mutations are common and have high VAFs in buccal tissue, risk of transformation (relative to others) is low. This indicates that models purely describing clonality/fitness may not necessarily translate to neoplastic risk.

Authors' response: We thank the reviewer for this comment. We try to clarify these points below.

First, the reviewer says that "*the authors describe that while NOTCH1 driver mutations are common and have high VAFs in buccal tissue...*". We may be misunderstanding this sentence, but we want to clarify that the VAFs of the NOTCH1 clones that we report in the paper were almost always extremely low, as we are detecting microscopic clones. Specifically, the median VAF of the reported NOTCH1 mutations was 0.0007. It is the sum of many microscopic clones that leads to a substantial fraction of cells having mutations in NOTCH1 in the oral epithelium, without any clone reaching high VAFs. This is also shown in **Fig. 4a**, and is in stark contrast with the pattern in blood, which is why a technology like targeted NanoSeq is needed. The high polyclonality is a characteristic feature of the clonal landscapes of every epithelia that we have studied to date (including skin -Martincorena et al, *Science*, 2015, PMID:25999502-, oesophagus -Martincorena et al, *Science*, 2018, PMID:30337457-, endometrium -Moore et al, *Nature*, 2020, PMID:32350471-, and bladder -Lawson et al, *Science*, 2020 PMID:33004514-, for example).

The reviewer then asks about the clinical implications of these small clones, which is a very interesting question. The discovery of clones with cancer-driver mutations in polyclonal solid tissues has caused some confusion about their relevance to cancer. To preempt this confusion, in an extended supplementary note we explained how such clones are expected and consistent with classical multistage models. These simple models provide a quantitative link between mutation rates, clone sizes and cancer risk. We kindly refer the reviewer to **Supplementary Notes 5 and 6** where we discuss this in detail. A toy example may help clarify the way in which the observed microscopic clonal expansions are relevant to cancer risk. If mutation of *TP53* is essential for a cell to eventually form an oral tumour (upon further acquisition of additional driver mutations), the multistage models predict that an individual with twice as many cells with *TP53* mutations as another, all other things being equal, has a 2-fold higher risk of cancer (as it has twice as many cells at risk of acquiring additional hits and evolving into tumours). The higher number of cells with *TP53* mutations could be due to mutagenic exposures doubling the number of mutations per cell or to selectogenic/promoter exposures doubling the size or frequency of *TP53* clones, or a combination of the two. Thus, a single microscopic clone with a *TP53* mutation in a given individual is not expected to increase risk significantly, but, on aggregate across clones, individuals with a higher fraction of *TP53* mutations are expected to be at higher risk of cancer.

As we explain in **Supplementary Note 5**, multistage models with clonal expansions help connect the recent discoveries on clonal and polyclonal landscapes to the classical literature on carcinogenesis. We note that these models are not dissimilar to the risk models used in clonal haematopoiesis. In blood, where a major clone often dominates (**Fig. 4a**), the size of that clone, the gene mutated in it, and the growth rate of the clone will dominate the probability of subsequent hits and the risk of transformation. In solid epithelia, which are composed of thousands of clones, the aggregate risk of large numbers of clones is expected to determine cancer risk.

The current study already shows clear associations between oral cancer risk factors (e.g. smoking, alcohol, poor oral health) and changes in mutation rates and clonal frequencies, in the directions expected if these clones contribute to cancer risk. A more direct quantification of the connection of clonal landscapes and cancer risk will require future case-control studies and longitudinal studies. Whereas this is beyond the scope of the current study, we believe that the availability of targeted NanoSeq and the framework presented in the current study makes it possible to design such studies (**Fig. 4j**).

I hope this clarifies the reviewer's questions but we are grateful for any further comments to increase the clarity of this section.

6. The authors suggest that 28 of 49 genes from buccal swabs are under positive selection. Replication in another dataset or validation in an experimental model would increase confidence in this potential novel finding. The claim that ~62,000 mutations are anticipated to be driver mutations seems potentially substantial, and would benefit from a validation of a subset of novel observed mutations to increase confidence that this is a suitable claim to make.

Authors' response: We thank the reviewer for these suggestions. Firstly, we would like to clarify that in the current study we reported 49 genes under positive selection in normal epithelium out of 239 genes sequenced (rather than 28 out of 49). In the original manuscript, we had noted in **Supplementary Note 4** that the signal of selection in 3 of these 49 genes (*DNMT3A*, *TET2* and *FOXPI*) was likely due to low-level blood contamination of the buccal swabs. Therefore, in our revised manuscript we have clarified in the main text that 46 (out of 239 genes) are bona fide drivers in oral epithelium.

Regarding the second point, unfortunately, it is not feasible to replicate all these genes in another dataset as replicating infrequent drivers would require an effort of at least the same size as the present study. This is very hard to do both logistically (the collection and sequencing of >1,000 buccal swabs was a major endeavour requiring 2 years of work) and financially (the amount of sequencing required for this study was ~50 terabases, comparable to sequencing ~25,000 standard whole-exomes at 50× coverage).

However, we can provide several additional analyses to support the robustness of our driver discovery results.

First, several external datasets support the validity of our driver discovery results:

1. For blood, all 14 genes that we found under positive selection using targeted NanoSeq and dNdScv are known clonal haematopoiesis drivers (**Fig. 1**). These results support our methodology for driver discovery using targeted NanoSeq.
2. For the buccal swabs, given that ours is by far the largest dataset of its kind, we do not expect all the drivers to be known, however, two previous studies of the driver landscape in normal oesophagus (Martincorena et al, *Science*, 2018, PMID:30337457; Yokoyama et al, *Nature*, 2019, PMID:30602793) together reported 17 genes under selection in normal squamous oesophagus. Of these, 14 are under selection ($q\text{-value}<0.05$) in our buccal swab study (*NOTCH1*, *TP53*, *NOTCH2*, *FAT1*, *NOTCH3*, *ARID1A*, *CUL3*, *AJUBA*, *ARID2*, *TP63*, *CCND1*, *PPM1D*, *ZFP36L2*, *CHEK2*) and we also observed significantly recurrent hotspot mutations in *PIK3CA*. This level of validation is remarkable, considering that we do not expect the driver landscape of oral and oesophageal epithelium to be identical.
3. Comparing the driver landscape in the buccal swabs with the driver landscape in head and neck cancers provides further support for our drivers. We find that 21 of the 46 driver genes in normal oral mucosa have significant evidence of positive selection in a dataset of 500 head and neck cancers from TCGA (nominal $P\text{-values}<0.05$).

Second, we can carry out an internal cross-validation to provide further support for the robustness of the new drivers and of our discovery approach. To do so, we used 2-fold cross-validation, running *dNdScv* on a random 50% of the samples in the study and calculating how many of the significant genes in the training set (with our discovery cutoff of $q\text{-value}<0.01$) show statistical evidence of positive selection in the validation half of the dataset ($P\text{-value}<0.05$). Repeating this 2-fold cross-validation analysis 10 times yielded an average validation rate of ~94% (i.e. 94% of the genes found as significant using $q\text{-value}<0.01$ in the training set were validated as being under selection in the validation set). This and the analyses above, confirms the robustness of our driver discovery approach. We also note that *dNdScv* and our driver discovery strategy are standard and have been used by large numbers of studies in the somatic literature, and benchmarked by the TCGA-PCAWG consortium (Rheinbay et al, *Nature*, 2020, PMID:32025015).

We hope that these analyses provide additional support for the robustness of our driver discovery analysis.

Action: We thank the reviewer for their constructive suggestion. We now summarise these new analyses in a new section of **Supplementary Note 4**, and we cite this section in the main text when describing the driver genes discovered in the buccal dataset.

7. Based on Fig 2g, individuals with NOTCH1 driver mutations uniquely had much higher VAFs. Is there histopathology that such individuals already had dysplasia or neoplasia?

Authors' response: This point suggests a misunderstanding of our data, which is important to clarify, as it affects several figures. We apologise for any confusion.

Fig. 2g does not show that there are some individuals with *NOTCH1* driver mutations and high VAFs. In fact, the figure does not contain information about individual donors. **Fig. 2g shows the mean percentage of cells with NOTCH1 mutations in oral epithelium across donors.** This estimate is obtained by aggregating large numbers of clones per donor (see “Duplex VAF and unbiased VAF” section in Methods) and is an average across all individuals in our cohort between the ages of 65 and 85. The same approach and representation was used in our previous studies in skin (Martincorena et al, *Science*,

2015, PMID: 25999502), oesophagus (Martincorena et al, *Science*, 2018, PMID: 30337457), and bladder (Lawson et al, *Science*, 2020, PMID: 33004514).

Unlike blood, where some older individuals have one large dominant mutant clone at moderate to high VAF, the clonal landscape of normal oesophagus, skin or oral epithelium is composed of thousands of typically-microscopic clones (see **Fig. 4a**). This polyclonal landscape is represented in the image below (from Martincorena *et al.*, *Science*, 2018, PMID:30337457), where each square represents 1 cm² of histologically normal oesophagus, with individual clones shown as circles (purple clones correspond to *NOTCH1*, which is also ubiquitous in normal oesophagus).

Consistent with the results in skin, oesophagus, bladder and other epithelia, the oral landscape is also very polyclonal and *NOTCH1* clones are ubiquitous in our dataset. In fact, effectively all donors in our buccal dataset carry *NOTCH1* clones (774/774 buccal donors with duplex coverage ≥ 200 dx and ≥ 50 years old have at least one *NOTCH1* mutation) and these clones are almost invariably very small (the mean VAF of *NOTCH1* clones at sites with ≥ 1000 x coverage is $\sim 0.1\%$, and 99.9% of the *NOTCH1* mutations have VAFs $< 1\%$). So, to clarify the reviewer’s comment, we do not see a subset of individuals with *NOTCH1* mutations (they all have them), and these mutations are not of high VAF in those donors (they are almost universally small clones, as seen in previous studies of skin, oesophagus or bladder).

The reviewer however raises an interesting question about the extent of variation in driver density across individuals. This question is already considered in the epidemiological regressions and heritability analyses in **Fig. 4** of the manuscript. However, motivated by the reviewer, in the revision we are adding two new analyses to further describe and quantify the variation in driver density across donors:

1. New figure on the variation in driver density across donors: we have added two new plots to **Extended Data Fig. 5d** (reproduced below). The first plot shows the estimated mutant cell fraction per donor (by aggregating all mutations observed within a donor, using $2 \times \text{sum}(\text{duplexVAF})$, as described in the **Methods**). This shows the pervasiveness of these driver mutations across donors and the variation in the estimated fraction of mutant cells (summed across clones) per donor, for each gene. The second plot addresses another of the reviewer’s questions, by showing the distribution of unbiased VAFs (see **Methods** for the definition of “unbiased VAF”) of individual mutations across genes, which shows that the vast

majority of the observed clones are very small. This analysis graphically emphasises that the vast majority of mutations observed have very low VAFs (median unbiased VAF across genes <0.0005). Only a few mutations (out of $>300,000$) were found with VAFs $>2\%$ in the buccal swabs, and these mostly corresponded to common clonal haematopoiesis drivers (*DNMT3A*, *TET2* and *PPM1D*). Only 6 *TP53* mutations in the entire dataset (out of 8,079 *TP53* mutations found) had VAFs $>2\%$. We cannot confirm whether these are rare buccal clones or clonal hematopoiesis clones as matching blood was not available from these few donors and *TP53* is also a CH driver, but overall these results show that large clones are extremely rare in the buccal epithelium.

2. **Quantifying the extent of variation across genes:** To explore whether certain driver genes show a more extreme variation in mutation density across donors, while accounting for duplex coverage per gene, we then used a negative binomial regression (see a detailed description at the end of **Supplementary Note 4**, and **Supplementary Code** file). This confirms that there is significant heterogeneity (measured as overdispersion) in the driver density across donors in all genes studied. Reassuringly, clonal haematopoiesis genes (specifically *DNMT3A*, *TET2* and *PPM1D*) have considerably larger inter-individual variation in the fraction of mutant cells per donor than buccal drivers, which is expected given the ability of blood clones to grow semi-exponentially (unlike buccal clones), which exacerbates the inter-individual heterogeneity.

We hope that the analyses above clarify the reviewer's question and provide additional information on the variation in the clonal landscape across donors. To summarise, consistent with previous studies in skin, bladder, oesophagus and other solid tissues, our data reveals that oral epithelium in the general population is composed of large numbers of small mutant clones. All older donors in our cohort have *NOTCH1* mutations and these clones are almost universally small ($\text{VAF} \ll 1\%$), but on aggregate they account for a significant fraction of all cells in each donor ($\sim 10\%$, see **Extended Data Fig. 5d**, reproduced below). These analyses also show that the driver landscape is qualitatively similar across donors but that statistically significant heterogeneity is observed in the frequency of cells with driver mutations across donors. This heterogeneity is larger for clonal haematopoiesis drivers, as expected due to the semi-exponential growth of clones in blood. The epidemiological regression analyses in **Fig. 4** shed light on the exposures and lifestyle factors that explain some of this variation.

Finally, given the non-invasive nature of our collection, we do not have spatial or histological information to determine whether any of the few large clones look histologically aberrant, but we can confirm that large clones are extremely rare in our dataset. We also note that previous microscopy studies have shown that microscopic clones carrying driver mutations in skin, oesophagus or bladder epithelia are ubiquitous in histologically normal tissue. We do have high-quality phenotypic data on each participant that confirms no donor had a diagnosis of head and neck cancer or dysplasia, at the time of sampling.

Action: We thank the reviewer for this question, which has motivated us to strengthen this part of the manuscript. We hope that our comment sufficiently addresses the reviewer's question, and we welcome the opportunity to add these further analyses, which we believe highlights the remarkable size of this dataset, and better show the variation in the clonal landscape across donors and in clone sizes across genes. We have updated the main text to make these points accessible to future readers and added two new plots in **Extended Data Fig. 5d**.

New Extended Data Fig. 5d. Left. Estimated mutant cell fraction per donor for each gene. Every dot represents one donor in the dataset (restricted to 65-85 year old donors). The estimated mutant cell fraction represents the upper bound estimate using the sum of duplex VAFs multiplied by 2 for genes in diploid chromosomes (see Methods for an explanation on the assumptions and rationale). This analysis takes into account all non-synonymous mutations observed in each gene in each donor. The red line represents the median estimated cell fraction across donors. **Right.** Observed unbiased VAFs (shown as percentages) for all non-synonymous SNVs in each gene in each donor, restricted to sites with $\geq 1000\times$ coverage. This highlights that the vast majority of mutations observed across genes have very low VAFs, with only a small number of mutations having VAFs $> 2\%$ (largely in clonal haematopoiesis genes and a few in TP53).

8. These authors (Martincorena et al Cell 2017) have done nice work in TCGA showing the absence of negative selection in cancer tissues. In the present work, of noncancerous tissues, they now show that there is some negative selection of somatic mutations. Is this a contextual difference between cancer and non-cancer? And given the very small number of genes identified, perhaps it's still safe to say that negative selection is significantly less common for somatic versus germline mutations?

Authors' response: Thank you for the kind comment. As the reviewer notes, our previous study (Martincorena et al, *Cell*, 2017, PMID:29056346) showed that negative selection is rare in cancer evolution, with exome-wide dN/dS ratios very close to 1. Negative selection was not entirely absent in that study, as we found evidence of it in essential genes in haploid genomic regions (Fig. 3G of Martincorena et al, *Cell*, 2017). We also ran power calculations that showed that much larger studies would be needed to identify individual genes under negative selection in cancer or normal tissues. Fig. 3E in that study showed that datasets with hundreds of mutations per gene would be needed to detect negative selection sensitively on single genes (typically requiring cancer genomic datasets of tens of thousands of tumours). However, despite not having power to identify individual genes under negative selection in the 2017 study, we performed an exome-wide analysis that predicted that $\sim 0.5\%$ of genes might have dN/dS ratios < 0.75 in cancer genomes (Fig. 3C of the 2017 paper, PMID:29056346).

Reliably identifying individual genes under negative selection has remained elusive in cancer and normal tissues since then as it requires very large (or very deep datasets). The present study provides

convincing evidence of individual genes under somatic negative selection in oral epithelium. Thanks to the single-molecule sensitivity of targeted NanoSeq, our depth of sequencing, and the polyclonality of buccal swabs, the current study has surpassed the required number of mutations to begin to detect genes under negative selection. Overall, the median number of mutations per gene in the entire buccal swab dataset is 348, with 70% of genes (168 out of 239) having more than 200 mutations. As a result, we have now been able to detect negative selection in several genes, as described in the main text.

Although we now have the power to identify genes under negative selection, it is important to emphasise that the results remain consistent with those that we found in cancer genomes (Martincorena et al, *Cell*, 2017). As in cancer genomes, in the buccal swabs our exome-wide dN/dS ratios (excluding major driver genes) are close to 1 (dN/dS = 1.04, CI95%: 0.99-1.09, see **Fig. 2k in the current manuscript**). However, exome-wide dN/dS ratios are weighted averages across genes (as shown in the 2017 *Cell* paper) and significant negative selection is expected in a minority of genes, which is what we predicted in the 2017 *Cell* paper and what we have now confirmed in the current study.

To further clarify this point, and motivated by the reviewer's comment, in the revision we provide a new figure that shows the observed dN/dS ratios per gene for the 239 genes in the dataset, with 95% confidence intervals (**Extended Data Fig. 5f, also reproduced below**). This new figure shows that dN/dS ratios are very close to 1 in the vast majority of genes in the study, with only a small minority of genes showing clear negative selection. Overall, this pattern is consistent with that inferred from cancer genomes in Martincorena et al, *Cell*, 2017 (see Fig. 3A and C from that study for comparison), although a more detailed comparison of negative selection in cancer vs oral epithelium is not possible due to the lack of comparatively deep cancer datasets.

Thus, as the reviewer suggests, it remains safe to say that the accumulation of somatic mutations in cancer genomes and in normal tissues is remarkably different to that in the germline, with the vast majority of mutations accumulating neutrally (exome-wide dN/dS~1), but with a minority of genes having clear signs of negative or positive selection. And, thanks to targeted NanoSeq, we now have the ability to detect individual genes under negative selection in polyclonal normal tissues.

Action: We thank the reviewer for raising this question. We have clarified this point in the main text and we have added a new figure (**Extended Data Fig. 5f**) to strengthen this point.

New **Extended Data Fig. 5f**, showing that dN/dS ratios for most genes are close to 1. dN/dS estimates per gene are more precise for missense mutations than for truncating mutations (nonsense and essential splice sites) as missense mutations are more frequent.

9. Using prior germline genetic determinants of CH based on population-based WES/WGS studies (Kar SP et al *Nat Genet* 2022, Kessler MD et al *Nature* 2022), what are the mutational burdens in driver and non-driver genes for those with increasingly greater polygenic predisposition to CH? Might this give more insights into more clinically relevant clonality?

Authors' response: We thank the reviewer for this question, which has motivated us to add additional analyses on the associations of CH and cancer risk alleles with the mutation landscape in oral epithelium.

As far as we know, there is no publicly available polygenic risk score (PGS) for CH. Instead, we tested 34 germline variants associated with CH, from a curated list by Liu et al (*Leuk Res.*, 2023, PMID:37421681), which included the studies suggested by the reviewer, and from Kar et al (*Nature Genetics*, 2022, PMID:35835912). We also tested 18 SNPs associated with higher risk of oral cancer or head and neck cancer. To increase our power to detect associations with mutation rates and drivers, we combined the array genotypes with our targeted sequencing data to improve the genotyping of some of these sites. We then repeated the association analyses (using the same model as described in the GWAS analysis, **Supplementary Note 8**) to test whether these SNPs showed associations with the SNV mutation rate, the Signature A or B burdens, the density of driver mutations in *TP53* or *NOTCH1* in the buccal swabs, and the density of driver mutations in *DNMT3A* or *TET2* in the blood samples.

This analysis is now described in the main text and at the end of **Supplementary Note 8** and the *P*-values and effect sizes are reported in the new **Extended Data Table 6**. Although the replication was hampered by our limited power for GWAS, we were able to find several cases with nominally significant *P*-values (*P*-value<0.05) in the direction expected from previous studies. Using BH multiple testing correction, this analysis revealed only one significant association. This was an interesting association between a SNP in *ALDH2* (rs4767364) and the rate of SigB/SBS16 (*P*-value = 0.000094, *q*-value = 0.02), supporting that the association between rs4767364 and the risk of head and neck cancer reported by McKay et al (*PLoS Genet.*, 2011, PMID:21437268) is driven by higher sensitivity to alcohol. This result is particularly relevant, as it is consistent with the known strong associations between two Asian-specific SNPs in *ALDH2* (rs671) and *ADH1B* (rs1229984) and head and neck cancer risk, susceptibility to alcohol, and the burden of SBS16 in normal oesophagus (Yokoyama *et al*, *Nature*, 2019; PMID:30602793). See also our response to question #13 for more details on these SNPs. Whereas the increased risk conferred by the rs4767364 SNP is weaker than the two Asian-specific SNPs, rs4767364 shows moderate to high allele frequencies across all populations (0.3 in our cohort), which allowed us to confirm that the suspected link between this SNP and cancer is driven by alcohol consumption.

Action: We thank the reviewer for this constructive suggestion, which has allowed us to uncover an interesting association between a common SNP and the rate of alcohol-induced SBS16, providing a suggestive mechanism for its association with head and neck cancer risk. We have included a mention to this result in the main text, and a longer description in **Supplementary Note 8** and **Extended Data Table 6**.

10. This metric on line 369 (“Notably, these regressions suggest that one additional year of life causes as many mutations in the oral epithelium as ~2.8 pack-years or ~19.1 drink-years (see **Supplementary Note 7** for caveats and interpretation)”) is provocative. It would benefit from confidence intervals and a clarification of the age range where this is valid in the main text.

Action: Thank you. Following this request, we have added confidence intervals to these estimates. The estimates with confidence intervals are: 15.2 SNVs/year (CI95%: 13.8-16.6), 5.4 SNVs/pack-year (CI95%: 4.0-6.8), and 0.80 SNVs/drink-year (CI95%: 0.60-1.00). These have also been added to the **Supplementary Code** file for reproducibility. This suggests that one additional year of life causes as many mutations as ~2.8 pack-years (Fieller’s CI95%: 2.03-3.63) or ~19.1 drink-years (CI95%: 13.9-24.3). Beyond adding the confidence intervals, however, there are several factors that need to be considered when interpreting these estimates. Since this required more space than we could devote in the main text, these are described in greater detail in **Supplementary Note 7**, which we cite prominently in the main text.

11. The inference that cancer is arising more from mutagenesis than selectogenesis does not appear expected. As the authors show and others have shown, driver mutations appear to be quite common without expansion, and others have suggested that many of these mutations have happened very early in life (Mitchell E et al Nature 2022, Fabre MA et al Nature 2022). So one might infer changes in selection across the life course may ‘unlock’ the potential of these quiescent driver mutations. Additional clarification would be helpful.

Authors' response: We thank the reviewer for this interesting comment. Below we provide a clarification first, and new analyses afterwards.

Clarification: The observation that microscopic clones with driver mutations are pervasive in normal tissues has led some to invoke “tumour promotion” to explain how large numbers of small clones can exist in normal tissues without giving rise to cancer. However, it is critical to note that invoking promotion is not needed, as these clones are not inconsistent with classical multistage models where carcinogens act purely as mutagens (without promotion/selectogenesis). This is because cells need multiple driver mutations to evolve into tumours (see **Supplementary Note 5**). The clones that we and others have observed across a range of tissues typically carry just one (sometimes two) driver mutations and are multiple steps away from a cancer genome (that is, they are not simply ready to expand into tumours upon exposure to a selectogen/promoter). For most cancer-driver genes, the fraction of cells that carry a driver mutation in most tissues is <1%. This means that the probability of a single cell acquiring a full complement of 5-6 drivers, given known mutation rates and clone sizes, remains extremely low ($\ll 1e-10$, e.g. see equation 5 in **Supplementary Note 5**). These classical models help explain why the observation of large numbers of small clones with 1 or 2 driver mutations is not inconsistent with the observed incidence of cancer (even without selectogens/promoters), and why the frequency of such clones is expected to correlate to cancer risk (i.e. they are clinically relevant on aggregate). We offer this clarification to explain that the observation of large numbers of clones with driver mutations in normal tissues does not by itself suggest that mutagenesis or selectogenesis is more important in early carcinogenesis. Thus, we would argue that the extent to which different carcinogens and risk factors act through increases in mutation rates (mutagens) or changes in selection (promoters/selectogens) remains unknown. Having said that, in the paragraph below we refine our previous analyses of mutagenesis and selectogenesis to offer a more balanced summary.

New analyses: By measuring somatic mutation rates, mutation signatures and driver frequencies in individuals with variable risk factors, we can begin to quantify the extent to which different risk factors act as mutagens or selectogens/promoters, at least with regards to their effects on early clonal expansions in normal tissues. In the original manuscript, one sentence said: “*Using different models suggested that most of the association between driver densities and oral cancer risk factors (pack-years, drink-years, and oral health), is explained by mutagenesis rather than selectogenesis*”. This sentence referred to our regression results that showed that smoking, alcohol consumption and poor oral health all contributed to statistically-significant increases in mutation rates, but to less significant changes in selection (clonal expansion). Whereas the results remain unchanged, we have refined their interpretation in the revision based on additional analyses motivated by the reviewers. First, in the revision we have carried out power simulations, which confirm that we have more power to detect changes in mutation rates than selection. This is because the number of passenger mutations per donor (used to infer mutation rates per donor) is larger than the number of driver mutations (used to quantify selection per gene per donor). As a result, comparing the importance of mutagenic vs selectogenic effects on the basis of statistical significance alone can be misleading. To improve this section, we have changed **Fig. 4f** to show side-by-side the effect sizes of the increases in mutation rates and in selection with smoking, which shows similar effect sizes of smoking on mutation rates and on selection. Whereas these results were already described in the original version of the manuscript and the results have not changed, we have improved the way we summarise the results in the main text to show effect sizes and *P*-values, and avoid claims that mutagenesis is quantitatively more important than selectogenesis.

We are grateful to the reviewer for their suggestions, which we think have improved this section of the manuscript.

Actions: We have revised the main text and **Supplementary Note 7** to acknowledge our higher sensitivity to differences in mutation rates than selection, adding a section on power simulations. We have also modified **Fig. 4f** to show the increase in mutation rate and in selection (clonal expansion) for smoking side by side.

12. The authors make several claims about exposures causing various genomic features. With cross-sectional data, one must be cautious about these claims as there are likely numerous potential confounders correlated with tobacco smoking, alcohol, etc.

Authors' response: We fully agree with the reviewer that caution is needed when interpreting epidemiological associations due to potential confounders. We have acknowledged this extensively in the main text and in the **Supplementary Material**. We kindly refer the reviewer to **Supplementary Note 7** where confounders are discussed in detail, for example in the interpretation of slopes or of the interaction effects between smoking and alcohol.

13. The authors note that they found no genome-wide signals for their GWAS of mutation rates or driver densities. Given the very small sample size this is not surprising. A power assessment accompanying this result would be expected.

Authors' response: We are very grateful for this comment, which we think has strengthened the manuscript.

We agree that lack of GWAS hits for mutation rates or driver densities may not be surprising given the available sample size. However, it would not have been inconceivable to find variants with effect sizes on mutation rates or selection large enough to be detectable in our cohort, which warranted attempting a first GWAS on somatic mutation rates. For example, as mentioned in our response to question #9, two SNPs in genes involved in alcohol metabolism are very common in individuals of Asian ancestry: *ALDH2* (rs671) (A) and *ADH1B* (rs1229984) (G). These alleles are strong risk factors for oesophageal cancer, and a study in 2019 measuring mutation rates in normal oesophagus and ESCC cancers from TCGA was able to show a significant association between these risk alleles and the rate of SBS16 somatic mutations with a small cohort of samples (Yokoyama et al, *Nature*, 2019; PMID:30602793). This was possible because these alleles are both very common in Asia and have strong effect sizes. Within our dataset there were no donors with rs671 and only ten with rs1229984. Despite our low power, the association between rs1229984 and signature B was nominally significant (P -value = 0.05) thanks to the strong effect of the allele. We also found a significant association for a SNP in *ALDH2* (rs4767364) associated with upper aerodigestive tract cancer (McKay et al, *PLoS Genet.*, 2011, PMID:21437268). As explained in answer #9, this SNP showed association with signature B even after multiple hypothesis correction ($P = 0.0001$, $q = 0.03$), supporting that the association between rs4767364 and upper aerodigestive tract cancer reported by McKay et al is driven by higher sensitivity to alcohol. Of interest, contrary to the Asian-specific SNPs (rs671 in *ALDH2* and rs1229984 in *ADH1B*), this SNP rs4767364 shows moderate to high allele frequencies across all populations (0.30 in our cohort).

The reviewer also makes an excellent suggestion in requesting a power assessment, as this can help interpret any negative results. In the revision, we have included power simulations for several variables. Statistical power depends on several factors, including the cohort size, the frequency of the risk allele, and the shape of the distribution of each variable. To simulate power for variables with different distributions, we used a bootstrapping approach. Briefly, from each variable of interest, and for a range of effect sizes (r) and risk allele frequencies (p), we sampled with replacement from the original variable and we multiplied a fraction of them (p) by the simulated effect size (r). For each combination of effect

size and SNP fraction we performed 200 simulations, reporting power as the fraction of significant tests ($\alpha < 0.05$ -to show our power for epidemiological tests- or $\alpha < 5e-8$ -to show power at the genome-wide significance cutoff for GWAS-). The figures below (included in the manuscript as **Extended Data Fig. 9e**) show 2-dimensional grids for the estimated power at each combination of effect size and SNP fraction for three important and representative variables in our study (mutation burden, *NOTCH1* driver mutation frequency, and *TP53* driver mutation frequency).

As discussed above, these analyses show that we are better powered to detect associations with mutation rates than selection on single genes (as the former is inferred from all genes whereas the latter relies on data from individual genes). But they also reveal that our dataset is well powered to identify sufficiently strong associations at a genome-wide significance level. For example, the *ALDH2* and *ADH1B* risk alleles mentioned above appear to increase SBS16 burden in normal oesophagus more than 2-fold (based on limited data from Yokoyama et al, *Nature*, 2019, PMID:30602793). We should be powered to detect similar effect sizes even for relatively rare SNPs (>0.5%).

Actions: These analyses and results are described in detail at the end of **Supplementary Note 7** and in **Extended Data Fig. 9e**. We also mention them in the revised main text. We thank the reviewer for these useful suggestions.

Minor:

1. There is an open parenthesis on line 31 without a closed parenthesis in the sentence.

Authors' response: Thank you, we have corrected this.

2. “In vivo” on line 45, “in vitro” on line 61, etc, should italicized per convention.

Authors' response: Thank you. We would normally use italics but we are following Nature’s convention not to italicise these words.

3. Fig 1e shows a linear association with age and mutation burden. Is this the same for driver mutations? For VAF>2% driver mutations in blood, there is a logarithmic relationship with age.

Authors' response: That is correct. Mutation burden increases linearly in both blood (**Fig. 1e**) and buccal epithelium (**Fig. 2b**). However, driver mutations behave differently in blood (as the reviewer points out), but not in buccal swabs (see **Fig. 4a** and **Supplementary Note 6** for a detailed description of these differences and for the underlying clonal dynamics).

4. In the dataset with buccal swabs, 37% were smokers. This should be sufficiently powered to compare genic selection between smokers and non-smokers indicating selectogenicity effects.

Authors' response: Thank you. Yes, that is indeed the case. This is shown in the main text regression models (**Fig. 4e**), in the dN/dS analyses by smoking group (**Fig. 4f**), and in the extended regression models for selectogenesis (**Extended Data Fig. 9d**). This has been emphasised in the revision to avoid confusion.

5. Based on Fig 2d, it appears that most with buccal swabs had multiple nonsynonymous mutations in known driver genes. Subsetting to those inferred to be driver mutations would provide further clarity.

Authors' response: Yes, that is correct. We detect a large number of driver mutations per donor. For example, we detected an average of ~20 non-synonymous mutations in *NOTCH1* per donor, ~8-9 *TP53* and *FAT1* mutations per donor, etc. These numbers are also available from **Fig. 2e**, which show the total number of mutations seen per gene across the entire cohort of 1,042 donors. The vast majority of these mutations are driver mutations under positive selection (e.g. dN/dS ratios show that ~90% and ~97% of missense and nonsense mutations, respectively, in *NOTCH1* are driver mutations under selection, and ~95% and ~98% for *TP53*).

For clarity, the fraction of non-synonymous mutations that are driver mutations is already taken into consideration in most analyses in the paper, including in the estimated % of cells with driver mutations for each gene (**Fig. 2g**) and in the epidemiological regressions in **Fig. 4**. This is described in detail in the **Methods** and **Supplementary Material**, following a previously described strategy (e.g. Martincorena et al, *Cell*, 2017, PMID:29056346; Lawson et al, *Science*, 2020, PMID:33004514).

6. Natural premature menopause (i.e., premature ovarian failure), but not surgical premature menopause, has been associated with a greater prevalence of CH based on WES/WGS. If age of menopause is available, then it would be interesting to see if mutational burden is increased among women with natural premature menopause. The authors note that overall mutational burden was not different by sex but there may be a difference with this subgroup.

Authors' response: We thank the reviewer for this interesting suggestion. To address this question, we requested information on the age of menopause from the TwinsUK registry. Age of menopause was available for 693 donors in our dataset. We then performed regression analyses (using mixed-effect models to account for the twin structure in the data) between the age of menopause and the oral mutation landscape. No significant associations were found between age of menopause and mutation rates or driver frequencies of the oral epithelium (SNV burden, SigA and SigB burden, DNV burden, *NOTCH1* or *TP53* frequencies).

We thank the reviewer for their comments and suggestions, which we think have helped us improve and strengthen the manuscript.

Referee #3 (Remarks to the Author):

Lawson et al performed single-molecule sequencing to describe the landscape of somatic mutations in over 1,000 individuals of oral epithelium and 371 blood samples. They reported a new version of NanoSeq, which improve the whole-genome coverage to allow genome-wide, somatic mutation profiling in normal tissues. They reported additional genes and mutations under selection of clonal expansion through this large cohort. They also claimed to identify more somatic mutations in the blood given their ability to detect ultra-low VAF mutations from single molecule sequencing. The strength of the study is generating a large cohort of sequencing of normal oral tissue, and the analyses cover broad topics, from detecting new drivers to linking mutational signature with risk factors to infer somatic heritability. However, the novelty of the study is not clearly described, with some results are over interpreted and some conclusions require further investigation.

Major issues

1. The first part of the paper, developing a genome-wide single-molecule mutation detection protocol, is an extension of Abascal et. al. 2021 study, with increased whole-genome coverage compared to the old version of NanoSeq. However, Figure 1 was not focused on comparing whole-genome coverage between the old version and new version of NanoSeq, but instead, focused on comparing mutation detection accuracy of new NanoSeq to standard duplex sequencing, which was largely demonstrated in the 2021 paper using old NanoSeq. Moreover, in the second part of the study, the analyses were mostly based on targeted NanoSeq sequencing, so the advantage of whole-genome NanoSeq is very unclear. Similarly, they showed that NanoSeq can accurately detect low VAF mutations from FFPE samples, but the advantage appears coming from using ddBTPs, which was already in the old version. Together, the first part does not read as reporting a new technology, but showing additional benchmarking of NanoSeq, which was mostly reported in the 2021 study, with minimal improvement on the genome-wide coverage. This makes the novelty of new NanoSeq very confusing to the readers.

Authors' response: Thank you for these comments, which have helped us make this section clearer.

To clarify, a key limitation of the 2021 protocol was the fact that it only covered ~30% of the genome. Although this was sufficient to provide accurate estimates of mutation rates and mutation signatures using shallow whole-genome sequencing, the 2021 protocol was not appropriate for targeted sequencing and driver discovery as it only covered a fraction of the coding sequences of interest. A new NanoSeq protocol with full genome representation AND error rates $<5e-9$ errors/bp was needed to enable comprehensive and accurate studies of mutation rates and driver landscapes in any tissue, which are the two protocols that we introduce here. The reason why we focused **Fig. 1** on error rates is because most duplex sequencing protocols designed for targeted capture already provide genome-wide representation (prior to bait capture), but they do so with error rates around or above $\sim 1e-7$ errors/bp. This remains a key limitation of standard duplex sequencing methods, as this error rate is comparable to the mutation load of somatic tissues. The key to our new protocols is that they provide full genome representation in the library (pre capture) AND error rates $<5e-9$ errors/bp, for the first time enabling accurate whole-genome single-molecule sequencing as well as accurate targeted and whole-exome single-molecule sequencing. We understand that the confusion may have arisen from us using whole-genome libraries (pre-capture) to compare error rates. This is simply because error rates are more accurately measured with whole-genome data, but the accuracy applies to targeted libraries since errors are fixed before capture.

To address the reviewer comment and better clarify the novelty of the new protocols, we have made several changes in the revision.

1. We have improved the main text to better clarify the technical advances.
2. We have added two new figures (**Extended Data Fig. 1a,b**, reproduced below) to show the improvement in coverage across the genome, which we had not shown in the original submission (we thank the reviewer for this constructive suggestion).

3. We now provide a detailed description of the extent of R&D that was required to develop the two protocols introduced in the paper (see below).

In the manuscript, we introduced two versions of NanoSeq with full genome representation, one using sonication and exonuclease blunting, and another using random enzymatic fragmentation. It is important to note that both required extensive development over the span of 2 years to achieve genome-wide representation and low error rates, including in our FFPE samples. However, we failed to describe the extent of R&D and technical advances in the original text. The text below is an excerpt of the revised **Supplementary Note 1**, describing the technical innovations required to develop these two protocols.

In the original version of NanoSeq (Abascal et al. 2021), blunt-end restriction enzymes were used for genome fragmentation without end repair, which ensured error rates $<5 \times 10^{-9}$ errors/bp but led to partial (~30%) coverage of the genome. In the current study, we introduce two alternative genome fragmentation methods that provide full genome coverage whilst retaining the original error rates (MB-NanoSeq and US-NanoSeq). These developments required two years of extensive R&D. The paragraphs below describe some of this work and the challenges overcome:

MB-NanoSeq (sonication NanoSeq): The potential to use sonication and exonuclease digestion in NanoSeq was briefly introduced in our original NanoSeq publication (Abascal et al. 2021). However, the original proof-of-principle example protocol had very low library yields, approximately 2-10% of those obtained with the restriction enzyme method. Optimising library yields while retaining the $<5 \times 10^{-9}$ error rate required extensive testing and optimisation, including testing error rates in different sonication buffers for frozen and formalin-fixed DNA, testing different exonucleases for blunting, different enzyme concentrations, different adapter concentrations, and different DNA ligases and ligation conditions. After considerable optimisation we arrived at the protocol described here, which has similar and often greater library conversion efficiencies than the 2021 restriction enzyme protocol while providing full genome coverage and error rates $<5 \times 10^{-9}$ errors/bp. Amongst other changes, the mung bean nuclease concentration was increased to 5 units/reaction, the phosphorylation and A-tailing reactions were combined into a single reaction and the units of each enzyme were increased, T4 DNA ligase was substituted with NEB Ultra II and the adapter concentration was increased. Background noise in the qPCR reaction, resulting from adapter dimers, was reduced by diluting the ligation reaction prior to stringent SPRI clean-up. This protocol has been pipelined at the Sanger Institute for automated library preparation in 96-well plates, and is the version of NanoSeq used for the sequencing of buccal swabs and blood samples in the present study.

*US-NanoSeq (enzymatic NanoSeq): With the aim of increasing yields further, we have also developed enzymatic targeted NanoSeq. Achieving $<5 \times 10^{-9}$ errors/bp using random enzymatic fragmentation required extensive optimisation. We first tested a commonly used enzymatic fragmentation kit called NEBNext Ultra II FS, which resulted in high error rates around 10^{-5} errors/bp, suggestive of the presence of polymerase activity in the fragmentation mix. We then tested NEB UltraShear, an enzymatic fragmentation mix that is formulated specifically to avoid error introduction during fragmentation. The standard UltraShear workflow utilises traditional end-repair processes. UltraShear followed by standard end repair resulted in excessive error rates (as shown in **Fig. 1**). We thus tested a number of alternative solutions, including removing end repair altogether (relying on pre-existing blunt ends for A-tailing and ligation), including or excluding Mung Bean nuclease to increase the blunting of overhangs, testing error rates with and without SPRI clean-up of the UltraShear buffer, and testing UltraShear in alternative fragmentation buffers to minimise any undesired copying of errors between strands. We concluded that UltraShear in UltraShear buffer, followed by SPRI cleanup of the UltraShear buffer (with or without Mung Bean) and A-tailing in the presence of ddBTPs without additional end repair greatly reduced error rates in frozen cord blood DNA, but still led to higher than optimal error rates. This was particularly visible in the formalin-*

damaged DNA, where an error rate on the order of $\sim 2e-8$ errors/bp and error-derived strand asymmetries were seen. Further R&D led to replacing UltraShear buffer with NEB's r1.1 buffer, which finally removed the low-level transfer of errors between strands, achieving NanoSeq-like error rates on both frozen and our formalin-damaged DNA. This high accuracy came at a cost of increased strand dropout (loss of one strand due to ssDNA breaks). After further R&D, this was solved by the incorporation of NAD⁺ in the r1.1 buffer, leading to the enzymatic fragmentation protocol that we describe below. (...)

Optimisation of target capture of NanoSeq libraries: Additional R&D was required to optimise the library quantification and bottleneck size for targeted sequencing. Although we introduced the qPCR and bottleneck steps in the 2021 protocol, new improvements in the current manuscript include: (1) improved equations for modelling sequencing efficiency (see below), which help ensure optimal duplicate rates on any input DNA and gene panel (including exomes), and (2) optimisation of the amount of amplified library (ng) per fmol of original library to be inputted into bait capture. The latter is important as enough copies of each original molecule need to be used for bait capture to reliably obtain sequencing data from both strands of DNA. By optimising mass-to-fmol ratio, we maximised our ability to multiplex libraries into hybrid capture reactions, hence reducing the overall cost of large cohort projects.

Action: We thank the reviewer for their constructive comment which has helped us improve this section to better explain the technical advance and the challenges overcome with the two new protocols. As explained above, we have improved the main text to clarify these advances, we have added two new supplementary figures to show the improvements in coverage compared to the 2021 protocol (**Extended Data Fig. 1 a,b**), and we have added the text above to **Supplementary Note 1** to better explain the R&D work required.

As we explain above, to our knowledge these are the first protocols with full genome coverage and error rates $< 5e-9$ errors/bp compatible with deep targeted sequencing. This is a key advance in error-corrected sequencing, as they are the first protocols to enable accurate studies of mutation rates, mutation signatures and driver landscapes on any tissue. And, as we explain above, developing these protocols was not trivial after our 2021 paper. Importantly, the impact of the two new fragmentation protocols that we introduce is likely to extend beyond NanoSeq or duplex sequencing. This is because the new protocols should be compatible with other error-corrected sequencing methods that rely on sequencing both strands of DNA (such as CODEC, HiDEF-seq or SMM-seq). For example, CODEC (Bae et al., *Nature Genetics*, 2023, PMID:37106072) and HiDEF-seq (Hong Liu et al., *Nature*, 2024, PMID:38867045) already used some of the improvements in the 2021 NanoSeq paper to achieve lower error rates (e.g. restriction enzymes and ddBTPs). We expect that the two new fragmentation protocols introduced here could be readily adopted by these and other methods to achieve lower error rates with full genome coverage. We have also made this clearer in the revised manuscript.

2. The strength of the paper is oral epithelium bulk sequencing at scale to detect ultra low VAF mutations and understand their selection, but the highlights of the main findings are mostly confirming the role of NOTCH1 in clonal expanding and TP53 in driving tumor development, which is known (PMID: 36658434; PMID: 36266286). The study should clarify the novel findings. In addition, the novelty of the study might be increased if the authors can dive into some new pathogenic variants identified through selection, and using cancer data to provide insights into their functional impacts.

Authors' response: We thank the reviewer for highlighting this.

A challenge for us when writing this manuscript was to summarise a large amount of data, new analyses and new concepts within the tight space constraints. This included describing the new method, the study design, the driver discovery and positive selection results, the negative selection analyses, the high-resolution maps of selection within genes, the non-coding drivers identified, the VUS annotation, and the mutational epidemiology analyses (including the new concept and analyses around selectogenesis). As a result, in each section we focused on the most statistically significant results, leaving a lot of material for the supplementary text. In retrospect, this may have downplayed the novelty of the dataset. For example, as the reviewer notes, in the driver discovery section we focused the attention on the top few drivers and on the similarities with skin and oesophagus, failing to emphasise the novelty of the results (e.g. the vast majority of the 46 drivers identified in oral epithelium have never been reported under selection in normal tissues).

In the revision, we have tried to clarify and expand on the novelty of some of our findings, including regarding driver discovery. This includes:

1. Clarifying the differences (as well as similarities) between oral and oesophageal epithelia. Although in the original manuscript we focused on the similarities between the oral landscape and the oesophageal landscape (which downplayed the novelty of our results), there are also notable differences. For example: (1) the driver density is remarkably higher in oesophagus (e.g. *NOTCH1* clones occupy over 30-40% of the ageing epithelium compared to ~6-12% in the oral epithelium), and (2) the strength of selection in some genes varies starkly (e.g. *KMT2D*, *NFE2L2* and *PIK3CA* are strong drivers in normal oesophagus but neutral or very weakly selected in oral epithelium).
2. Importantly, in the current manuscript we report 49 driver genes in buccal swabs, of which in 3 genes (*DNMT3A*, *TET2* and *FOXP1*) the signal of selection was likely attributable to low-level blood contamination. Of the 46 drivers in oral epithelium, 31 have not been described in skin or oesophageal epithelium, and the majority of them have not been described under selection in any normal tissue, to the best of our knowledge. We think that this is the largest addition of new genes under selection in normal tissues from a single study to date. These 31 genes are also clinically relevant, including multiple drivers of oral and head and neck cancers, which now emerge as being under selection in normal oral epithelium (e.g. *RHOA*, *RAC1*, *EPHA2*, *ZNF750*, *HLA-B*, *EP300* and *KDM6A* -all of which are significant in TCGA HNSC tumours by dNdScv-). In the revision, we have highlighted these points in the main text and we have added a new supplementary section with details on all the genes discovered under positive selection in the oral epithelium.
3. The absence of key drivers is an equally important result, which we failed to adequately highlight in the submission. An intriguing observation in previous studies on oesophagus was that some of the most common driver genes in oesophageal cancers were not detected as significantly mutated in normal oesophagus. However, whether this was due to insufficient statistical power was unclear given the small size of previous datasets. The very high depth of the current study provides a much clearer distinction between the selection landscape of normal oral epithelium and oral cancers or head-and-neck cancers. For example, we find that mutations in *CDKN2A*, *NFE2L2*, *PTEN*, *HLA-A*, *SMAD4*, *B2M* and *RBI*, among others, are remarkably neutral (or very weakly selected) in oral epithelium despite being common drivers in HNSC. This suggests that selection on these genes is likely a later event in HNSC development, potentially in combination with other mutations. It is interesting that this list includes genes like *HLA-A* and *B2M*, which are believed to be immune escape genes, and so may be expected to be selected later in carcinogenesis, and not as first hits. Also interestingly, *CDKN2A* loss (which may enable growing clones to escape cellular senescence) and *SMAD4* loss are also known as relatively late events in colorectal, pancreatic and oesophageal adenocarcinoma evolution, rather than as first hits (e.g. Weaver et al, *Nat Genet.*, 2014, PMID:24952744; Papageorgis et al, *Cancer Res.*, 2011, PMID:21245094; Cowan and Maitra, *Cancer J.*, 2014, PMID:24445769). Thus, the high depth of our study not only has identified many novel drivers in normal epithelium, but it provides suggestive information on whether key cancer drivers may act as first or later hits.

- Finally, our study provides the largest collection of driver mutations from a single study, enabling high-resolution analyses of selection within key cancer genes. This includes >20,000 mutations in *NOTCH1* and >8,000 in *TP53*, with 24 driver genes having >1,000 mutations per gene. To study selection on single sites and in functionally related groups of sites, we also introduce new dN/dS algorithms in this paper, leading to the discovery of 1,220 amino acid changes under significant positive selection, including new non-coding and synonymous driver sites in key driver genes. Importantly, given the reviewer's comment, this also allowed us to inform on the clinical relevance of many variants of uncertain significance (VUS). In the revised manuscript, and as requested by the reviewer, we have expanded on these results, both in the main text and in an extended **Supplementary Note 4**.

Action: We thank the reviewer for this constructive comment, which has motivated us to better describe the novelty of the driver discovery results. We describe the novelty of the results more clearly in the main text (within the space constraints). We have added a new figure (**Extended Data Fig. 5e**), highlighting the differential selection of some genes in HNSC vs oral epithelium. We have also greatly extended the **Supplementary Note 4** with many more details on the pattern of selection across genes and within individual genes (including a new supporting figure showing the concentration of selected sites in the forkhead-associated domain of *CHEK2* in **Extended Data Fig. 6j**).

New **Extended Data Fig. 5e** and **Extended Data Fig. 6j**, complementing the in-depth description of our driver results in the extended **Supplementary Note 4**.

3. The authors try to investigate the role of environment-induced mutagenesis vs selection on increased cancer risk. However, it is unclear if the current cohort is sufficient to answer the question. Drivers with increased VAF can be difficult to detect from sequencing solid tissue due to sampling, which may explain the results in Fig4a, where large clones can be more easily detected from blood sample. To suggest stronger effect of mutation burden than selection, multi-sampling or comparing normal vs premalignant lesions are needed.

Authors' response: We thank the reviewer for this question, which has considerably helped us improve how we compare the balance between mutagenesis and selectogenesis.

Before describing the new analyses, we think that there are several clarifications needed to address this point:

1. The difference between oral epithelium and blood in **Fig. 4a** is not due to sampling but it reflects a key difference in the nature of the clonal landscape between blood and solid tissues. Whereas in blood some clones can grow almost exponentially to account for a significant fraction of the stem cell pool, clonal growth is highly constrained in solid tissues, with large numbers of typically microscopic clones dominating the landscape. We and others have previously described this pattern in other solid tissues, including skin (Martincorena et al, *Science*, 2015, PMID:25999502), oesophagus (Martincorena et al, *Science*, 2018, PMID:30337457), or bladder (Lawson et al, *Science*, 2020 PMID:33004514). Given that buccal swabs sample an area of the mouth (typically several cm²) rather than the entire mouth, they can indeed miss a large clone elsewhere. However, if large clones were relatively common (as they are in blood), by sampling several square centimeters of tissue across 1,042 donors, we should have been able to find many of them. Their paucity in our dataset shows that such clones are indeed very rare in normal oral epithelium in the general population. Thus, the difference between blood and oral epithelium in **Fig. 4a** is not due to sampling but to the extreme polyclonality of oral epithelium compared to blood (consistent with previous studies in other solid tissues).
2. Since we can separately quantify mutation rates and selection in our data, systematic studies of polyclonal landscapes can start to disentangle the mode of action of different carcinogens, answering whether a carcinogen (or risk factor) acts by inducing mutations (mutagens) or promoting clonal expansions (selectogens or promoters). To help formalise this under a well established mathematical framework, in the **Supplementary Notes 5 and 6** we use classical multi-stage models to explain how an increase in mutation rates or an increase in clone sizes is expected to impact on cancer risk. These models emphasise that whereas the risk of transformation of an individual microscopic clone is very small, the aggregate number of cells with driver mutations is expected to determine or influence cancer risk. For example, *TP53* mutations are a key step in the development of oral cancers and we find them in a few percent of all oral epithelial cells in an average middle-age individual. Under a multistage model, carcinogenic exposures that double the mutation rate (mutagens) or that double the size (or frequency) of *TP53* clones (selectogens) are expected to double cancer risk, as they double the number of cells at risk of subsequent hits and transformation. It is these effects that have remained largely unmeasurable to date but that we can now quantify with a technology like NanoSeq, as we can now systematically measure mutation rates and mutant cell frequencies in large cohorts of individuals exposed to a variety of risk factors.
3. We hope that the points above clarify the importance of studying mutation rates and selection even on microscopic clones. The reviewer, however, may be referring more specifically to the challenge of studying selectogenesis on rare but large premalignant lesions. If an individual has a large dysplastic lesion on one side of the cheek, this may not be detected by sampling the other cheek. Of course, we acknowledge that this is the case and that this is a limitation of buccal swabs for early detection (saliva or mouth rinses may be preferable for a sensitive detection of every large clone). However, we emphasise that the mutagenic and selectogenic effects that we are trying to quantify focus on an earlier stage of the carcinogenic process (i.e. effects on clones in histologically normal tissue). Other designs, such as systematic sequencing of large collections of premalignant lesions, would be needed to quantify selectogenic effects in later, more advanced, precancerous lesions. We have added a clarification in the main text to highlight that our claims refer to carcinogenic effects in very early clones.

New analyses: Having clarified the points above, we want to mention that in the revision we have expanded and refined our quantification of mutagenic and selectogenic effects on the normal oral landscape. Motivated by reviewers 2 and 3, we have added power simulations to our study, which confirm that we have more statistical power to detect mutagenic associations than selectogenic associations. This is because we have more mutations per donor to estimate mutation rates than to estimate selection at a single gene level. Thus, to more fairly compare the extent of mutagenic and selectogenic effects, we now report both effect sizes and *P*-values (e.g. see the new **Fig. 4f** below, which shows side-by-side the effect sizes of the mutagenic and selectogenic effects of smoking on *NOTCH1* and *TP53*). We think that these changes provide a more balanced and fair comparison of mutagenesis and selectogenesis, showing that smoking causes changes in mutation rates and in selection (clonal

expansion) to a similar degree. Whereas these results were already present in the original manuscript and they have not changed, we think that they are better presented in the revised manuscript.

Action: In the revised manuscript, we have clarified the points above. This includes changes in the main text, to increase clarity and to acknowledge the importance of statistical power in the epidemiological regressions. We have also modified **Fig. 4f** (pasted below) to show the effect sizes of mutagenic and selectogenic associations (note that an increase in dN/dS is linearly related to an increase in the mutant cell fraction due to selection, as explained in **Supplementary Note 2**). Finally, to avoid confusion, we now clarify in the main text that our analyses on mutagenesis and selectogenesis are restricted to effects on clones present in histologically normal tissues, and so on effects in very early carcinogenesis.

4. Although the concept that linking somatic mutations in normal tissue with epidemiological factors to improve cancer risk prediction is interesting, the study did not clearly show, for example, how knowing the clonal landscape related to smoking and alcohol use can improve the prediction beyond having smoking and alcohol drinking data. Moreover, from the clinical perspective, the advantage of clonal profiling using NanoSeq, compared to existing cancer screening approaches, is not clearly demonstrated, perhaps given limited clinical information of the study cohort. However, as the investigators did not build any cancer risk prediction model, the clinical utility and significance should be further clarified in the text.

Authors' response: We thank the reviewer for this interesting comment, which we think requires a clarification. We think that a technology like targeted NanoSeq now allows large cohort studies to start building mechanistic risk models connecting exposures to cancer risk through their effects on clonal landscapes. We only mentioned this idea briefly in the Discussion due to space constraints, but we think that such models can be built in two stages (see **Fig. 4j**, pasted below): (1) models connecting cancer risk factors to changes in mutation rates and clonal expansions, and (2) models connecting changes in mutation rates and selection with changes in cancer risk.

In our study, we have attempted to show how large-scale studies of the mutation and clonal landscape in healthy individuals with variable cancer risk factors can shed light on step 1. However, in order to build models for step 2, we would need case-control designs (e.g. longitudinal samples of buccal swabs before and after a cancer diagnosis, or cross-sectional case-control studies using contralateral sampling of the oral epithelium in patients just diagnosed with an oral cancer). We have initiated efforts in this direction (e.g. a case-control study of buccal swabs from Taiwan in collaboration with NIH and WHO-IARC). However, the current study was designed to inform step 1, studying the impact of risk factors on mutation rates and clonal landscapes in cancer-free individuals to provide mechanistic insights into their mode of action rather than to correlate changes in the clonal landscape with cancer risk.

Action: We have clarified this point in the revised manuscript.

Minor concerns:

1. Given age differences between cohorts and its importance in mutations, age should be adjusted in several comparisons between their cohort and other cohorts.

Authors' response: Thank you. We think that we have adjusted for age in all analyses and in comparisons to other cohorts, but please do not hesitate to highlight any place where we may have failed to do so.

2. Some references were missing, for example, for the paper in line 138-141.

Authors' response: Thank you very much for this and for the previous comments, which we think have helped us improve the manuscript.

Referee #4 (Remarks to the Author):

In this manuscript, Lawson et al report an upgraded version of NanoSeq, a duplex sequencing method with an ultra-low error rate and full genome coverage, making it a powerful tool for studying somatic mutation and selection in normal tissues, particularly polyclonal tissues. The authors applied this method to buccal swab samples (oral epithelium) collected from a large cohort of twin participants, which reveals a rich landscape of somatic mutation and selection in human normal tissues at epidemiological scale for the first time.

The study sets up a new paradigm for the future research on somatic mutagenesis in normal tissues. It is impressive, novel and impactful in several key aspects: (1) Previous studies of somatic mutation in solid normal tissues were limited to small cohorts, limiting insights into mutation selection across diverse populations. The current study for the first time pushed the boundary to epidemiological scale and reveal groundbreaking findings on the influence of germline and environmental factors on somatic mutation and selection. This advance was enabled by NanoSeq, which allows for accurate detection of low-frequency somatic mutations in polyclonal tissues at a lower cost and without labor-intensive processes. Besides, buccal swab collection is non-invasive and logistically feasible, allowing for the recruitment of a large cohort without significant burden on participants. (2) It is a herculean effort to recruit the cohort, systematically collect data and metadata, and conduct comprehensive and rigorous analyses. Perhaps, this is not surprising, as it is consistent with the high-quality standards of publications from this group. (3) Most importantly, the study reports many novel findings and has conceptually advanced our knowledge of somatic mutation and selection in normal tissues. For example, negative selection was found to be weak or missing in previous studies. The current study leveraged the incredibly high mutation density across each gene and identified significant negative selections on 9 genes. (4) In addition, the manuscript is well-structured and well-written, equipped with incredibly detailed supplementary notes.

Authors' response: Thank you, we are very grateful for the positive assessment of our manuscript.

I believe the following points should be addressed and clarified before publication.

In the buccal swab cohort, 37% participants are smokers. Given this, why was SBS4 not detected in the mutational signature analysis? It is associated with tobacco smoking and commonly observed in oral cancer. Additionally, why did the authors choose to use Sigfit instead of other widely used methods, such as SigProfilerExtractor, for the mutational signature analysis?

Authors' response: Thank you for this comment, which has allowed us to expand on this interesting result, and more generally strengthen the signature analyses in the manuscript.

Actions: The absence of the SBS4 signature is indeed striking and important. To strengthen this part of the manuscript, in the revision we provide several new analyses:

1. **Aggregate spectra:** In the revision, we have added the aggregate spectra across non-drinking never smokers and non-drinking heavy smokers, which supports a striking lack of SBS4 (or SBS92) in heavy smokers. This is now shown in the new **Extended Data Fig. 8c** (also reproduced below). A formal statistical analysis is also described below (bullet point #4).
2. **Absence of SBS4 with SigFit and SigProfiler:** in the original manuscript, we performed signature extractions with SigFit, exploring solutions with 2 to 7 signatures. Deconvolutions with more than 2 signatures did not significantly improve the goodness-of-fit, and visual inspection of the additional signatures did not reveal any hint of SBS4 or other potentially relevant signatures like APOBEC (SBS2 and SBS13). Following this reviewer comment, we have repeated the deconvolution using SigProfiler. SigProfiler also concluded that the optimal solution is 2 signatures, which were virtually identical to those obtained with SigFit. We also

confirm that solutions with more signatures (with much lower stability) did not provide any hint of SBS4/2/13.

- Lack of smoking indel signatures:** SBS4 is known to be associated with the COSMIC indel signature ID3 and the COSMIC double base substitution signature DBS2, but we found no evidence of these (**Extended Data Fig. 8d,e**), comparing heavy smokers vs non-smokers. This provides further suggestive evidence for the absence of SBS4.
- Likelihood-Ratio Tests (LRT) for the presence of signatures:** To complement the *de novo* mutation signature discovery with a more sensitive approach, we used a supervised approach with a sensitive statistical test. Briefly, for each sample with ≥ 200 mutations (except the donor with CHOP chemotherapy), we compared the likelihood of a model fitted with the two observed signatures (SigA+SigB) to a model including an additional COSMIC signature using a LRT with 1 degree of freedom. Expectation-Maximisation was used to obtain the maximum likelihood estimates for the contribution of each signature in both models. This test was run across all samples and for COSMIC signatures SBS2 (APOBEC), SBS13 (APOBEC), SBS4 (smoking), SBS18 (reactive oxygen species) and SBS92 (smoking). The resulting *P*-values were subject to Benjamini–Hochberg multiple testing correction. No significant evidence of any of these signatures was found in any of the samples. As a caveat to this analysis, we note that the extracted SigA and SigB signatures could theoretically contain low-level contributions of some of the signatures. Whereas small contributions of these signatures below our detection sensitivity remain possible, these new analyses, together with the aggregated spectra across never-smokers and heavy-smokers, strongly support a remarkable absence or paucity of SBS4 (as well as SBS92 and APOBEC signatures) in our data.

New Extended Data Fig. 8c.

The lack of SBS4 mutations is striking but seems consistent with recent cancer genomic data from head and neck tumours, which suggest that SBS4 contributes comparatively little to oral cancers compared to laryngeal cancers or lung cancers (Torrens et al, *medRxiv*, 2024, PMID:38699364). Possible explanations for the lack of SBS4 may include: (1) a protective role of the multilayered squamous epithelia, which may reduce the exposure of basal stem cells to benzo[a]pyrene, compared to the more exposed stem cells in the thin epithelia of the larynx or the lung, and (2) a lower expression of *CYP1A1* in oral epithelium than lung and larynx, which is the main metabolizer of benzo[a]pyrene, and required for its mutagenic effects (Torrens et al, *medRxiv*, 2024, PMID:38699364).

Remarkably, our results instead show strong associations between smoking and the rates of both SBS5 and SBS16 (the latter probably indirectly through alcohol, see **Supplementary Note 7**). The molecular mechanisms behind SBS5 are unknown, so the basis for its association with smoking in the oral epithelium is unclear, but we discuss some possibilities in **Supplementary Notes 3 and 7**.

Taken together, our results suggest that smoking seems to increase oral cancer risk through an increase in SBS5 and SBS16 mutagenesis and, possibly, through selectogenic effects on clonal expansions (see **Fig. 4f, Extended Data Fig. 9d**), rather than through the classical SBS4 and SBS92 mutagenesis.

We thank the reviewer for this helpful comment.

In the regression analysis, sex was not significantly associated with differences in mutation rates, signatures or driver densities when correcting for confounders. Can the authors discuss or speculate on this finding, especially given the known sex bias in oral cancer?

Authors' response: Thank you for the interesting comment. The possibility of a difference in somatic mutation rates between men and women was a question that we were particularly interested in, given the higher incidence of oral cancer (and of several other cancers) in men, as well as their lower life expectancy in most developed countries. As the reviewer notes, no statistically significant difference was found. Our new power calculations suggest that we should have been powered to detect a difference >5% in the rate of SNVs/year between men and women (see new **Extended Data Fig. 9e**, reproduced below), which sets an approximate upper bound for a possible difference in somatic mutation rates between men and women, at least in the oral epithelium.

The simplest explanation for the higher oral cancer incidence between men and women is thus likely environmental. This appears to be supported by epidemiological studies in related cancer types, such as oesophageal SCCs, which estimate that tobacco and alcohol consumption alone “account for nearly all of the observed differences in the incidence of esophageal SCC between men and women” (Pelucchi et al, *Alcohol Res Health*, 2006, PMID:17373408).

Lines 162-163: the authors performed whole-genome NanoSeq on 16 samples to the genome-wide mutation rate. Why did they use the original restriction enzyme protocol instead of the new protocols (according to the description in Supplementary Note 3). In general, I feel the nomenclature could be clearer, especially when describing the comparison between previous and current versions of NanoSeq.

Authors' response: We thank the reviewer for raising this question and we apologise for the confusion.

Following this suggestion, we have tried to make the nomenclature clearer. There are currently three versions of NanoSeq: (1) the 2021 restriction enzyme NanoSeq protocol (Abascal et al, *Nature*, 2021, PMID:33911282), which we now refer to as RE-NanoSeq when comparing protocols, (2) the new sonication + mung bean nuclease NanoSeq (MB-NanoSeq), which is the protocol used in the buccal swabs and the blood samples, and (3) the new random enzymatic fragmentation protocol using Ultrasear (US-NanoSeq), which is only shown in **Fig. 1** and in the **Supplementary Material**.

The reason why we used RE-NanoSeq for the 16 whole-genome samples was purely pragmatic. RE-NanoSeq is pipelined at the Sanger Institute for shallow whole-genome NanoSeq, whereas at the time of the experiment the MB-NanoSeq protocol had only been pipelined for targeted and whole-exome sequencing. Another practical consideration was that RE-NanoSeq is cheaper for mutation burden analyses than whole-genome MB-NanoSeq. This is because RE-NanoSeq only covers ~30% of the genome, which means that the cost of the matched normal (a NEAT or non-bottlenecked RE-NanoSeq library) is ~30% of a standard matched normal. Since the goal of sequencing these 16 whole-genomes was only to calculate mutation burdens and signatures, at the time we chose RE-NanoSeq for simplicity and cost.

When estimating mutational burdens, has the authors considered the fraction of cells that carry each mutation (by summing VAFs)?

Authors' response: Thank you. Yes, VAFs are implicitly taken into account in the calculation of mutation burdens in NanoSeq. The calling pipeline in NanoSeq makes base calls, using read bundles to call both reference (wild-type) and non-reference (mutant) bases for each original molecule of DNA. Mutation burdens are then calculated as the number of all mutant bases divided by the number of all bases sequenced with duplex information, aggregated across all duplex reads. Thus, a mutation seen in 10 molecules will be counted 10 times, which is mathematically identical to aggregating VAFs. Two important advantages of calculating mutation burdens in this way are: (1) that mutation burdens are unaffected by clonality, and (2) that the same calling filters are applied to reference and non-reference bases offering a natural way to account for the callable genome size, obtaining more accurate mutation burden estimates.

Action: We have added a note in the description of the mutation calling by NanoSeq to acknowledge that VAFs or the fraction of cells carrying a mutation is implicitly taken into account.

Line 85: can the full-genome duplex coverage be shown and compared with the previous NanoSeq version in a plot?

Action: We thank the reviewer for this excellent suggestion and we apologise for not having shown this in the original submission. We have added two new figures (**Extended Data Fig. 1a,b** reproduced below) comparing the partial coverage of RE-NanoSeq with the full genome coverage of MB-NanoSeq and US-NanoSeq. The first figure shows a representative example of the coverage in a given locus of the genome (a region of *TP53*), which highlights the highly uneven and partial coverage of the old RE-NanoSeq protocol for targeted/exome applications. The second figure shows Lorenz curves of the distribution of coverage across sites for the different protocols. We also provide Gini coefficients of the evenness of coverage for the different protocols (~0.66 for RE-NanoSeq, and ~0.20 for MB/US-NanoSeq).

We thank the reviewer for their positive comments and constructive suggestions, which we think have helped us improve the manuscript.